# Action Poisoning Attacks on Linear Contextual Bandits

**Guanlin Liu**                                                                    *glnliu@ucdavis.edu*
*Department of Electrical and Computer Engineering*
*University of California, Davis*

**Lifeng Lai**                                                                     *lflai@ucdavis.edu*
*Department of Electrical and Computer Engineering*
*University of California, Davis*

**Reviewed on OpenReview:** *https:// openreview. net/ forum? id=yhGCKUsKJS*

## Abstract

Contextual bandit algorithms have many applicants in a variety of scenarios. In order to develop trustworthy contextual bandit systems, understanding the impacts of various adversarial attacks on contextual bandit algorithms is essential. In this paper, we propose a new class of attacks: action poisoning attacks, where an adversary can change the action signal selected by the agent. We design action poisoning attack schemes against disjoint linear contextual bandit algorithms in both white-box and black-box settings. We further analyze the cost of the proposed attack strategies for a very popular and widely used bandit algorithm: LinUCB. We show that, in both white-box and black-box settings, the proposed attack schemes can force the LinUCB agent to pull a target arm very frequently by spending only logarithm cost. We also extend the proposed attack strategies to generalized linear models and show the effectiveness of the proposed strategies.

## 1 Introduction

Multiple armed bandits(MABs), a popular framework of sequential decision making model, has been widely investigated and has many applicants in a variety of scenarios (Chapelle et al., 2014; Lai et al., 2011; Kveton et al., 2015). The contextual bandits model is an extension of the multi-armed bandits model with contextual information. At each round, the reward is associated with both the arm (a.k.a, action) and the context. Contextual bandits algorithms have a broad range of applications, such as recommender systems (Li et al., 2010), wireless networks (Saxena et al., 2019), etc.

In the modern industry-scale applications of bandit algorithms, action decisions, reward signal collection, and policy iterations are normally implemented in a distributed network. Action decisions and reward signals may need to be transmitted over communication links. For example, in recommendation systems, the transitions of the decisions and the reward signal rely on a feedback loop between the recommendation system and the user. When data packets containing the reward signals and action decisions etc are transmitted through the network, the adversary can implement adversarial attacks by intercepting and modifying these data packets. As the result, poisoning attacks on contextual bandits could possibly happen. In many applications of contextual bandits, an adversary may have an incentive to force the contextual bandits system to learn a specific policy. For example, a restaurant may attack the bandit systems to force the systems into increasing the restaurant's exposure. Thus, understanding the risks of different kinds of adversarial attacks on contextual bandits is essential for the safe applications of the contextual bandit model and designing robust contextual bandit systems.

Depending on the target of the poisoning attacks, the poisoning attacks against contextual linear bandits can be categorized into four types: reward poisoning attack, action poisoning attack, context poisoning attack and the mix of them. In this paper, we aim to investigate the impact of action poisoning attacks on contextual bandit models. To our best knowledge, this paper is the first work to analyze the impact

of action poisoning attacks on contextual bandit models. Detailed comparisons of various types of attacks against contextual bandits will be provided in Section 3. We note that the goal of this paper is not to promote any particular type of poisoning attack. Rather, our goal is to understand the potential risks of action poisoning attacks. We note that for the safe applications and design of robust contextual bandit algorithms, it is essential to address all possible weaknesses of the models and understanding the risks of different kinds of adversarial attacks. Our hope is that the understanding of the potential risks of action poisoning attacks could motivate follow up research to design algorithms that are robust to such attacks.

In this paper, we study the action poisoning attack against disjoint linear contextual bandit (Li et al., 2010; Kong et al., 2020; Garcelon et al., 2020; Huang et al., 2021) in both white-box and black-box settings. In the white-box setting, we assume that the attacker knows the coefficient vectors associated with arms. Thus, at each round, the attacker knows the mean rewards of all arms. While it is often unrealistic to exactly know the coefficient vectors, the understanding of the white-box attacks could provide valuable insights on how to design the more practical black-box attacks. In the black-box setting, we assume that the attacker has no prior information about the arms and does not know the agent's algorithm. The limited information that the attacker has are the context information, the action signal chosen by the agent, and the reward signal generated from the environment. In both white-box and black-box settings, the attacker aims to manipulate the agent into frequently pulling a target arm chosen by the attacker with a minimum cost. The cost is measublack by the number of rounds that the attacker changes the actions selected by the agent. The contributions of this paper are:

**(1)** We propose a new online action poisoning attack against contextual bandit in which the attacker aims to force the agent to frequently pull a target arm chosen by the attacker via strategically changing the agent's actions.

**(2)** We introduce a white-box attack strategy that can manipulate any sublinear-regret disjoint linear contextual bandit agent into pulling a target arm $T - o(T)$ rounds over a horizon of $T$ rounds, while incurring a cost that is sublinear dependent on $T$.

**(3)** We design a black-box attack strategy whose performance nearly matches that of the white-box attack strategy. We apply the black-box attack strategy against a very popular and widely used bandit algorithm: LinUCB (Li et al., 2010). We show that our proposed attack scheme can force the LinUCB agent into pulling a target arm $T - O(\log^3 T)$ times with attack cost scaling as $O(\log^3 T)$.

**(4)** We extend the proposed attack strategies to generalized linear contextual bandits. We further analyze the cost of the proposed attack strategies for a generalized linear contextual bandit algorithm: UCB-GLM (Li et al., 2017).

**(5)** We evaluate our attack strategies using both synthetic and real datasets. We observe empirically that the total cost of our black-box attack is sublinear for a variety of contextual bandit algorithms.

## 2 Related Work

In this section, we discuss related works on two parts: attacks that cause standard bandit algorithms to fail and robust bandit algorithms that can defend attacks.

**Attacks Models.** While there are many existing works addressing adversarial attacks on supervised learning models (Szegedy et al., 2014; Moosavi-Dezfooli et al., 2017; Cohen et al., 2019; Dohmatob, 2019; Wang et al., 2019; Dasgupta et al., 2019; Cicalese et al., 2020), the understanding of adversarial attacks on contextual bandit models is less complete. Of particular relevance to our work is a line of interesting recent work on adversarial attacks on MABs (Jun et al., 2018; Liu & Shroff, 2019; Liu & Lai, 2020b) and on linear contextual bandits (Ma et al., 2018; Garcelon et al., 2020). In recent works in MABs setting, the types of attacks include both reward poisoning attacks and action poisoning attacks. In the reward poisoning attacks, there is an adversary who can manipulate the reward signal received by the agent (Jun et al., 2018; Liu & Shroff, 2019). In the action poisoning attacks, the adversary can manipulate the action signal chosen by the agent before the environment receives it (Liu & Lai, 2020b). Existing works on adversarial attacks against linear contextual bandits focus on the reward (Ma et al., 2018; Garcelon et al., 2020) or context poisoning

attacks (Garcelon et al., 2020). In the context poisoning attacks, the adversary can modify the context observed by the agent without changing the reward associated with the context. (Wang et al., 2022) defines the concept of attackability of linear stochastic bandits and introduces the sufficient and necessary conditions on attackability. There are also some recent interesting work on adversarial attacks against reinforcement learning (RL) algorithms under various setting (Behzadan & Munir, 2017; Huang & Zhu, 2019; Ma et al., 2019; Zhang et al., 2020; Sun et al., 2021; Rakhsha et al., 2020; 2021; Liu & Lai, 2021; Rangi et al., 2022). Although there are some recent works on the action poisoning attacks against MABs and RL, the action poisoning attack on contextual linear bandit is not a simple extension of the case of MAB or RL. Firstly, in the MAB settings the rewards only depend on the arm (action), while in the contextual bandit setting, the rewards depend on both the arm and the context (state). Secondly, (Liu & Lai, 2021) discusses the action poisoning attack in the tabular RL case where the number of states is finite. In the linear contextual bandit problem, the number of contexts is infinite. These factors make the design of attack strategies and performance analysis for the contextual linear bandit problems much more challenging.

**Robust algorithms.** Lots of efforts have been made to design robust bandit algorithms to defend adversarial attacks. In the MABs setting, (Lykouris et al., 2018) introduces a bandit algorithm, called Multilayer Active Arm Elimination Race algorithm, that is robust to reward poisoning attacks. (Gupta et al., 2019) presents an algorithm named BARBAR that is robust to reward poisoning attacks and the regret of the proposed algorithm is nearly optimal. (Guan et al., 2020) proposes algorithms that are robust to a reward poisoning attack model where an adversary attacks with a certain probability at each round. (Feng et al., 2020) proves that Thompson Sampling, UCB, and $\epsilon$-greedy can be modified to be robust to self-interested reward poisoning attacks. (Liu & Lai, 2020a) introduce a bandit algorithm, called MOUCB, that is robust to action poisoning attacks. The algorithms for the context-free stochastic multi-armed bandit (MAB) settings are not suited for our settings. In the linear contextual bandit setting, (Bogunovic et al., 2021) proposes two variants of stochastic linear bandit algorithm that is robust to reward poisoning attacks, which separately work on known attack budget case and agnostic attack budget case. (Bogunovic et al., 2021) also proves that a simple greedy algorithm based on linear regression can be robust to linear contextual bandits with shablack coefficient under a stringent diversity assumption on the contexts. (Ding et al., 2022) provides a robust linear contextual bandit algorithm that works under both the reward poisoning attacks and context poisoning attacks. (He et al., 2022) provides nearly optimal algorithms for linear contextual bandits with adversarial corruptions.

## 3    Problem Setup

### 3.1    Review of Linear Contextual Bandit

Consider the standard disjoint linear contextual bandit model in which the environment consists of $K$ arms. In each round $t = 1, 2, 3, \ldots, T$, the agent observes a context $x_t \in \mathcal{D}$ where $\mathcal{D} \subset \mathbb{R}^d$, pulls an arm $I_t$ and receives a reward $r_{t,I_t}$. Each arm $i$ is associated with an unknown but fixed coefficient vector $\theta_i \in \Theta$ where $\Theta \subset \mathbb{R}^d$. In each round $t$, the reward satisfies

$$r_{t,I_t} = \langle x_t, \theta_{I_t} \rangle + \eta_t, \tag{1}$$

where $\eta_t$ is a conditionally independent zero-mean $R$-subgaussian noise and $\langle \cdot, \cdot \rangle$ denotes the inner product. Hence, the expected reward of arm $i$ under context $x_t$ follows the linear setting

$$\mathbb{E}[r_{t,i}] = \langle x_t, \theta_i \rangle \tag{2}$$

for all $t$ and all arm $i$. If we consider the $\sigma$-algebra $F_t = \sigma(\eta_1, \ldots, \eta_t)$, $\eta_t$ becomes $F_t$ measurable which is a conditionally independent zero-mean random variable. The agent aims to minimize the cumulative pseudo-regret

$$\bar{R}_T = \sum_{t=1}^{T} \left( \langle x_t, \theta_{I_t^*} \rangle - \langle x_t, \theta_{I_t} \rangle \right),$$

where $I_t^* = \arg\max_i \langle x_t, \theta_i \rangle$.

In this paper, we assume that there exist $L > 0$ and $S > 0$, such that for all round $t$ and arm $i$, $\|x_t\|_2 \leq L$ and $\|\theta_i\|_2 \leq S$, where $\|\cdot\|_2$ denotes the $\ell_2$-norm. We assume that there exist $\mathcal{D} \subset \mathbb{R}^d$ such that for all $x \in \mathcal{D}$ and all arm $i$, $\langle x, \theta_i \rangle > 0$. In addition, for all $t$, $x_t \in \mathcal{D}$.

### 3.2 Action Poisoning Attack Model

In this paper, we introduce a novel adversary setting, in which the attacker can manipulate the action chosen by the agent. In particular, at each round $t$, after the agent chooses an arm $I_t$, the attacker can manipulate the agent's action by changing $I_t$ to another $I_t^0 \in \{1, \ldots, K\}$. If the attacker decides not to attack, $I_t^0 = I_t$. The environment generates a random reward $r_{t,I_t^0}$ based on the post-attack arm $I_t^0$ and the context $x_t$. Then the agent and the attacker receive reward $r_{t,I_t^0}$ from the environment. Since the agent does not know the attacker's manipulations and the presence of the attacker, the agent will still view $r_{t,I_t^0}$ as the reward corresponding to the arm $I_t$. The action poisoning attack model is summarized in Algorithm 1 .

---

**Algorithm 1** Action poisoning attacks on contextual linear bandit agent

---

1: **for** $t = 1, 2, \ldots, T$ **do**
2:     Agent chooses arm $I_t$ after observing the context $x_t$.
3:     Attacker observes the agent's action $I_t$. If the attacker decides to attack, it manipulates the action to $I_t^0$. If the attacker does not attack, $I_t^0 = I_t$.
4:     The environment generates reward $r_{t,I_t^0}$ according to arm $I_t^0$ and context $x_t$.
5:     The agent and attacker receive reward $r_{t,I_t^0}$.
6: **end for**

---

The goal of the attacker is to design attack strategy to manipulate the agent into pulling a target arm very frequently but by making attacks as rarely as possible. Without loss of generality, we assume arm $K$ is the "attack target" arm or target arm. Define the set of rounds when the attacker decides to attack as $\mathcal{C} := \{t : t \leq T, I_t^0 \neq I_t\}$. The cumulative attack cost is the total number of rounds where the attacker decides to attack, i.e., $|\mathcal{C}|$. The attacker can monitor the contexts, the actions of the agent and the reward signals from the environment.

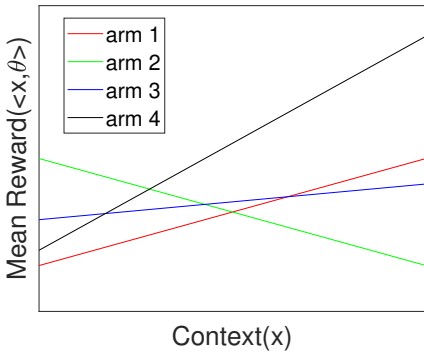

Figure 1: An example of one dimension linear contextual bandit model.

As the action poisoning attack only changes the actions, it can impact but does not have direct control of the agent's observations. Furthermore, when the action space is discrete and finite, the ability of the action poisoning attacker is severely limited. It is reasonable to limit the choice of the target policy. Here we introduce an important assumption that the target arm is not the worst arm:

**Assumption 1.** *For all $x \in \mathcal{D}$, the mean reward of the target arm satisfies $\langle x, \theta_K \rangle > \min_{i \in [K]} \langle x, \theta_i \rangle$.*

If the target arm is the worst arm in most contexts, the attacker should change the target arm to a better arm or the optimal arm so that the agent learns that the target set is optimal for almost every context. In this case, the cost of attack may be up to $O(T)$. Assumption 1 does not imply that the target arm is optimal

at some contexts. The target arm could be sub-optimal for all contexts. Fig. 1 shows an example of one dimension linear contextual bandit model, where the $x$-axis represents the contexts and the $y$-axis represents the mean rewards of arms under different contexts. As shown in Fig. 1, arms 3 and 4 satisfy Assumption 1. In addition, arm 3 is not optimal at any context.

Under Assumption 1, there exists $\alpha \in (0, \frac{1}{2})$ such that

$$\max_{x \in \mathcal{D}} \frac{\min_i \langle x, \theta_i \rangle}{\langle x, \theta_K \rangle} \leq (1 - 2\alpha).$$

Equivalently, Assumption 1 implies that there exists $\alpha \in (0, \frac{1}{2})$, such that for all $t$, we have

$$(1 - 2\alpha)\langle x_t, \theta_K \rangle \geq \min_{i \in [K]} \langle x_t, \theta_i \rangle. \tag{3}$$

Equation 3 describes the relation of the target arm and the worst arm at context $x_t$. The action poisoning attack can only indirectly impacts the agent's rewards. The mean rewards at $x_t$ after the action attacks must be larger or equal to $\min_{i \in [K]} \langle x_t, \theta_i \rangle$. Under Equation 3, $(1 - 2\alpha)\langle x_t, \theta_K \rangle \geq \min_{i \in [K]} \langle x_t, \theta_i \rangle$ at any context $x_t$. Then, with $\langle x_t, \theta_K \rangle > 0$, $(1-\alpha)\langle x_t, \theta_K \rangle > \min_{i \in [K]} \langle x_t, \theta_i \rangle$ at any context $x_t$. Thus, the attacker can indirectly change the agent's mean reward of the non-target arm to $(1 - \alpha)\langle x_t, \theta_K \rangle$. Then, the agent's estimate of each non-target arm's coefficient vector is close to $(1-\alpha)\theta_K$, which is worse than the target arm at any context. Equation 3 brings the possibility of successful action attacks.

Assumption 1 is necessary in our analysis to prove a formal bound of the attack cost. In Appendix A, we show that this assumption is necessary in the sense that, if this assumption does not hold, there exist a sequence of contexts $\{x_t\}_{t \in [T]}$ such that no efficient action poisoning attack scheme can successfully attack LinUCB. Certainly, these sequences of contexts are worst case scenarios for attacks. In practice, if these worst case scenarios do not occur, the proposed algorithms in Sections 4.2 and 4.3 may still work even if the target arm is the worst in a small portion of the contexts (as illustrated in the numerical example section).

### 3.3 Comparisons with Different Poisoning Attacks

We now compare the three types of poisoning attacks against contextual linear bandit: reward poisoning attack, action poisoning attack and context poisoning attack. In the reward poisoning attack (Ma et al., 2018; Garcelon et al., 2020), after the agent observes context $x_t$ and chooses arm $I_t$, the environment will generate reward $r_{t,I_t}$ based on context $x_t$ and arm $I_t$. Then, the attacker can change the reward $r_{t,I_t}$ to $\widetilde{r}_t$ and feed $\widetilde{r}_t$ to the agent.

Compablack with the reward poisoning attacks, the action poisoning attack consideblack in this paper is more difficult to carry out. In particular, as the action poisoning attack only changes the action, it can impact but does not have direct control of the reward signal. By changing the action $I_t$ to $I_t^0$, the reward received by the agent is changed from $r_{t,I_t}$ to $r_{t,I_t^0}$ which is a random variable drawn from a distribution based on the action $I_t^0$ and context $x_t$. This is in contrast to reward poisoning attacks where an attacker has direct control and can change the reward signal to any value $\widetilde{r}_t$ of his choice.

In the context poisoning attack (Garcelon et al., 2020), the attacker changes the context shown to the agent. The reward is generated based on the true context $x_t$ and the agent's action $I_t$. Nevertheless, the agent's action may be indirectly impacted by the manipulation of the context, and so as the reward. Since the attacker attacks before the agent pulls an arm, the context poisoning attack is the most difficult to carry out.

For numerical comparison, as our paper is the first paper that discusses the action poisoning attack against contextual linear bandits, there is no existing action poisoning attack method to compare. One could run simulations to compare with reward or context poisoning attacks, but the attack costs are defined differently, due to the different nature of attacks. For example, for reward poisoning attacks, (Garcelon et al., 2020) proposed a reward poisoning attack method whose attack cost scales as $O(\log(T)^2)$. However, the definition of the attack cost in (Garcelon et al., 2020) is different from that of our paper. The cost of the reward attack is defined as the cumulative differences between the post-attack rewards and the pre-attack rewards. The

cost of the action attack is defined as the number of rounds that the attacker changes the actions selected by the agent. Although the definition of the attack cost of these two different kinds of attacks are different, the attack cost of our proposed white-box attack strategy scales on $O(\log(T)^2)$, which is same with the results in (Garcelon et al., 2020).

As mentioned in the introduction, the goal of this paper is not to promote any particular types of poisoning attacks. Instead, our goal is to understand the potential risks of action poisoning attacks, as the safe applications and design of robust contextual bandit algorithm relies on the addressing all possible weakness of the models.

## 4 Attack Schemes and Cost Analysis

In this section, we introduce the proposed action poisoning attack schemes in the white-box setting and black-box setting respectively. In order to demonstrate the significant security threat of action poisoning attacks to linear contextual bandits, we investigate our attack strategies against a widely used algorithm: LinUCB algorithm. Furthermore, we analyze the attack cost of our action poisoning attack schemes.

### 4.1 Overview of LinUCB

For reader's convenience, we first provide a brief overview of the LinUCB algorithm (Li et al., 2010). The LinUCB algorithm is summarized in Algorithm 2. The main steps of LinUCB are to obtain estimates of the unknown parameters $\theta_i$ using past observations and then make decisions based on these estimates. Define $\tau_i(t) := \{s : s \le t, I_s = i\}$ as the set of rounds up to $t$ where the agent pulls arm $i$. Let $N_i(t) = |\tau_i(t)|$. Then, at round $t$, the $\ell_2$-regularized least-squares estimate of $\theta_i$ with regularization parameter $\lambda > 0$ is obtained by (Li et al., 2010)

$$\hat{\theta}_{t,i} = V_{t,i}^{-1} \sum_{k \in \tau_i(t-1)} r_{t,i} x_k, \tag{4}$$

where

$$V_{t,i} = \sum_{k \in \tau_i(t-1)} x_k x_k^\top + \lambda \mathbf{I}$$

with $\mathbf{I}$ being identity matrix.

After $\hat{\theta}_i$'s are obtained, at each round, an upper confidence bound of the mean reward has to be calculated for each arm (step 4 of Algorithm 2). Then, the LinUCB algorithm picks the arm with the largest upper confidence bound (step 5 of Algorithm 2). By following the setup in "optimism in the face of uncertainty linear algorithm" (OFUL) (Abbasi-Yadkori et al., 2011), we set

$$\omega(N) = \sqrt{\lambda} S + R\sqrt{2 \log K/\delta + d \log\left(1 + L^2 N/(\lambda d)\right)},$$

and

$$\beta_{t,i} = \omega(N_i(t)) = \sqrt{\lambda} S + R\sqrt{2 \log K/\delta + d \log\left(1 + L^2 N_i(t)/(\lambda d)\right)}.$$

It is easy to verify that $\omega(N)$ is a monotonically increasing function over $N \in (0, +\infty)$.

### 4.2 White-box Attack

We first consider the white-box attack scenario, in which the attacker has knowledge of the environment. In particular, in the white-box attack scenario, the attacker knows the coefficient vectors $\theta_i$'s for all $i$. The understanding of this scenario could provide useful insights for the more practical black-box attack scenario to be discussed in Section 4.3.

Our proposed attack strategy works as follows. When the agent chooses arm $K$, the attacker does not attack. When the agent chooses arm $I_t \ne K$, the attacker changes it to arm

$$I_t^0 = \begin{cases} K & \text{with probability} \quad \epsilon_t \\ I_t^\dagger & \text{with probability} \quad 1 - \epsilon_t \end{cases} \tag{5}$$

---

**Algorithm 2** Contextual LinUCB (Li et al., 2010)

---

**Require:** $\lambda$, $K$, $T$, $L$, $S$.

1: Initialize for every arm $i$, $V_i \leftarrow \lambda \mathbf{I}$, $b_i \leftarrow \mathbf{0}$, $\hat{\theta}_i \leftarrow V_i^{-1} b_i$.

2: **for** $t = 1, 2, \ldots, T$ **do**

3:     Observe the context $x_t$.

4:     Compute the upper confidence bound: $p_{t,i} \leftarrow x_t^\top \hat{\theta}_i + \beta_{t,i} \sqrt{x_t^\top V_i^{-1} x_t}$ for all arm $i \in [K]$.

5:     Pull arm $I_t = \arg\max_i p_{t,i}$.

6:     The environment generates reward $r_t$ according to arm $I_t$. The agent receive $r_t$.

7:     $V_{I_t} \leftarrow V_{I_t} + x_t x_t^\top$, $b_{I_t} \leftarrow b_{I_t} + r_t x_t$, $\hat{\theta}_{I_t} \leftarrow V_{I_t}^{-1} b_{I_t}$.

8: **end for**

---

where $I_t^\dagger = \arg\min_i \langle x_t, \theta_i \rangle$ and

$$\epsilon_t = \frac{(1-\alpha)\langle x_t, \theta_K \rangle - \min_i \langle x_t, \theta_i \rangle}{\langle x_t, \theta_K \rangle - \min_i \langle x_t, \theta_i \rangle}. \tag{6}$$

We now explain the main idea behind the attack strategy specified in Equation 5 and Equation 6. Intuitively speaking, using Equation 5 and Equation 6, the attacker can manipulate the agent into learning some particular coefficient vectors. In particular, for arm $K$ (the target arm), the agent obtains the true coefficient vector $\theta_K$. For any arm $i \neq K$, because of the attacks, the agent will obtain an estimate of arm $i$'s coefficient vector that is close to $(1-\alpha)\theta_K$. This can be seen as follows. Under Assumption 1 and the fact that the mean rewards are positive, we have that $1/2 < \epsilon_t < 1-\alpha$. Under the white-box attack, when the agent pulls a non-target arm $I_t \neq K$, the post-attack action $I_t^0$ is a random variable and the mean reward received by the agent should satisfy

$$\mathbb{E}[r_{t,I_t^0}|F_{t-1}, I_t] = \mathbb{E}[\langle x_t, \theta_{I_t^0} \rangle|F_{t-1}, I_t] = \epsilon_t \langle x_t, \theta_K \rangle + (1-\epsilon_t)\min_i \langle x_t, \theta_i \rangle = (1-\alpha)\langle x_t, \theta_K \rangle. \tag{7}$$

Because of the positive mean reward assumption and the fact that $0 < 1-\alpha < 1$, the target arm $K$ appears to be the optimal arm for any context. Hence the proposed white-box attack strategy can easily mislead an agent who employs a contextual bandit algorithm with sublinear pseudo-regret, as shown in the following proposition.

**Proposition 1.** *For $\delta > 0$. Suppose the agent employs a linear contextual bandit algorithm whose cumulative pseudo-regret is upper bounded by $o(T)$ with probability at least $1 - \delta$. When using the proposed white-box attack scheme, with probability at least $1 - \delta$, the attacker can force the agent to pull the target arm for $T - o(T)$ times, while the total attack cost $|\mathcal{C}|$ is upper bounded by $o(T)$.*

Proposition 1 applies to any contextual bandit algorithm whose cumulative pseudo-regret is upper bounded by $o(T)$. To further illustrate the proposed attack scheme, we now provide a finer analysis the impact of this attack on LinUCB described in Algorithm 2. We first build a high probability bound for the estimate of parameter at the agent, when the agent computes the estimate of $\theta_i$ by equation 4 and the attacker employs the white-box attack. Recall that $\omega(N) = \sqrt{\lambda}S + R\sqrt{2\log K/\delta + d\log(1 + L^2 N/(\lambda d))}$.

**Lemma 1.** *Under the proposed white-box attack, the estimate of $\theta_i$ for each arm $i$ obtained by LinUCB agent as described in Algorithm 2 satisfies*

$$|x_t^\top \hat{\theta}_{t,i} - x_t^\top (1-\alpha)\theta_K| \leq \left(\omega(N_i(t)) + LS\sqrt{0.5\log(2KT/\delta)}\right)\|x_t\|_{V_{t,i}^{-1}}, \tag{8}$$

*with probability $1 - 2(K-1)\delta/K$, for all arm $i \neq K$ and all $t \geq 0$. Here, $\|x\|_V = \sqrt{x^\top V x}$ is the weighted norm of vector $x$ for a positive definite matrix $V$.*

Lemma 1 shows that, under our white-box attack, the agent's estimate of the parameter of non-target arm, i.e. $\hat{\theta}_i$, will converge to $(1-\alpha)\theta_K$. Thus, the agent is misled to believe that arm $K$ is the optimal arm for every context in most rounds. The following theorem provides an upper bound of the cumulative cost of the attack.

**Theorem 1.** *Define $\gamma = \min_{x \in \mathcal{D}} \langle x, \theta_K \rangle$. Under the same assumptions as in Lemma 1, for any $\delta > 0$ with probability at least $1 - 2\delta$, for all $T \geq 0$, the attacker can manipulate the LinUCB agent into pulling the target arm in at least $T - |\mathcal{C}|$ rounds, using an attack cost*

$$|\mathcal{C}| \leq \frac{2d(K-1)}{(\alpha\gamma)^2} \log\left(1 + TL^2/(d\lambda)\right) \left(2\omega(T) + LS\sqrt{0.5\log\left(2KT/\delta\right)}\right)^2. \tag{9}$$

Theorem 1 shows that our white-box attack strategy can force LinUCB agent into pulling the target arm $T - O(\log^2 T)$ times with attack cost scaled only as $O(\log^2 T)$.

*Proof.* Detailed proof of Theorem 1 can be found in Appendix B.3. Here we provide sketch of the main proof idea.

For round $t$ and context $x_t$, if LinUCB pulls arm $i \neq K$, we have

$$x_t^\top \hat{\theta}_{t,K} + \beta_{t,K}\sqrt{x_t^\top V_{t,K}^{-1} x_t} \leq x_t^\top \hat{\theta}_{t,i} + \beta_{t,i}\sqrt{x_t^\top V_{t,i}^{-1} x_t}.$$

Since the attacker does not attack the target arm, the confidence bound of arm $K$ does not change and $x_t^\top \theta_K \leq x_t^\top \hat{\theta}_{t,K} + \beta_{t,K}\sqrt{x_t^\top V_{t,K}^{-1} x_t}$ holds with probability $1 - \frac{\delta}{K}$.

Then, by Lemma 1,

$$x_t^\top \theta_K \leq x_t^\top (1-\alpha)\theta_K + \beta_{t,i}\|x_t\|_{V_{t,i}^{-1}} + \left(LS\sqrt{\frac{1}{2}\log\left(\frac{2KT}{\delta}\right)} + \omega\left(N_i(t)\right)\right)\|x_t\|_{V_{t,i}^{-1}}. \tag{10}$$

By multiplying both sides $\mathbb{1}_{\{I_t=i\}}$ and summing over rounds, we have

$$\begin{aligned}
&\sum_{k=1}^{T} \mathbb{1}_{\{I_k=i\}} \alpha x_k^\top \theta_K \\
&\leq \sum_{k=1}^{T} \mathbb{1}_{\{I_k=i\}} \left(\beta_{k,i} + \sqrt{\lambda}S + LS\sqrt{\frac{1}{2}\log\left(\frac{2KT}{\delta}\right)} + R\sqrt{2\log\frac{K}{\delta} + d\log\left(1 + \frac{L^2 N_i(k)}{\lambda d}\right)}\right)\|x_k\|_{V_{k,i}^{-1}}.
\end{aligned} \tag{11}$$

Here, we use Lemma 11 from (Abbasi-Yadkori et al., 2011) and obtain

$$\sum_{k=1}^{T} \mathbb{1}_{\{I_k=i\}}\|x_k\|_{V_{k,i}^{-1}} \leq \sqrt{N_i(t)2d\log\left(1 + \frac{tL^2}{d\lambda}\right)}. \tag{12}$$

Set $\gamma = \min_{x \in \mathcal{D}} \langle x, \theta_K \rangle$. Since $N_i(t) = \sum_{k=1}^{T} \mathbb{1}_{\{I_k=i\}}$, we have

$$N_i(t) \leq \frac{2d}{(\alpha\gamma)^2} \log\left(1 + \frac{tL^2}{d\lambda}\right) \left(2\sqrt{\lambda}S + LS\sqrt{\frac{1}{2}\log\left(\frac{2KT}{\delta}\right)} + 2R\sqrt{2\log\frac{K}{\delta} + d\log\left(1 + \frac{tL^2}{\lambda d}\right)}\right)^2. \tag{13}$$

$$\square$$

### 4.3 Black-box Attack

We now focus on the more practical black-box setting, in which the attacker does not know any of arm's coefficient vector. The attacker knows the value of $\alpha$ (or a lower bound) in which the Equation 3 holds for all $t$. Although the attacker does not know the coefficient vectors, the attacker can compute an estimate of the unknown parameters by using past observations. On the other hand, there are multiple challenges brought by the estimation errors that need to properly addressed.

The proposed black-box attack strategy works as follows. When the agent chooses arm $K$, the attacker does not attack. When the agent chooses arm $I_t \neq K$, the attacker changes it to arm

$$I_t^0 = \begin{cases} K & \text{with probability} \quad \epsilon_t \\ I_t^\dagger & \text{with probability} \quad 1 - \epsilon_t \end{cases} \tag{14}$$

where

$$I_t^\dagger = \arg \min_{i \neq K} \left( \langle x_t, \hat{\theta}_{t,i}^0 \rangle - \beta_{t,i}^0 \|x_t\|_{(V_{t,i}^0)^{-1}} \right), \tag{15}$$

and

$$\beta_{t,i}^0 = \phi_i \left( \omega(N_i^\dagger(t)) + LS\sqrt{0.5 \log(2KT/\delta)} \right),$$

$\phi_i = 1/\alpha$ when $i \neq K$ and $\phi_K = 2$, and

$$\epsilon_t = \text{clip} \left( \frac{1}{2}, \frac{(1-\alpha)\langle x_t, \hat{\theta}_{t,K}^0 \rangle - \langle x_t, \hat{\theta}_{t,I_t^\dagger}^0 \rangle}{\langle x_t, \hat{\theta}_{t,K}^0 \rangle - \langle x_t, \hat{\theta}_{t,I_t^\dagger}^0 \rangle}, 1 - \alpha \right), \tag{16}$$

with $\text{clip}(a, x, b) = \min(b, \max(x, a))$ where $a \leq b$.

For notational convenience, we set $I_t^\dagger = K$ and $\epsilon_t = 1$ when $I_t = K$. We define that, if $i \neq K$,

$$\tau_i^\dagger(t) := \{s : s \leq t, I_s^\dagger = i\}$$

and $N_i^\dagger(t) = |\tau_i^\dagger(t)|$. We also define $\tau_K^\dagger(t) := \{s : s \leq t\}$ and $N_K^\dagger(t) = |\tau_K^\dagger(t)|$.

$$\hat{\theta}_{t,i}^0 = \left( V_{t,i}^0 \right)^{-1} \sum_{k \in \tau_i^\dagger(t-1)} w_{k,i} r_{k,I_k^0} x_k, \tag{17}$$

where

$$V_{t,i}^0 = \sum_{k \in \tau_i^\dagger(t-1)} x_k x_k^\top + \lambda \mathbf{I}$$

and

$$w_{t,i} = \begin{cases} 1/\epsilon_t & \text{if } i = I_t^0 = K \\ 1/(1-\epsilon_t) & \text{if } i = I_t^0 = I_t^\dagger \\ 0 & \text{if } i \neq I_t^0 \end{cases}. \tag{18}$$

Here, $\hat{\theta}_{t,i}^0$ is the estimation of $\theta_i$ by the attacker, while $\hat{\theta}_{t,i}$ in Equation 4 is the estimation of $\theta_i$ at the agent side. We will show in Lemma 2 and Lemma 4 that $\hat{\theta}_{t,i}^0$ will be close to the true value of $\theta_i$ while $\hat{\theta}_{t,i}$ will be close to a sub-optimal value chosen by the attacker. This disparity gives the attacker the advantage for carrying out the attack.

We now highlight the main idea of why our black-box attack strategy works. As discussed in Section 4.2, if the attacker knows the coefficient vectors of all arms, the proposed white-box attack scheme can mislead the agent to believe that the coefficient vector of every non-target arm is $(1 - \alpha)\theta_K$, hence the agent will think the target arm is optimal. In the black-box setting, the attacker does not know the coefficient vector for any arm. The attacker should estimate an coefficient vector of each arm. Then, the attacker will use the estimated coefficient vector to replace the true coefficient vector in the white-box attack scheme. As the attacker does not know the true values of $\theta_i$'s, we need to design the estimator $\hat{\theta}_{t,i}^0$, the attack choice $I_t^\dagger$ and the probability $\epsilon_t$ carefully. In the following, we explain the main ideas behind our design choices. 1): Firstly, we explain why we design estimator $\hat{\theta}_{t,i}^0$ using the form Equation 17, in which the attacker employs the importance sampling to obtain an estimate of $\theta_i$. There are two reasons for this. The first reason is that, for a successful attack, the number of observations in arm $i \neq K$ will be limited. Hence if the importance sampling is not used, the estimation variance of the mean reward $\langle x, \theta_i \rangle$ at the attacker side for some contexts $x$ may be large. The second reason is that the attacker's action is stochastic when the agent

pulls a non-target arm. Thus, the attacker uses the observations at round $t$ when the attacker pulls arm $i$ with certain probability, i.e. when $t \in \tau_i^{\dagger}$, to estimate $\theta_i$. Since the agent's action is deterministic, the agent uses the observations at round $t$ when the agent pulls arm $i$, i.e. when $t \in \tau_i$, to estimate $\theta_i$. 2): Secondly, we explain ideas behind the choice of $I_t^{\dagger}$ in Equation 15. Under our black-box attack, when the agent pulls a non-target arm $I_t \neq K$, the mean reward received by the agent satisfies

$$\mathbb{E}[r_{t,I_t^0}|F_{t-1}, I_t] = \mathbb{E}[\langle x_t, \theta_{I_t^0}\rangle|F_{t-1}, I_t] = \epsilon_t\langle x_t, \theta_K\rangle + (1-\epsilon_t)\langle x_t, \theta_{I_t^{\dagger}}\rangle. \tag{19}$$

In white-box attack scheme, $I_t^{\dagger}$ is the worst arm at context $x_t$. In the black-box setting, the attacker does not know a priori which arm is the worst. In the proposed black-box attack scheme, as indicated in Equation 15, we use the lower confidence bound (LCB) method to explore the worst arm and arm $I_t^{\dagger}$ has the smallest lower confidence bound. 3): Finally, we provide reasons why we choose $\epsilon_t$ using Equation 16. In our white-box attack scheme, we have that $1/2 < \epsilon_t < 1 - \alpha$. Thus, in our black-box attack scheme, we limit the choice of $\epsilon_t$ to $[1/2, 1 - \alpha]$. Furthermore, in Equation 6 used for the white-box attack, $\epsilon_t$ is computed by the true mean reward. Now, in the black-box attack, as the attacker does not know the true coefficient vector, the attacker uses an estimation of $\theta$ to compute the second term in the clip function in Equation 16.

In summary, our design of $\hat{\theta}_{t,i}^0$, $I_t^{\dagger}$ and $\epsilon_t$ can ensure that the attacker's estimation $\hat{\theta}_{t,i}^0$ is close to $\theta_i$, while the agent's estimation $\hat{\theta}_{t,i}$ will be close to $(1-\alpha)\theta_K$. In the following, we make these statements precise, and formally analyze the performance of the proposed black-box attack scheme. First, we analyze the estimation $\hat{\theta}_{t,i}^0$ at the attacker side. We establish a confidence ellipsoid of $\langle x_t, \hat{\theta}_{t,i}^0\rangle$ at the attacker.

**Lemma 2.** *Assume the attacker performs the proposed black-box action poisoning attack. With probability $1 - 2\delta$, we have*

$$|x_t^{\top}\hat{\theta}_{t,i}^0 - x_t^{\top}\theta_i| \leq \beta_{t,i}^0 \|x_t\|_{(V_{t,i}^0)^{-1}}$$

*holds for all arm $i$ and all $t \geq 0$.*

Lemma 2 shows that $\hat{\theta}_i^0$ lies in an ellipsoid with center at $\theta_i$ with high probability, which implies that the attacker has good estimate. We then analyze the estimation $\hat{\theta}_{t,i}$ at the agent side. The following lemma provides an upper bound on the difference between $\mathbb{E}[r_{t,I_t^0}|F_{t-1}, I_t]$ and $(1-\alpha)\langle x_t, \theta_K\rangle$.

**Lemma 3.** *Under the black-box attack, the estimate obtained by an LinUCB agent satisfies*

$$\left|\mathbb{E}[r_{t,I_t^0}|F_{t-1}, I_t] - (1-\alpha)\langle x_t, \theta_K\rangle\right| \leq (1-\alpha)\beta_{t,K}^0\|x_t\|_{(V_{t,K}^0)^{-1}} + (1+\alpha)\beta_{t,I_t^{\dagger}}^0\|x_t\|_{\left(V_{t,I_t^{\dagger}}^0\right)^{-1}}. \tag{20}$$

*simultaneously for all $t \geq 0$ when $I_t \neq K$, with probability $1 - 2\delta$.*

The bound in Lemma 3 consists of the confidence ellipsoid of the estimate of arm $I_t^{\dagger}$ and that of arm $K$. As mentioned above, for a successful attack, the number of pulls on arm $I_t^{\dagger}$ will be limited. Thus, in our proposed algorithm, the attacker use the importance sampling to obtain the estimate of $\theta_i$, which will increases the number of observations that can be used to estimate the coefficient vector of arm $I_t^{\dagger}$. Using Lemma 3, we have the following lemma regarding the estimation $\hat{\theta}_{t,i}$ at the agent side.

**Lemma 4.** *Consider the same assumption as in Lemma 2. With a probability at least $1 - \frac{(3K-1)\delta}{K}$, the estimate $\hat{\theta}_{t,i}$ obtained by the LinUCB agent will satisfy*

$$|x_t^{\top}\hat{\theta}_{t,i} - x_t^{\top}(1-\alpha)\theta_K| \leq \left(1 + 4d/\alpha\sqrt{K\log\left(1 + tL^2/(d\lambda)\right)}\right)\left(\omega(t) + LS\sqrt{0.5\log\left(2KT/\delta\right)}\right)\|x_t\|_{V_{t,i}^{-1}}, \tag{21}$$

*simultaneously for all arm $i \neq K$ and all $t \geq 0$.*

Lemma 4 shows that, under the proposed black-box attack scheme, the agent's estimate of the parameter of the non-target arm, i.e. $\hat{\theta}_i$, will converge to $(1-\alpha)\theta_K$. Hence the agent will believe that the target arm $K$ is the optimal arm for any context in most rounds. Using these supporting lemmas, we can then analyze the performance of the proposed black-box attack strategy.

**Theorem 2.** *Define $\gamma = \min_{x \in \mathcal{D}} \langle x, \theta_K \rangle$. Under the same assumptions as in Lemma 4, with probability at least $1 - 3\delta$, for all $T \geq 0$, the attacker can manipulate a LinUCB agent into pulling the target arm in at least $T - |\mathcal{C}|$ rounds, using an attack cost*

$$|\mathcal{C}| \leq \frac{2d(K-1)}{(\alpha\gamma)^2} \left( 2 + \frac{4d}{\alpha} \sqrt{K \log\left(1 + \frac{TL^2}{d\lambda}\right)} \right)^2 \log\left(1 + TL^2/(d\lambda)\right) \left( \omega(T) + LS\sqrt{0.5 \log(2KT/\delta)} \right)^2. \tag{22}$$

*Proof.* Detailed proof of Theorem 2 can be found in Appendix C.4. Here we provide a sketch of the main proof idea.

For round $t$ and context $x_t$, if LinUCB pulls arm $i \neq K$, we have

$$x_t^\top \hat{\theta}_{t,K} + \beta_{t,K}\sqrt{x_t^\top V_{t,K}^{-1} x_t} \leq x_t^\top \hat{\theta}_{t,i} + \beta_{t,i}\sqrt{x_t^\top V_{t,i}^{-1} x_t}.$$

Since the attacker does not attack the target arm, the confidence bound of arm $K$ does not change and $x_t^\top \theta_K \leq x_t^\top \hat{\theta}_{t,K} + \beta_{t,K}\sqrt{x_t^\top V_{t,K}^{-1} x_t}$ holds with probability $1 - \frac{\delta}{K}$.

Thus, by Lemma 4,

$$\begin{aligned}
x_t^\top \theta_K \leq &\, x_t^\top (1-\alpha)\theta_K + \omega(N_i(t))\|x_t\|_{V_{t,i}^{-1}} \\
&+ \left( 1 + \frac{4d}{\alpha}\sqrt{K \log\left(1 + \frac{tL^2}{d\lambda}\right)} \right) \left( \omega(t) + LS\sqrt{\frac{1}{2}\log\left(\frac{2KT}{\delta}\right)} \right) \|x_t\|_{V_{t,i}^{-1}}.
\end{aligned} \tag{23}$$

By multiplying both sides by $\mathbb{1}_{\{I_t = i\}}$ and summing over rounds, we have

$$\begin{aligned}
&\sum_{k=1}^{T} \mathbb{1}_{\{I_k = i\}} \alpha x_k^\top \theta_K \\
&\leq \sum_{k=1}^{T} \mathbb{1}_{\{I_k = i\}} \left( 2 + \frac{4d}{\alpha}\sqrt{K \log\left(1 + \frac{kL^2}{d\lambda}\right)} \right) \left( \omega(t) + LS\sqrt{\frac{1}{2}\log\left(\frac{2KT}{\delta}\right)} \right) \|x_k\|_{V_{k,i}^{-1}}.
\end{aligned} \tag{24}$$

Here, we use Lemma 11 from (Abbasi-Yadkori et al., 2011) and get

$$\sum_{k=1}^{T} \mathbb{1}_{\{I_k = i\}} \|x_k\|_{V_{k,i}^{-1}}^2 \leq \sqrt{N_i(t) 2d \log\left(1 + \frac{tL^2}{d\lambda}\right)}. \tag{25}$$

Thus, we have

$$\begin{aligned}
N_i(t) &= \sum_{k=1}^{T} \mathbb{1}_{\{I_k = i\}} \\
&\leq \frac{2d}{(\alpha\gamma)^2} \log\left(1 + \frac{tL^2}{d\lambda}\right) \left( 2 + \frac{4d}{\alpha}\sqrt{K \log\left(1 + \frac{tL^2}{d\lambda}\right)} \right)^2 \left( \omega(t) + LS\sqrt{\frac{1}{2}\log\left(\frac{2KT}{\delta}\right)} \right)^2,
\end{aligned} \tag{26}$$

where $\gamma = \min_{x \in \mathcal{D}} \langle x, \theta_K \rangle$. $\qquad\square$

Theorem 2 shows that our black-box attack strategy can manipulate a LinUCB agent into pulling a target arm $T - O(\log^3 T)$ times with attack cost scaling as $O(\log^3 T)$. Compablack with the result for the white-box attack, the black-box attack only brings an additional $\log T$ factor.

### 4.4 Generalized Linear Models

We note that the proposed attack strategies can also be extended to generalized linear models. Detailed analysis of the cost of the proposed attack strategies for the generalized linear contextual bandit model can be found in Appendix D.

## 5 Numerical Experiments

In this section, we provide numerical examples to illustrate the impact of proposed action poisoning attack schemes.

### 5.1 Attack Linear Contextual Bandit Algorithms

We first empirically evaluate the performance of the proposed action poisoning attack schemes on three contextual bandit algorithms: LinUCB (Abbasi-Yadkori et al., 2011), LinTS (Agrawal & Goyal, 2013), and $\epsilon$-Greedy. We run the experiments on three datasets:

**Synthetic data:** The dimension of contexts and the coefficient vectors is $d = 6$. We set the first entry of every context and coefficient vector to 1. The other entries of every context and coefficient vector are uniformly drawn from $(-\frac{1}{\sqrt{d-1}}, \frac{1}{\sqrt{d-1}})$. Thus, $\|x\|_2 \leq \sqrt{2}$, $\|\theta\|_2 \leq \sqrt{2}$ and mean rewards $\langle x, \theta \rangle > 0$. The reward noise $\eta_t$ is drawn from a Gaussian distribution $\mathcal{N}(0, 0.01)$.

**Jester dataset (Goldberg et al., 2001):** Jester contains 4.1 million ratings of jokes in which the rating values scale from $-10.00$ to $+10.00$. We normalize the rating to $[0, 1]$. The dataset includes 100 jokes and the ratings were collected from 73,421 users between April 1999 - May 2003. We consider a subset of 10 jokes and 38432 users. Every jokes are rated by each user. We perform a low-rank matrix factorization ($d = 6$) on the ratings data and obtain the features for both users and jokes. At each round, the environment randomly select a user as the context and the reward noise is drawn from a Gaussian distribution $\mathcal{N}(0, 0.01)$.

**MovieLens 25M dataset: (Harper & Konstan, 2015)** MovieLens 25M dataset contains 25 million 5-star ratings of 62,000 movies by 162,000 users. The preprocessing of this data is almost the same as the Jester dataset, except that we consider a subset of 10 movies and 7344 users. At each round, the environment randomly select a user as the context and the reward noise is drawn from $\mathcal{N}(0, 0.01)$.

We set $\delta = 0.1$ and $\lambda = 2$. For all the experiments, we set the total number of rounds $T = 10^6$ and the number of arms $K = 10$. We independently run ten repeated experiments. Results reported are averaged over the ten experiments. We set $\alpha$ to 0.2 for the two proposed attack strategies, hence the target arm may be the worst arm in some rounds. Each of the individual experimental runs costs up to 10 minutes on one physical CPU core. The type of CPU is Intel Core i7-8700.

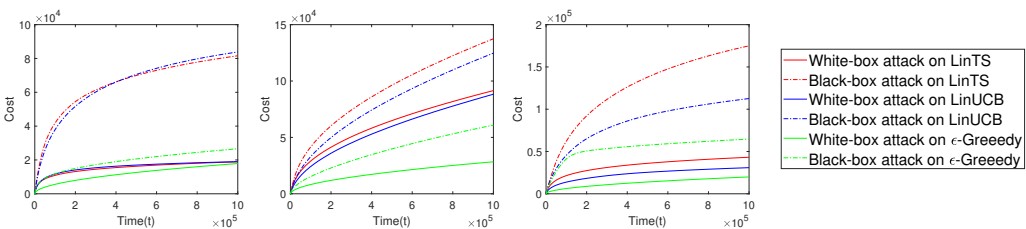

Figure 2: The cumulative cost of the attacks for the synthetic (Left), Jester (Center) and MovieLens (Right) datasets.

The results are shown in Table 1 and Figure 2. These experiments show that the action poisoning attacks can force the three agents to pull the target arm very frequently, while the agents rarely pull the target arm under no attack. Under the attacks, the true regret of the agent becomes linear as the target arm is not optimal for most context. Table 1 show the number of rounds the agent pulls the target arm among $10^6$ total rounds. In the synthetic dataset, under the proposed white-box attacks, the target arm is pulled more

|  | Synthetic | Jester | MovieLens |
|---|---|---|---|
| $\epsilon$-Greeedy without attacks | 2124.6 | 5908.7 | 3273.5 |
| White-box attack on $\epsilon$-Greeedy | 982122.5 | 971650.9 | 980065.6 |
| Black-box attack on $\epsilon$-Greeedy | 973378.5 | 939090.2 | 935293.8 |
| LinUCB without attacks | 8680.9 | 16927.2 | 13303.4 |
| White-box attack on LinUCB | 981018.7 | 911676.9 | 969118.6 |
| Black-box attack on LinUCB | 916140.8 | 875284.7 | 887373.1 |
| LinTS without attacks | 5046.9 | 18038.0 | 9759.0 |
| White-box attack on LinTS | 981112.8 | 908488.3 | 956821.1 |
| Black-box attack on LinTS | 918403.8 | 862556.8 | 825034.8 |

Table 1: Average number of rounds when the agent pulls the target arm over $T = 10^6$ rounds.

than 98.1% of the times by the three agent (see Table 1). The target arm is pulled more than 91.6% of the times in the worst case (the black-box attacks on LinUCB). Fig 2 shows the cumulative cost of the attacks on three agents for the three datasets. The results show that the attack cost $|\mathcal{C}|$ of every attack scheme on every agent for every dataset scales sublinearly, which exposes a significant security threat of the action poisoning attacks on linear contextual bandits.

## 5.2 Attack Robust Algorithms

We now discuss existing robust linear bandit algorithms (Ding et al., 2022; Bogunovic et al., 2021) and evaluate the performance of the proposed attack strategy on these algorithms.

In particular, (Bogunovic et al., 2021) focuses on a special case in which the context and coefficient vectors are assumed to be fixed over rounds, and developed Robust Phased Elimination (RPE) algorithm. In contrast, our paper focuses on the general setting where the contexts are different for each round and coefficients are different for each arm. Hence the RPE algorithm will not work for the contextual bandit setting consideblack in our paper. (Bogunovic et al., 2021) also proves that a simple greedy algorithm based on linear regression can be robust to linear contextual bandits with shablack coefficient under a stringent diversity assumption on the contexts. We empirically evaluate the action attacks on the greedy algorithm in (Bogunovic et al., 2021). The greedy algorithm is designed in the shablack coefficient setting and may not work in the disjoint setting.

(Ding et al., 2022) provides a linear contextual bandit algorithm that is robust to rewards attacks and context attacks. The scheme in (Ding et al., 2022) could be used to defend against the action attacks. However, our numerical results show that the proposed attack strategy can successfully defeat the scheme in (Ding et al., 2022). In the following, we empirically evaluate the performance of the proposed action poisoning attack schemes on RobustBandit algorithm in (Ding et al., 2022).

(He et al., 2022) provides nearly optimal algorithms for linear contextual bandits with adversarial corruptions. (He et al., 2022) consider two cases: 1) the agent knows the corruption budget; and 2) the agent does not know the corruption budget. We also empirically evaluate the action attacks on CW-OFUL algorithm in (He et al., 2022).

We use the same synthetic data setting in Section 5.1. We set $\delta = 0.1$ and $\lambda = 2$. For all the experiments, we set the total number of rounds $T = 10^6$ and the number of arms $K = 10$. We independently run ten repeated experiments. Results reported are averaged over the ten experiments. We set $\alpha$ to 0.2 for the two proposed attack strategies. For the action attacks on CW-OFUL algorithm, we run the simulation on the two cases that the agent know the corruption budget and the agent does not know the corruption budget. For the case that the agent know the corruption budget, we set the attack budget to 3000 and the attacker will stop the attack if the budget is exhaust. For the case that the agent does not know the corruption budget, we does not limit the attack budget.

The simulation results in Figure 3 show that our proposed white-box action poisoning attack can force the greedy contextual agent in (Bogunovic et al., 2021) to pull the target arm. Although the black-box

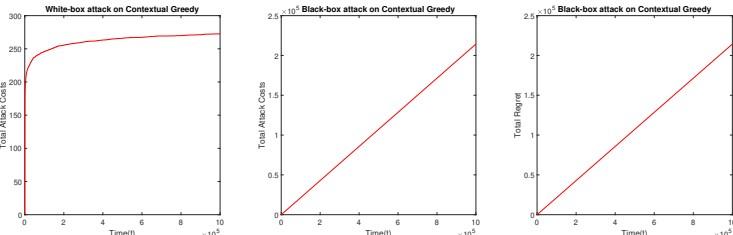

Figure 3: The cumulative cost of the attacks against greedy contextual algorithm in (Bogunovic et al., 2021).

action poisoning attack strategy fails, it causes linear regret on the agent. The greedy contextual algorithm in (Bogunovic et al., 2021) highly relies a stringent diversity assumption on the contexts. In the disjoint setting, the agent will pull each arm in some correlated contexts. The stringent diversity assumption may not be satisfied in disjoint setting. The greedy contextual algorithm in (Bogunovic et al., 2021) will induced to linear regret as shown in Figure 3.

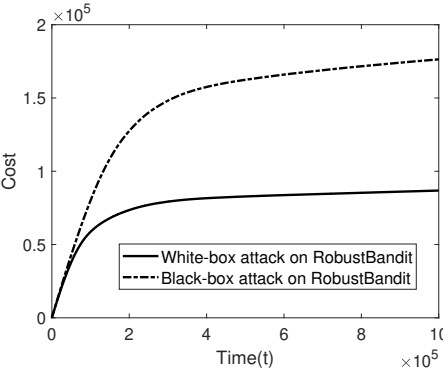

Figure 4: The cumulative cost of the attacks against RobustBandit algorithm for the synthetic datasets.

The simulation results in Figure 4 show that our proposed action poisoning attack can force the RobustBandit agent to pull the target arm with $T - o(T)$ times. The attack cost also scales sublinearly on $T$. The reason why the RobustBandit agent cannot defend against the action attacks is that its regret scales on $O(C\sqrt{T})$. If the attack cost $C = O(\sqrt{T})$, the regret will be linear and the agent will be fooled.

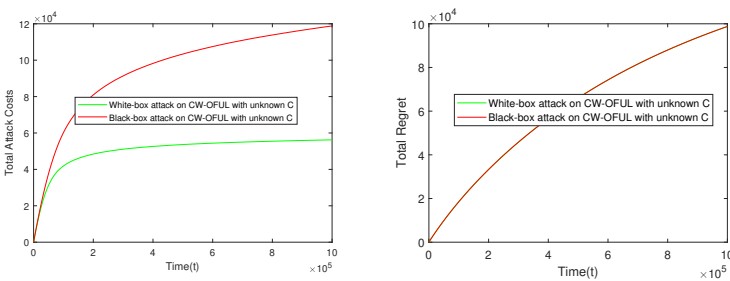

Figure 5: The cumulative cost of the attacks against CW-OFUL algorithm for the synthetic datasets.

The simulation results in Figure 5 show that our proposed action poisoning attack can force the CW-OFUL agent to pull the target arm in the case that the agent does not know the corruption budget and the attack budget is unlimited. The attack cost also scales sublinearly on $T$. The reason why the CW-OFUL agent cannot defend against the action attacks is that its regret scales on $O(T)$ once the corruption level is larger

than $\sqrt{T}$. CW-OFUL agent can defend the action poisoning attack and achieve sublinear regret when the agent knows the corruption budget and the attack budget is limited.

## 5.3 The Attack Performance When the Assumption 1 Violates

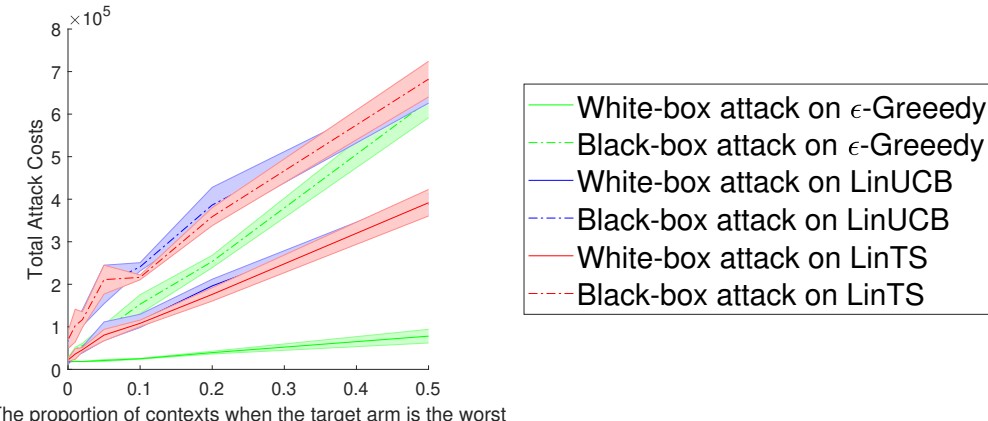

Figure 6: The total costs of the attacks for the synthetic datasets when the Assumption 1 violates.

We now evaluate the sensitivity of the proposed algorithms on Assumption 1. We empirically evaluate the performance of the proposed action poisoning attack schemes on three contextual bandit algorithms: LinUCB, LinTS, and $\epsilon$-greedy, when the target arm is the worst for some contexts.

We use the same synthetic data setting in Section 5.1. We set $\delta = 0.1$ and $\lambda = 2$. For all the experiments, we set the total number of rounds $T = 10^6$ and the number of arms $K = 10$. We independently run ten repeated experiments. Results reported are averaged over the ten experiments. We set $\alpha$ to 0.2 for the two proposed attack strategies. We manipulate the simulation environments so that the proportions of contexts, at which the target arm is the worst, are separately controlled around: 0.001, 0.01, 0.02, 0.05, 0.1, 0.2, 0.5.

The results in Figure 6 show the total costs under different proportions of contexts when the target arm is the worst. The $x$-axis represents the proportions of context when the target arm is the worst. The $y$-axis represents the total cost over $10^6$ rounds. The results shows that the total attack costs scale linearly on the proportions of contexts when the target arm is the worst. When the proportions of contexts when the target arm is the worst is small, our proposed attack strategies can still efficiently attacks.

## 5.4 The Attack Performance When the Rewards Are Normalized in [-1,1]

Here, we would like to note that we make the positive reward assumption for the formal analysis, our results actually can be generalized to the case with negative rewards. If the positive reward assumption does not hold, for the case with negative rewards, the attacker can preprocess the reward by adding a positive constant to all rewards such that all rewards become positive. As long as Assumption 1 holds, the proposed attack strategy can mislead the agent to believe that the target arm is optimal regardless of the positive reward assumption.

To illustrate this, we evaluate the performance of the proposed algorithms when negative rewards exist. We use the same synthetic data setting in Section 5.1. We set $\delta = 0.1$ and $\lambda = 2$. For all the experiments, we set the total number of rounds $T = 10^6$ and the number of arms $K = 10$. We independently run ten repeated experiments. Results reported are averaged over the ten experiments. We set $\alpha$ to 0.2 for the two proposed attack strategies. We normalize the rewards in [-1,1]. The attacker preprocesses the rewards by adding a constant $c = 3$ to the rewards in his algorithms. In the white-box attack setting, the attacker adds $c = 3$ to every $\langle x, \theta \rangle$ and use the preprocessed $\langle x, \theta \rangle$ to compute $\epsilon_t$. In the black-box attack setting, the attacker adds $c = 3$ to every reward $r_t$ and use the preprocessed $r_t$ to compute $\hat{\theta}$ and $\epsilon_t$.

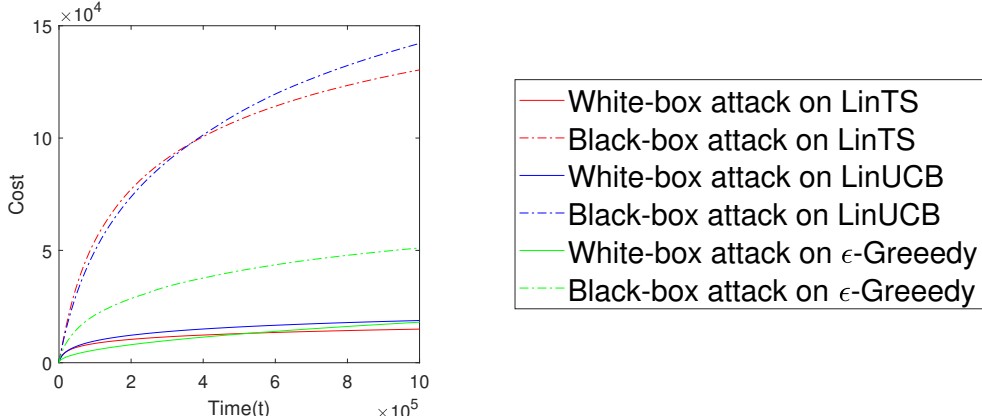

Figure 7: The cumulative cost of the attacks when the rewards are normalized in [-1,1].

The results in Figure 7 show that our proposed attack strategies still work for the case with negative rewards.

## 6 Future Works

In this section, we discuss several future directions.

**Linear Bandits with Shablack Coefficients.** In this paper, we discuss the disjoint linear contextual model, similar to those consideblack in (Li et al., 2010; Kong et al., 2020; Garcelon et al., 2020; Huang et al., 2021), where each arm is associated with its own coefficient vector. At each time, the agent observes a contextual vector $x_t$ that is shablack with all arms.

Our model is different from the linear contextual bandit model with shablack coefficient. In that model, the coefficient vector $\theta^*$ is shablack with all arms. At each time, each arm $i$ is given a specific contextual vector $x_{i,t}$. These contextual vectors form the decision set. The decision set changes over time and can even be infinite. In this model, adversarial attacks may not always be successful. For example, (Wang et al., 2022) shows that some attack goals can never be achieved in linear stochastic bandits due to the shablack coefficient. This may also occur in the linear contextual bandit model with shablack coefficient. It is of interest to conduct rigorous analysis of action attacks in the shablack coefficient model in the future work.

**Attack Strategy without Assumption 1.** In Section 5.3, we empirically evaluate the sensitive of the proposed algorithms on Assumption 1. The results shows that the total attack costs seem to scale linearly on the proportions of contexts when the target arm is the worst. It is of interest to further theoretically analyze the phenomenon shown in the simulation results in Section 5.3. However, using current tools analysis developed in the paper, it is challenging to formally analyze the relationship between the effectiveness of the action attack strategies and the portion of contexts when the target arm is the worst. On the other hand, we can try to find a new algorithm that can force the agent to choose the target arm when the target arm is not the worst. In the following, we provide our initial ideas for this direction.

In the action attack strategies consideblack in the paper, the attacker misleads the agent to obtain an estimate of the non-target arms' coefficient vectors that are close to $(1-\alpha)\theta_K$. However, this can not be always achieved without Assumption 1, since the target arm is the worst at some contexts but $(1-\alpha)\theta_K$ is worse than the target arm. Action attacks can not force the agent to learn $(1-\alpha)\theta_K$. We need some new attack strategies. Here, we provide our initial thoughts on theoretical insights to solve this challenge in the white-box attack scenario.

If the Assumption 1 does not hold, we set

$$\mathcal{D}_K := \{x \in \mathcal{D} : \langle x, \theta_K \rangle = \min_{i \in [K]} \langle x, \theta_i \rangle\}$$

as the set of context where the target arm is the worst. We consider such sequence of contexts $\{x_t\}_{t \in [T]}$ that $x_t \in \mathcal{D}_K$ for all $t < T_K$ and $T_K$ linearly depends on $T$.

We consider a $r$-neighborhood of $\mathcal{D}_K$:

$$\widetilde{\mathcal{D}}_K := \cup_{x \in \mathcal{D}_K} B_r(x),$$

where $B_r(x) = \{x' \in \mathcal{D} : \|x - x'\|_2 < r\}$. In the white-box attack case, we can find a coefficient vector $\theta^\dagger$ such that $x^\top \theta_K > x^\top \theta^\dagger$ for all context $x \in \mathcal{D}_K$ and $x^\top \theta_K > x^\top \theta^\dagger$ for all context $x \in \mathcal{D}/\widetilde{\mathcal{D}}_K$. Then the attacker force the agent to obtain an estimate of the non-target arms' coefficient vectors that are close to $\theta^\dagger$. Then the target arm is the optimal in $\mathcal{D}/\widetilde{\mathcal{D}}_K$, and the distribution of $\widetilde{\mathcal{D}}_K$ is limited.

The proposed attack strategies in this paper may make the target arm to be the optimal in $\mathcal{D}/\widetilde{\mathcal{D}}_K$ with some $r$. Thus, even when the target arm is the worst arm, the attack algorithm designed in this paper can still work.

It is of interest to conduct rigorous analysis of this strategy in the future work. Furthermore, it is of interest to carry out the design and analysis for algorithms for the black box attack strategy under this scenario, which will be more challenging.

## 7 Conclusion

In this paper, we have proposed a class of action poisoning attacks on linear contextual bandits. We have shown that our white-box attack strategy is able to force any linear contextual bandit agent, whose regret scales sublinearly with the total number of rounds, into pulling a target arm chosen by the attacker. We have also shown that our white-box attack strategy can force LinUCB agent into pulling a target arm $T - O(\log^2 T)$ times with attack cost scaled as $O(\log^2 T)$. We have further shown that the proposed black-box attack strategy can force LinUCB agent into pulling a target arm $T - O(\log^3 T)$ times with attack cost scaled as $O(\log^3 T)$. Our results expose a significant security threat to contextual bandit algorithms. In the future, we will investigate the defense strategy to mitigate the effects of this attack.

## 8 Acknowledgement

This work was supported in part by the National Science Foundation under Grants ECCS-1824553, CCF-1908258 and ECCS-2000415.

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

# A    Necessity of Assumption 1

In the main paper, we show that, if Assumption 1 holds, our proposed attack schemes can successfully attack LinUCB regardless the context.

Here, we highlight the necessity of Assumption 1 in the sense that, if Assumption 1 does not hold, for some sequence of context $\{x_t\}_{t\in[T]}$, there is no efficient action poisoning attack scheme that can successfully attack LinUCB. If the Assumption 1 does not hold, we set

$$\mathcal{D}_K := \{x \in \mathcal{D} | \langle x, \theta_K \rangle = \min_{i\in[K]} \langle x, \theta_i \rangle \}$$

as the set of context where the target arm is the worst. We consider such sequence of contexts $\{x_t\}_{t\in[T]}$ that $x_t \in \mathcal{D}_K$ for all $t < T_K$ and $T_K$ linearly depends on $T$.

If the attacker does not attack the target arm, the target arm is the worst in $\mathcal{D}_K$. No matter how we change the non-target arms to another arm, the cost and loss will linearly depend on $T_K$ and therefore linearly depend on $T$.

If the attacker attacks the target arm when $x_t \in \mathcal{D}_K$ and changes the target arm to a better arm or the optimal arm so that the agent learns that the target set is optimal even in the context set $\mathcal{D}_K$, the cost of attack may be up to $O(T_K)$ and hence scales as $O(T)$. We define

$$\tau_{K,[K-1]}(t) = \{s : s \le t, I_s = K \text{ but } I_s^0 \ne K \}$$

as the set of rounds up to $t$ where the attacker attacks arm $K$ and $C_K(t) = |\tau_{K,[K-1]}(t)|$ as the attack cost of attacking arm $K$. We can show that if we attack the target arm, for any attack scheme,

$$
\begin{aligned}
\|\hat{\theta}_{t,K} - \theta_K\|_{V_{t,K}} &\le \|V_{t,K}^{-1} \sum_{s\in\tau_{K,[K-1]}(t)} LSx_s\|_{V_{t,K}} \\
&\le \sum_{s\in\tau_{K,[K-1]}(t)} LS\sqrt{x_s^\top V_{t,K}^{-1} x_s} \\
&\le LS\sqrt{C_K(t) \sum_{s\in\tau_{K,[K-1]}(t)} \|x_s\|_{V_{t,K}^{-1}}^2} \\
&\le LS\sqrt{2dC_K(t) \log\left(1 + \frac{N_K(t)L^2}{d\lambda}\right)},
\end{aligned}
\tag{27}
$$

where the last inequality is obtained by the Lemma 11 from (Abbasi-Yadkori et al., 2011). Then, the average estimation error of the mean reward of arm $K$ is bounded by

$$
\begin{aligned}
1/T_K \sum_{t=1}^{T_K} |x_t^\top (\hat{\theta}_{t,K} - \theta_K)| \\
\le 1/T_K \sum_{t=1}^{T_K} \|\hat{\theta}_{t,K} - \theta_K\|_{V_{t,K}} \|x_t\|_{V_{t,K}^{-1}} \\
\le 2dLS \log\left(1 + \frac{N_K(T_K)L^2}{d\lambda}\right) \sqrt{C_K(T_K)/T_K}.
\end{aligned}
\tag{28}
$$

Note that we consider a sequence of contexts $\{x_t\}_{t\in[T]}$ that $x_t \in \mathcal{D}_K$ for all $t < T_K$ and $\tau$ linearly depends on $T$. In the action poisoning attacks, if the attacker tries to blackuce the rewards of non-target arms, he can change the non-target arm to a worse arm or the worst arm. However, the target arm $K$ is the worst arm for all $t < T_K$. The mean rewards of the non-target arm at the contexts $\{x_t\}_{t\in[T_K]}$ must be larger or equal to $x_t^\top \theta_K$. If the attacker tries to force the agent to pull the target arm in linear time at the first $T_K$ rounds, the attacker needs to let the post-attack arm $K$ be $\Delta$-optimal than the original arm $K$. If the attacker tries

to let the post-attack arm $K$ be $\Delta$-optimal than the original arm $K$, we have $C_K(T_K) = \widetilde{O}(T_K)$, where $\widetilde{O}$ ignores the logarithm dependency.

In summary, without the prior information of the context distribution or the context sequence, there is no action poisoning attack scheme that can always efficiently attack without Assumption 1. However, in some situations, some action poisoning attack scheme may still work even if the target arm is the worst in a small portion of the contexts, as shown in the numerical result session.

# B   Attack Cost Analysis of White-box Setting

## B.1   Proof of Proposition 1

When the agent pulls a non-target arm $I_t \neq K$, the mean reward received by the agent should satisfy

$$\mathbb{E}[r_{t,I_t^0}|F_{t-1}, I_t] = (1-\alpha)\langle x_t, \theta_K \rangle.$$

In the observation of the agent, the target arm becomes optimal and the non-target arms are associated with the coefficient vector $(1-\alpha)\theta_K$. In addition, the cumulative pseudo-regret should satisfy

$$\bar{R}_T = \sum_{t=1}^{T} \mathbb{1}_{\{I_t \neq K\}} \alpha \langle x_t, \theta_K \rangle \geq \sum_{t=1}^{T} \mathbb{1}_{\{I_t \neq K\}} \alpha \gamma.$$

We define $\gamma = \min_{x \in \mathcal{D}} \langle x, \theta_K \rangle$. If $\bar{R}_T$ is upper bounded by $o(T)$, $\sum_{t=1}^{T} \mathbb{1}_{\{I_t \neq K\}}$ is also upper bounded by $o(T)$.

## B.2   Proof of Lemma 1

If the agent computes an estimate of $\theta_i$ by equation 4 and $V_{t,i} = \left( \sum_{k \in \tau_i(t-1)} x_k x_k^\top + \lambda \mathbf{I} \right)$, we have

$$
\begin{aligned}
&x_t^\top \hat{\theta}_{t,i} - x_t^\top (1-\alpha)\theta_K \\
=&x_t^\top V_{t,i}^{-1} \left( \sum_{k \in \tau_i(t-1)} r_{t,I_k^0} x_k \right) - x_t^\top V_{t,i}^{-1} V_{t,i}(1-\alpha)\theta_K \\
=&x_t^\top V_{t,i}^{-1} \left( \sum_{k \in \tau_i(t-1)} x_k \left( r_{t,I_k^0} - (1-\alpha)x_k^\top \theta_K \right) \right) - \lambda x_t^\top V_{t,i}^{-1}(1-\alpha)\theta_K \\
=&\sum_{k \in \tau_i(t-1)} x_t^\top V_{t,i}^{-1} x_k \left( x_k^\top \theta_{I_k^0} + \eta_k - (1-\alpha)x_k^\top \theta_K \right) - \lambda x_t^\top V_{t,i}^{-1}(1-\alpha)\theta_K,
\end{aligned}
\tag{29}
$$

and by triangle inequality,

$$
\begin{aligned}
&|x_t^\top \hat{\theta}_{t,i} - x_t^\top (1-\alpha)\theta_K| \\
\leq& \left| \sum_{k \in \tau_i(t-1)} x_t^\top V_{t,i}^{-1} x_k \left( x_k^\top \theta_{I_k^0} - (1-\alpha)x_k^\top \theta_K \right) \right| + \left| \sum_{k \in \tau_i(t-1)} x_t^\top V_{t,i}^{-1} x_k \eta_k \right| + \left| \lambda x_t^\top V_{t,i}^{-1}(1-\alpha)\theta_K \right|.
\end{aligned}
\tag{30}
$$

Now we separately bound the three items in the RHS of Equation 30.

(1) In our model, the mean reward is bounded by $0 < \langle x_t, \theta_i \rangle \leq \|x_t\|_2^2 \|\theta_i\|_2^2 = LS$. Since the mean rewards are bounded and the rewards are generated independently, we have $0 \leq \left| x_k^\top \theta_{I_k^0} - (1-\alpha)x_k^\top \theta_K \right| \leq LS$ and $\mathbb{E}[x_k^\top \theta_{I_k^0}|F_{k-1}] = (1-\alpha)x_k^\top \theta_K$, where the post-attack action $I_k^0$ is the random variable. Thus, $\left\{ x_t^\top V_{t,i}^{-1} x_k \left( x_k^\top \theta_{I_k^0} - (1-\alpha)x_k^\top \theta_K \right) \right\}_{k \in \tau_i(t-1)}$ is a bounded martingale difference sequence w.r.t the filtration $\{F_k\}_{k \in \tau_i(t-1)}$.

Then, by Azuma's inequality,

$$\mathbb{P}(|\sum_{k \in \tau_i(t-1)} x_t^\top V_{t,i}^{-1} x_k \left( x_k^\top \theta_{I_k^0} - (1-\alpha)x_k^\top \theta_K \right)| \geq B)$$

$$\leq 2\exp\left( \frac{-2B^2}{\sum_{k \in \tau_i(t-1)}(x_t^\top V_{t,i}^{-1} x_k LS)^2} \right) \tag{31}$$

$$= P_{t,i},$$

where $B$ represents confidence bound. In order to ensure the confidence bounds hold for all arms and all round $t$ simultaneously, we set $P_{t,i} = \frac{\delta}{KT}$ so

$$B = LS\sqrt{\frac{1}{2}\log\left(\frac{2KT}{\delta}\right)\sum_{k \in \tau_i(t-1)}(x_t^\top V_{t,i}^{-1} x_k)^2} \leq LS\sqrt{\frac{1}{2}\log\left(\frac{2KT}{\delta}\right)}\|x_t\|_{V_{t,i}^{-1}}, \tag{32}$$

where the last inequality is obtained from the fact that

$$\|x_t\|_{V_{t,i}^{-1}}^2 = x_t^\top V_{t,i}^{-1} \left( \sum_{k \in \tau_i(t-1)} x_k x_k^\top + \lambda\mathbf{I} \right) V_{t,i}^{-1} x_t$$

$$\geq x_t^\top V_{t,i}^{-1} \left( \sum_{k \in \tau_i(t-1)} x_k x_k^\top \right) V_{t,i}^{-1} x_t \tag{33}$$

$$= \sum_{k \in \tau_i(t-1)} (x_t^\top V_{t,i}^{-1} x_k)^2.$$

In other words, with probability $1 - \delta$, we have

$$\left| \sum_{k \in \tau_i(t-1)} x_t^\top V_{t,i}^{-1} x_k \left( x_k^\top \theta_{I_k^0} - (1-\alpha)x_k^\top \theta_K \right) \right| \leq LS\sqrt{\frac{1}{2}\log\left(\frac{2KT}{\delta}\right)}\|x_t\|_{V_{t,i}^{-1}}, \tag{34}$$

for all arms and all $t$.

(2) Note that $V_{t,i} = \sum_{k \in \tau_i(t-1)} x_k x_k^\top + \lambda\mathbf{I}$ is positive definite. We define $\langle x, y \rangle_V = x^\top V y$ as the weighted inner-product. According to Cauchy-Schwarz inequality, we have

$$\left| \sum_{k \in \tau_i(t-1)} x_t^\top V_{t,i}^{-1} x_k \eta_k \right| \leq \|x_t\|_{V_{t,i}^{-1}} \left\| \sum_{k \in \tau_i(t-1)} x_k \eta_k \right\|_{V_{t,i}^{-1}}. \tag{35}$$

Assume that $\lambda \geq L$. From Theorem 1 and Lemma 11 in (Abbasi-Yadkori et al., 2011), we know that for any $\delta > 0$, with probability at least $1 - \delta$

$$\left\| \sum_{k \in \tau_i(t-1)} x_k \eta_k \right\|_{V_{t,i}^{-1}}^2 \leq 2R^2 \log\left( \frac{K \det(V_{t,i})^{1/2}\det(\lambda\mathbf{I})^{-1/2}}{\delta} \right) \tag{36}$$

$$\leq R^2 \left( 2\log\frac{K}{\delta} + d\log\left(1 + \frac{L^2 N_i(t)}{\lambda d}\right) \right),$$

for all arms and all $t > 0$.

(3) For the third part of the right hand side of Equation 30,

$$|\lambda x_t^\top V_{t,i}^{-1}(1-\alpha)\theta_K| \leq \|(1-\alpha)\lambda\theta_K\|_{V_{t,i}^{-1}}\|x_t\|_{V_{t,i}^{-1}}. \tag{37}$$

Since $V_{t,i} \succeq \lambda \mathbf{I}$, the maximum eigenvalue of $V_{t,i}^{-1}$ is smaller or equal to $1/\lambda$. Thus,

$$\|(1-\alpha)\lambda\theta_K\|_{V_{t,i}^{-1}}^2 \leq \frac{1}{\lambda}\|(1-\alpha)\lambda\theta_K\|_2^2 \leq (1-\alpha)^2\lambda S^2.$$

In summary,

$$
\begin{aligned}
&|x_t^\top \hat{\theta}_{t,i} - x_t^\top(1-\alpha)\theta_K| \\
&\leq \left((1-\alpha)\sqrt{\lambda}S + LS\sqrt{\frac{1}{2}\log\left(\frac{2KT}{\delta}\right)} + R\sqrt{2\log\frac{K}{\delta} + d\log\left(1 + \frac{L^2 N_i(t)}{\lambda d}\right)}\right)\|x_t\|_{V_{t,i}^{-1}}.
\end{aligned}
\tag{38}
$$

### B.3 Proof of Theorem 1

For round $t$ and context $x_t$, if LinUCB pulls arm $i \neq K$, we have

$$x_t^\top \hat{\theta}_{t,K} + \beta_{t,K}\sqrt{x_t^\top V_{t,K}^{-1}x_t} \leq x_t^\top \hat{\theta}_{t,i} + \beta_{t,i}\sqrt{x_t^\top V_{t,i}^{-1}x_t}.$$

Recall $\beta_{t,i} = \sqrt{\lambda}S + R\sqrt{2\log\frac{K}{\delta} + d\log\left(1 + \frac{L^2 N_i(t)}{\lambda d}\right)}$.

Since the attacker does not attack the target arm, the confidence bound of arm $K$ does not change and $x_t^\top \theta_K \leq x_t^\top \hat{\theta}_{t,K} + \beta_{t,K}\sqrt{x_t^\top V_{t,K}^{-1}x_t}$ holds with probability $1 - \frac{\delta}{K}$.

Then, by Lemma 1,

$$
\begin{aligned}
x_t^\top \theta_K &\leq x_t^\top \hat{\theta}_{t,K} + \beta_{t,K}\sqrt{x_t^\top V_{t,K}^{-1}x_t} \\
&\leq x_t^\top \hat{\theta}_{t,i} + \beta_{t,i}\sqrt{x_t^\top V_{t,i}^{-1}x_t} \\
&\leq x_t^\top(1-\alpha)\theta_K + \beta_{t,i}\|x_t\|_{V_{t,i}^{-1}} + \left(LS\sqrt{\frac{1}{2}\log\left(\frac{2KT}{\delta}\right)} + \omega\left(N_i(t)\right)\right)\|x_t\|_{V_{t,i}^{-1}}.
\end{aligned}
\tag{39}
$$

By multiplying both sides $\mathbb{1}_{\{I_t=i\}}$ and summing over rounds, we have

$$
\begin{aligned}
&\sum_{k=1}^{T} \mathbb{1}_{\{I_k=i\}}\alpha x_k^\top \theta_K \\
&\leq \sum_{k=1}^{T} \mathbb{1}_{\{I_k=i\}}\left(\beta_{k,i} + \sqrt{\lambda}S + LS\sqrt{\frac{1}{2}\log\left(\frac{2KT}{\delta}\right)} + R\sqrt{2\log\frac{K}{\delta} + d\log\left(1 + \frac{L^2 N_i(k)}{\lambda d}\right)}\right)\|x_k\|_{V_{k,i}^{-1}}.
\end{aligned}
\tag{40}
$$

Here, we use Lemma 11 from (Abbasi-Yadkori et al., 2011) and obtain

$$
\begin{aligned}
\sum_{k=1}^{T} \mathbb{1}_{\{I_k=i\}}\|x_k\|_{V_{k,i}^{-1}}^2 &\leq 2d\log(1 + \frac{N_i(t)L^2}{d\lambda}) \\
&\leq 2d\log\left(1 + \frac{tL^2}{d\lambda}\right).
\end{aligned}
\tag{41}
$$

According to $\sum_{k=1}^{T} \mathbb{1}_{\{I_k=i\}}\|x_k\|_{V_{k,i}^{-1}} \leq \sqrt{N_i(t)\sum_{k=1}^{T}\mathbb{1}_{\{I_k=i\}}\|x_k\|_{V_{k,i}^{-1}}^2}$, we have

$$\sum_{k=1}^{T} \mathbb{1}_{\{I_k=i\}}\|x_k\|_{V_{k,i}^{-1}} \leq \sqrt{N_i(t)2d\log\left(1 + \frac{tL^2}{d\lambda}\right)}.
\tag{42}$$

Thus, we have

$$
\sum_{k=1}^{T} \mathbb{1}_{\{I_k=i\}} \alpha x_k^\top \theta_K
$$
$$
\leq \sqrt{N_i(t) 2d \log\left(1 + \frac{tL^2}{d\lambda}\right)} \left(LS\sqrt{\frac{1}{2}\log\left(\frac{2KT}{\delta}\right)} + 2\sqrt{\lambda}S + 2R\sqrt{2\log\frac{K}{\delta} + d\log\left(1 + \frac{tL^2}{\lambda d}\right)}\right). \tag{43}
$$

and

$$
\sum_{k=1}^{T} \mathbb{1}_{\{I_k=i\}}
$$
$$
\leq \frac{1}{\alpha\gamma}\sqrt{N_i(t) 2d \log\left(1 + \frac{tL^2}{d\lambda}\right)} \left(LS\sqrt{\frac{1}{2}\log\left(\frac{2KT}{\delta}\right)} + 2\sqrt{\lambda}S + 2R\sqrt{2\log\frac{K}{\delta} + d\log\left(1 + \frac{tL^2}{\lambda d}\right)}\right), \tag{44}
$$

where $\gamma = \min_{x \in \mathcal{D}}\langle x, \theta_K\rangle$. Since $N_i(t) = \sum_{k=1}^{T}\mathbb{1}_{\{I_k=i\}}$, we have

$$
N_i(t) \leq \frac{2d}{(\alpha\gamma)^2}\log\left(1 + \frac{tL^2}{d\lambda}\right)\left(2\sqrt{\lambda}S + LS\sqrt{\frac{1}{2}\log\left(\frac{2KT}{\delta}\right)} + 2R\sqrt{2\log\frac{K}{\delta} + d\log\left(1 + \frac{tL^2}{\lambda d}\right)}\right)^2. \tag{45}
$$

## C  Attack Cost Analysis of Black-box Setting

### C.1  Proof of Lemma 2

Since the estimate of $\theta_i$ obtained by the agent satisfies

$$
\hat{\theta}_{t,i}^0 = \left(V_{t,i}^0\right)^{-1}\left(\sum_{k \in \tau_i^\dagger(t-1)} w_{k,i} r_{k,I_k^0} x_k\right), \tag{46}
$$

we have

$$
x_t^\top \hat{\theta}_{t,i}^0 - x_t^\top \theta_i
$$
$$
= x_t^\top \left(V_{t,i}^0\right)^{-1}\left(\sum_{k \in \tau_i^\dagger(t-1)} w_{k,i} r_{k,I_k^0} x_k\right) - x_t^\top \left(V_{t,i}^0\right)^{-1} V_{t,i}^0 \theta_i
$$
$$
= x_t^\top \left(V_{t,i}^0\right)^{-1}\left(\sum_{k \in \tau_i^\dagger(t-1)} (w_{k,i} r_{k,I_k^0} - x_k^\top \theta_i) x_k\right) - \lambda x_t^\top \left(V_{t,i}^0\right)^{-1}\theta_i
$$
$$
= x_t^\top \left(V_{t,i}^0\right)^{-1}\left(\sum_{k \in \tau_i^\dagger(t-1)} (w_{k,i} x_k^\top \theta_{I_k^0} - x_k^\top \theta_i) x_k\right)
$$
$$
+ x_t^\top \left(V_{t,i}^0\right)^{-1}\left(\sum_{k \in \tau_i^\dagger(t-1)} w_{k,i}\eta_k\right) - \lambda x_t^\top \left(V_{t,i}^0\right)^{-1}\theta_i.
$$

Now we separately bound the three items in the RHS of Equation 47.

(1) We have $0 \leq \left|w_{k,i} x_k^\top \theta_{I_k^0} - x_k^\top \theta_i\right| \leq w_{k,i} LS$ and $\mathbb{E}[w_{k,i} x_k^\top \theta_{I_k^0}|F_{k-1}] = x_k^\top \theta_i$, where the post-attack action $I_k^0$ is the random variable. In addition, by the definition of $w_{k,i}$, we have that $w_{k,i} \leq 1/\alpha$ if $i \neq K$,

and $w_{k,i} \le 2$ if $i = K$. Thus, $\left\{ x_t^\top (V_{t,i}^0)^{-1} \sum_{k \in \tau_i^\dagger(t-1)} (w_{k,i} x_k^\top \theta_{I_k^0} - x_k^\top \theta_i) x_k \right\}_{k \in \tau_i(t-1)}$ is also a bounded martingale difference sequence w.r.t the filtration $\{F_k\}_{k \in \tau_i(t-1)}$. By following the steps in Section B.2, we have, with probability $1 - \frac{K-1}{K}\delta$, for any arm $i \ne K$ and any round $t$,

$$\left| x_t^\top \left(V_{t,i}^0\right)^{-1} \left( \sum_{k \in \tau_i^\dagger(t-1)} (w_{k,i} x_k^\top \theta_{I_k^0} - x_k^\top \theta_i) x_k \right) \right| \le \frac{LS}{\alpha} \sqrt{\frac{1}{2} \log\left(\frac{2KT}{\delta}\right)} \|x_t\|_{(V_{t,i}^0)^{-1}},$$

and with probability $1 - \frac{1}{K}\delta$, for arm $K$ and any round $t$,

$$\left| x_t^\top \left(V_{t,K}^0\right)^{-1} \left( \sum_{k \in \tau_K^\dagger(t-1)} (w_{k,K} x_k^\top \theta_{I_k^0} - x_k^\top \theta_K) x_k \right) \right| \le 2LS \sqrt{\frac{1}{2} \log\left(\frac{2KT}{\delta}\right)} \|x_t\|_{(V_{t,K}^0)^{-1}}.$$

(2) The confidence bound of the second item of the right side hand of Equation 47 can be obtained from Equation 36. With probability, $1 - \frac{K-1}{K}\delta$, for any arm $i \ne K$ and any round $t$,

$$\left| x_t^\top \left(V_{t,i}^0\right)^{-1} \left( \sum_{k \in \tau_i^\dagger(t-1)} w_{k,i} \eta_k \right) \right| \le \frac{R}{\alpha} \sqrt{2 \log \frac{K}{\delta} + d \log\left(1 + \frac{L^2 N_i^\dagger(t)}{\lambda d}\right)} \|x_t\|_{(V_{t,i}^0)^{-1}}. \tag{47}$$

With probability, $1 - \frac{1}{K}\delta$, for arm $K$ and any round $t$,

$$\left| x_t^\top \left(V_{t,K}^0\right)^{-1} \left( \sum_{k \in \tau_K^\dagger(t-1)} w_{k,K} \eta_k \right) \right| \le 2R \sqrt{2 \log \frac{K}{\delta} + d \log\left(1 + \frac{L^2 N_K^\dagger(t)}{\lambda d}\right)} \|x_t\|_{(V_{t,K}^0)^{-1}}. \tag{48}$$

(3) For the third part of the right hand side of Equation 47,

$$\left| \lambda x_t^\top \left(V_{t,i}^0\right)^{-1} \theta_i \right| \le \|\lambda \theta_i\|_{(V_{t,i}^0)^{-1}} \|x_t\|_{(V_{t,i}^0)^{-1}} \le \sqrt{\lambda} S \|x_t\|_{(V_{t,i}^0)^{-1}}. \tag{49}$$

In summary,

$$
\begin{aligned}
&|x_t^\top \hat{\theta}_{t,i}^0 - x_t^\top \theta_i| \\
&\le \phi_i \left( \sqrt{\lambda} S + LS \sqrt{\frac{1}{2} \log\left(\frac{2KT}{\delta}\right)} + R \sqrt{2 \log \frac{K}{\delta} + d \log\left(1 + \frac{L^2 N_i^\dagger(t)}{\lambda d}\right)} \right) \|x_t\|_{(V_{t,K}^0)^{-1}},
\end{aligned} \tag{50}
$$

where $\phi_i = 1/\alpha$ when $i \ne K$ and $\phi_K = 2$.

## C.2 Proof of Lemma 3

Recall the definition of $\epsilon_t$:

$$\epsilon_t = \text{clip}\left( \frac{1}{2}, \frac{(1-\alpha)\langle x_t, \hat{\theta}_{t,K}^0 \rangle - \langle x_t, \hat{\theta}_{t,I_t^\dagger}^0 \rangle}{\langle x_t, \hat{\theta}_{t,K}^0 \rangle - \langle x_t, \hat{\theta}_{t,I_t^\dagger}^0 \rangle}, 1 - \alpha \right), \tag{51}$$

and the definition of $I_t^\dagger$:

$$I_t^\dagger = \arg\min_{i \ne K} \left( \langle x_t, \hat{\theta}_{t,i}^0 \rangle - \beta_{t,i}^0 \|x_t\|_{(V_{t,i}^0)^{-1}} \right). \tag{52}$$

By Lemma 2, $\langle x_t, \hat{\theta}^0_{t,I^\dagger_t} \rangle - \beta^0_{t,I^\dagger_t} \|x_t\|_{(V^0_{t,I^\dagger_t})^{-1}} \leq \min_i \langle x_t, \theta_i \rangle$ with probability $1 - 2\delta$.

Because $\epsilon_t$ is bounded by $[1/2, 1-\alpha]$, we can analyze $\mathbb{E}[r_{t,I^0_t} | F_{t-1}, I_t]$ in four cases.

**Case 1**: when $\langle x_t, \hat{\theta}^0_{t,K} \rangle < \langle x_t, \hat{\theta}^0_{t,I^\dagger_t} \rangle$ and $\epsilon_t = 1 - \alpha$, we have

$$\mathbb{E}[r_{t,I^0_t} | F_{t-1}, I_t] = (1-\alpha)\langle x_t, \theta_K \rangle + \alpha \langle x_t, \theta_{I^\dagger_t} \rangle. \tag{53}$$

Then, by Lemma 2,

$$
\begin{aligned}
&(1-\alpha)x_t^\top \theta_K + \alpha x_t^\top \theta_{I^\dagger_t} - (1-\alpha)x_t^\top \theta_K \\
&\leq (1-\alpha)\left( x_t^\top \hat{\theta}^0_{t,K} + \beta^0_{t,K}\|x_t\|_{(V^0_{t,K})^{-1}} \right) + \alpha \left( x_t^\top \hat{\theta}^0_{t,I^\dagger_t} + \beta^0_{t,I^\dagger_t}\|x_t\|_{\left(V^0_{t,I^\dagger_t}\right)^{-1}} \right) - (1-\alpha)x_t^\top \theta_K \\
&\leq x_t^\top \hat{\theta}^0_{t,I^\dagger_t} + (1-\alpha)\beta^0_{t,K}\|x_t\|_{(V^0_{t,K})^{-1}} + \alpha\beta^0_{t,I^\dagger_t}\|x_t\|_{\left(V^0_{t,I^\dagger_t}\right)^{-1}} - (1-\alpha)x_t^\top \theta_K \\
&\leq (1-\alpha)\beta^0_{t,K}\|x_t\|_{(V^0_{t,K})^{-1}} + (1+\alpha)\beta^0_{t,I^\dagger_t}\|x_t\|_{\left(V^0_{t,I^\dagger_t}\right)^{-1}},
\end{aligned}
\tag{54}
$$

where the second inequality is obtained by the condition of Case 1 and the last inequality is obtained by $x_t^\top \hat{\theta}^0_{t,I^\dagger_t} - \beta^0_{t,I^\dagger_t}\|x_t\|_{\left(V^0_{t,I^\dagger_t}\right)^{-1}} \leq \min_i \langle x_t, \theta_i \rangle$ and Assumption 1.

On the other side, we have

$$(1-\alpha)x_t^\top \theta_K + \alpha x_t^\top \theta_{I^\dagger_t} - (1-\alpha)x_t^\top \theta_K = \alpha x_t^\top \theta_{I^\dagger_t} \geq 0. \tag{55}$$

**Case 2**: when $\langle x_t, \hat{\theta}^0_{t,K} \rangle \geq \langle x_t, \hat{\theta}^0_{t,I^\dagger_t} \rangle > (1-2\alpha)\langle x_t, \hat{\theta}^0_{t,K} \rangle$ and $\epsilon_t = 1/2$, we have

$$\mathbb{E}[r_{t,I^0_t} | F_{t-1}, I_t] = \frac{1}{2}\langle x_t, \theta_K \rangle + \frac{1}{2}\langle x_t, \theta_{I^\dagger_t} \rangle. \tag{56}$$

Then, by Lemma 2,

$$
\begin{aligned}
&\frac{1}{2}(x_t^\top \theta_K + x_t^\top \theta_{I^\dagger_t}) - (1-\alpha)x_t^\top \theta_K \\
&= \frac{1}{2}\left( x_t^\top \theta_{I^\dagger_t} - (1-2\alpha)x_t^\top \theta_K \right) \\
&\leq \frac{1}{2}\left( x_t^\top \hat{\theta}^0_{t,I^\dagger_t} + \beta^0_{t,I^\dagger_t}\|x_t\|_{\left(V^0_{t,I^\dagger_t}\right)^{-1}} - (1-2\alpha)x_t^\top \theta_K \right) \\
&\leq \beta^0_{t,I^\dagger_t}\|x_t\|_{\left(V^0_{t,I^\dagger_t}\right)^{-1}}
\end{aligned}
$$

where the last inequality is obtained by $x_t^\top \hat{\theta}^0_{t,I^\dagger_t} - \beta^0_{t,I^\dagger_t}\|x_t\|_{\left(V^0_{t,I^\dagger_t}\right)^{-1}} \leq \min_i \langle x_t, \theta_i \rangle$ and Assumption 1.

On the other side, by Lemma 2,

$$
\frac{1}{2}(x_t^\top \theta_K + x_t^\top \theta_{I_t^\dagger}) - (1-\alpha)x_t^\top \theta_K
$$

$$
\geq \frac{1}{2}\left( x_t^\top \hat{\theta}_{t,I_t^\dagger}^0 - \beta_{t,I_t^\dagger}^0 \|x_t\|_{\left(V_{t,I_t^\dagger}^0\right)^{-1}} \right) - \frac{1}{2}(1-2\alpha)\left( x_t^\top \hat{\theta}_{t,K}^0 + \beta_{t,K}^0 \|x_t\|_{(V_{t,K}^0)^{-1}} \right) \tag{57}
$$

$$
\geq -\frac{1}{2}\beta_{t,I_t^\dagger}^0 \|x_t\|_{\left(V_{t,I_t^\dagger}^0\right)^{-1}} - \frac{1}{2}(1-2\alpha)\beta_{t,K}^0 \|x_t\|_{(V_{t,K}^0)^{-1}}.
$$

where the last inequality is obtained by the conditions of Case 2.

**Case 3**: when $0 \leq \langle x_t, \hat{\theta}_{t,I_t^\dagger}^0 \rangle \leq (1-2\alpha)\langle x_t, \hat{\theta}_{t,K}^0 \rangle$ and $1/2 \leq \epsilon_t \leq 1-\alpha$, we have

$$
\mathbb{E}[r_{t,I_t^0}|F_{t-1}, I_t] = \epsilon_t \langle x_t, \theta_K \rangle + (1-\epsilon_t)\langle x_t, \theta_{I_t^\dagger} \rangle. \tag{58}
$$

We can find that

$$
\epsilon_t \langle x_t, \theta_K \rangle + (1-\epsilon_t)\langle x_t, \theta_{I_t^\dagger} \rangle - (1-\alpha)\langle x_t, \theta_K \rangle
$$

$$
=\epsilon_t(\langle x_t, \theta_K \rangle - \langle x_t, \theta_{I_t^\dagger} \rangle) + \langle x_t, \theta_{I_t^\dagger} \rangle - (1-\alpha)\langle x_t, \theta_K \rangle
$$

$$
=\epsilon_t(\langle x_t, \hat{\theta}_{t,K}^0 \rangle - \langle x_t, \hat{\theta}_{t,I_t^\dagger}^0 \rangle) + \langle x_t, \theta_{I_t^\dagger} \rangle - (1-\alpha)\langle x_t, \theta_K \rangle
$$

$$
+ \epsilon_t(\langle x_t, \hat{\theta}_{t,I_t^\dagger}^0 \rangle - \langle x_t, \theta_{I_t^\dagger} \rangle) + \epsilon_t(\langle x_t, \theta_K \rangle - \langle x_t, \hat{\theta}_{t,K}^0 \rangle) \tag{59}
$$

$$
=(1-\alpha)\langle x_t, \hat{\theta}_{t,K}^0 \rangle - \langle x_t, \hat{\theta}_{t,I_t^\dagger}^0 \rangle + \langle x_t, \theta_{I_t^\dagger} \rangle - (1-\alpha)\langle x_t, \theta_K \rangle
$$

$$
+ \epsilon_t(\langle x_t, \hat{\theta}_{t,I_t^\dagger}^0 \rangle - \langle x_t, \theta_{I_t^\dagger} \rangle) + \epsilon_t(\langle x_t, \theta_K \rangle - \langle x_t, \hat{\theta}_{t,K}^0 \rangle)
$$

$$
=(1-\alpha-\epsilon_t)\left( \langle x_t, \hat{\theta}_{t,K}^0 \rangle - \langle x_t, \theta_K \rangle \right) + (1-\epsilon_t)\left( \langle x_t, \hat{\theta}_{t,I_t^\dagger}^0 \rangle - \langle x_t, \theta_{I_t^\dagger} \rangle \right),
$$

which is equivalent to

$$
\left| \mathbb{E}[r_{t,I_t^0}|F_{t-1}, I_t] - (1-\alpha)\langle x_t, \theta_K \rangle \right|
$$

$$
\leq (1-\alpha-\epsilon_t)\beta_{t,K}^0 \|x_t\|_{(V_{t,K}^0)^{-1}} + (1-\epsilon_t)\beta_{t,I_t^\dagger}^0 \|x_t\|_{\left(V_{t,I_t^\dagger}^0\right)^{-1}}. \tag{60}
$$

**Case 4**: when $\langle x_t, \hat{\theta}_{t,I_t^\dagger}^0 \rangle < 0$ and $\epsilon_t = 1-\alpha$, we have

$$
\mathbb{E}[r_{t,I_t^0}|F_{t-1}, I_t] = (1-\alpha)\langle x_t, \theta_K \rangle + \alpha\langle x_t, \theta_{I_t^\dagger} \rangle. \tag{61}
$$

Then, by Lemma 2,

$$
(1-\alpha)x_t^\top \theta_K + \alpha x_t^\top \theta_{I_t^\dagger} - (1-\alpha)x_t^\top \theta_K
$$

$$
=\alpha x_t^\top \theta_{I_t^\dagger}
$$

$$
\leq \alpha x_t^\top \hat{\theta}_{t,I_t^\dagger}^0 + \alpha\beta_{t,I_t^\dagger}^0 \|x_t\|_{\left(V_{t,I_t^\dagger}^0\right)^{-1}} \tag{62}
$$

$$
\leq \alpha\beta_{t,I_t^\dagger}^0 \|x_t\|_{\left(V_{t,I_t^\dagger}^0\right)^{-1}},
$$

where the last inequality is obtained by the condition of Case 4. We also have

$$
(1-\alpha)x_t^\top \theta_K + \alpha x_t^\top \theta_{I_t^\dagger} - (1-\alpha)x_t^\top \theta_K = \alpha x_t^\top \theta_{I_t^\dagger} \geq 0. \tag{63}
$$

Combining these four cases, we have

$$
\begin{aligned}
&\left| \mathbb{E}[r_{t,I_t^0}|F_{t-1}, I_t] - (1-\alpha)\langle x_t, \theta_K\rangle \right| \\
&\leq (1-\alpha)\beta_{t,K}^0 \|x_t\|_{(V_{t,K}^0)^{-1}} + (1+\alpha)\beta_{t,I_t^\dagger}^0 \|x_t\|_{\left(V_{t,I_t^\dagger}^0\right)^{-1}}.
\end{aligned}
\tag{64}
$$

### C.3   Proof of Lemma 4

From Section B.2, we have, for any arm $i \neq K$,

$$
\begin{aligned}
&|x_t^\top \hat{\theta}_{t,i} - x_t^\top (1-\alpha)\theta_K| \\
&\leq \left| \sum_{k\in\tau_i(t-1)} x_t^\top V_{t,i}^{-1} x_k \left( x_k^\top \theta_{I_k^0} - (1-\alpha)x_k^\top \theta_K \right) \right| + \left| \sum_{k\in\tau_i(t-1)} x_t^\top V_{t,i}^{-1} x_k \eta_k \right| + \left| \lambda x_t^\top V_{t,i}^{-1}(1-\alpha)\theta_K \right| \\
&\leq \left| \sum_{k\in\tau_i(t-1)} x_t^\top V_{t,i}^{-1} x_k \left( x_k^\top \theta_{I_k^0} - \epsilon_k \langle x_k, \theta_K\rangle - (1-\epsilon_k)\langle x_k, \theta_{I_k^\dagger}\rangle \right) \right| \\
&\quad + \left| \sum_{k\in\tau_i(t-1)} x_t^\top V_{t,i}^{-1} x_k \left( \epsilon_k \langle x_k, \theta_K\rangle + (1-\epsilon_k)\langle x_k, \theta_{I_k^\dagger}\rangle - (1-\alpha)x_k^\top \theta_K \right) \right| \\
&\quad + \left| \sum_{k\in\tau_i(t-1)} x_t^\top V_{t,i}^{-1} x_k \eta_k \right| + \left| \lambda x_t^\top V_{t,i}^{-1}(1-\alpha)\theta_K \right|.
\end{aligned}
\tag{65}
$$

Now we separately bound the first item and second item in the RHS of Equation 65. The bounds of the third item and fourth item in the RHS of Equation 65 are provided in Section B.2.

(1) Since the mean rewards are bounded and the rewards are generated independently, we have $0 \leq \left| x_k^\top \theta_{I_k^0} - \epsilon_k \langle x_k, \theta_K\rangle - (1-\epsilon_k)\langle x_k, \theta_{I_k^\dagger}\rangle \right| \leq LS$ and $\mathbb{E}[x_k^\top \theta_{I_k^0}|F_{k-1}] = \epsilon_k \langle x_k, \theta_K\rangle + (1-\epsilon_k)\langle x_k, \theta_{I_k^\dagger}\rangle$.

Then $\left\{ x_t^\top V_{t,i}^{-1} x_k \left( x_k^\top \theta_{I_k^0} - \mathbb{E}[x_k^\top \theta_{I_k^0}|F_{k-1}] \right) \right\}_{k\in\tau_i(t-1)}$ is also a bounded martingale difference sequence w.r.t the filtration $\{F_k\}_{k\in\tau_i(t-1)}$. By following the steps in Section B.2, we have, with probability $1-\delta$, for any arm $i$ and any round $t$,

$$
\left| \sum_{k\in\tau_i(t-1)} x_t^\top V_{t,i}^{-1} x_k \left( x_k^\top \theta_{I_k^0} - \mathbb{E}[x_k^\top \theta_{I_k^0}|F_{k-1}] \right) \right| \leq LS\sqrt{\frac{1}{2}\log\left(\frac{2KT}{\delta}\right)} \|x_t\|_{V_{t,i}^{-1}}.
\tag{66}
$$

(2) From Equation 33 in Section B.2, we have

$$
\|x_t\|_{V_{t,i}^{-1}}^2 \geq \sum_{k\in\tau_i(t-1)} (x_t^\top V_{t,i}^{-1} x_k)^2.
\tag{67}
$$

Then, the second item of the right hand side of Equation 65 can be upper bounded by

$$
\left| \sum_{k \in \tau_i(t-1)} x_t^\top V_{t,i}^{-1} x_k \left( \epsilon_k \langle x_k, \theta_K \rangle + (1 - \epsilon_k) \langle x_t, \theta_{I_k^\dagger} \rangle - (1 - \alpha) x_k^\top \theta_K \right) \right|
$$

$$
\leq \sqrt{\sum_{k \in \tau_i(t-1)} \left( \mathbb{E}[r_{k,I_k^0} | F_{k-1}, I_k] - (1 - \alpha) x_k^\top \theta_K \right)^2} \sqrt{\sum_{k \in \tau_i(t-1)} (x_t^\top V_{t,i}^{-1} x_k)^2} \tag{68}
$$

$$
\leq \left( \sum_{k \in \tau_i(t-1)} \left( (1 - \alpha) \beta_{k,K}^0 \|x_k\|_{(V_{k,K}^0)^{-1}} + (1 + \alpha) \beta_{k,I_k^\dagger}^0 \|x_k\|_{\left(V_{k,I_k^\dagger}^0\right)^{-1}} \right)^2 \right)^{\frac{1}{2}} \|x_t\|_{V_{t,i}^{-1}},
$$

where the first inequality is obtained from Cauchy-Schwarz inequality, the second inequality is obtained from Lemma 3 and Equation 33.

In addition, by the fact that $(a + b)^2 \leq 2a^2 + 2b^2$ for any real number, we have

$$
\sum_{k \in \tau_i(t-1)} \left( (1 - \alpha) \beta_{k,K}^0 \|x_k\|_{(V_{k,K}^0)^{-1}} + (1 + \alpha) \beta_{k,I_k^\dagger}^0 \|x_k\|_{\left(V_{k,I_k^\dagger}^0\right)^{-1}} \right)^2
$$

$$
\leq \sum_{k \in \tau_i(t-1)} 2 \left( (1 - \alpha) \beta_{k,K}^0 \|x_k\|_{(V_{k,K}^0)^{-1}} \right)^2 + \sum_{k \in \tau_i(t-1)} 2 \left( (1 + \alpha) \beta_{k,I_k^\dagger}^0 \|x_k\|_{\left(V_{k,I_k^\dagger}^0\right)^{-1}} \right)^2. \tag{69}
$$

Here, we use Lemma 11 from (Abbasi-Yadkori et al., 2011) and get, for any arm $i$,

$$
\sum_{k \in \tau_i^\dagger(t-1)} \|x_k\|_{(V_{k,i}^0)^{-1}}^2 \leq 2d \log \left( 1 + \frac{N_i(t) L^2}{d\lambda} \right) \leq 2d \log \left( 1 + \frac{t L^2}{d\lambda} \right). \tag{70}
$$

By the fact that $\sum_i \tau_i(t-1) = \tau_K^\dagger(t-1)$, and $\sum_{i \neq K} \tau_i(t-1) = \sum_{i \neq K} \tau_i^\dagger(t-1)$, we have, for any arm $i$, $\tau_i(t-1) \subseteq \tau_K^\dagger(t-1)$, and $\tau_i(t-1) \subseteq \sum_{j \neq K} \tau_j^\dagger(t-1)$. Thus,

$$
\sum_{k \in \tau_i(t-1)} \|x_k\|_{(V_{k,K}^0)^{-1}}^2 \leq \sum_{k \in \tau_K^\dagger(t-1)} \|x_k\|_{(V_{k,K}^0)^{-1}}^2 \leq 2d \log \left( 1 + \frac{t L^2}{d\lambda} \right), \tag{71}
$$

and

$$
\sum_{k \in \tau_i(t-1)} \|x_k\|_{\left(V_{k,I_k^\dagger}^0\right)^{-1}}^2 \leq \sum_{i \neq K} \sum_{k \in \tau_i^\dagger(t-1)} \|x_k\|_{(V_{k,i}^0)^{-1}}^2 \leq 2(K-1) d \log \left( 1 + \frac{t L^2}{d\lambda} \right). \tag{72}
$$

By combining the definition of $\beta_{t,i}^0$, Equation 69, Equation 71 and Equation 72, we have

$$
\begin{aligned}
&\sum_{k \in \tau_i(t-1)} \left( (1-\alpha)\beta_{k,K}^0 \|x_k\|_{(V_{k,K}^0)^{-1}} + (1+\alpha)\beta_{k,I_k^\dagger}^0 \|x_k\|_{\left(V_{k,I_k^\dagger}^0\right)^{-1}} \right)^2 \\
&\leq \sum_{k \in \tau_i(t-1)} 2 \left( \beta_{k,K}^0 \|x_k\|_{(V_{k,K}^0)^{-1}} \right)^2 + \sum_{k \in \tau_i(t-1)} 2 \left( 2\beta_{k,I_k^\dagger}^0 \|x_k\|_{\left(V_{k,I_k^\dagger}^0\right)^{-1}} \right)^2 \\
&\leq 16d^2 \left( \omega(t) + LS\sqrt{\frac{1}{2}\log\left(\frac{2KT}{\delta}\right)} \right)^2 \log\left(1 + \frac{tL^2}{d\lambda}\right) \\
&\quad + \frac{16d^2(K-1)}{\alpha^2} \left( \omega(t) + LS\sqrt{\frac{1}{2}\log\left(\frac{2KT}{\delta}\right)} \right)^2 \log\left(1 + \frac{tL^2}{d\lambda}\right) \\
&\leq \frac{16d^2 K}{\alpha^2} \left( \omega(t) + LS\sqrt{\frac{1}{2}\log\left(\frac{2KT}{\delta}\right)} \right)^2 \log\left(1 + \frac{tL^2}{d\lambda}\right).
\end{aligned}
\tag{73}
$$

In summary, we have

$$
\begin{aligned}
&|x_t^\top \hat{\theta}_{t,i} - x_t^\top (1-\alpha)\theta_K| \\
&\leq \left( 1 + \frac{4d}{\alpha}\sqrt{K\log\left(1 + \frac{tL^2}{d\lambda}\right)} \right) \left( \omega(t) + LS\sqrt{\frac{1}{2}\log\left(\frac{2KT}{\delta}\right)} \right) \|x_t\|_{V_{t,i}^{-1}}.
\end{aligned}
\tag{74}
$$

### C.4 Proof of Theorem 2

For round $t$ and context $x_t$, if LinUCB pulls arm $i \neq K$, we have

$$
x_t^\top \hat{\theta}_{t,K} + \beta_{t,K}\sqrt{x_t^\top V_{t,K}^{-1} x_t} \leq x_t^\top \hat{\theta}_{t,i} + \beta_{t,i}\sqrt{x_t^\top V_{t,i}^{-1} x_t}.
$$

In this case, $\beta_{t,i} = \omega(N_i(t)) = \sqrt{\lambda}S + R\sqrt{2\log\frac{K}{\delta} + d\log\left(1 + \frac{L^2 N_i(t)}{\lambda d}\right)}$.

Since the attacker does not attack the target arm, the confidence bound of arm $K$ does not change and $x_t^\top \theta_K \leq x_t^\top \hat{\theta}_{t,K} + \beta_{t,K}\sqrt{x_t^\top V_{t,K}^{-1} x_t}$ holds with probability $1 - \frac{\delta}{K}$.

Thus, by Lemma 4,

$$
\begin{aligned}
x_t^\top \theta_K &\leq x_t^\top \hat{\theta}_{t,i} + \beta_{t,i}\sqrt{x_t^\top V_{t,i}^{-1} x_t} \\
&\leq x_t^\top (1-\alpha)\theta_K + \omega(N_i(t))\|x_t\|_{V_{t,i}^{-1}} \\
&\quad + \left( 1 + \frac{4d}{\alpha}\sqrt{K\log\left(1 + \frac{tL^2}{d\lambda}\right)} \right) \left( \omega(t) + LS\sqrt{\frac{1}{2}\log\left(\frac{2KT}{\delta}\right)} \right) \|x_t\|_{V_{t,i}^{-1}}.
\end{aligned}
\tag{75}
$$

By multiplying both sides by $\mathbb{1}_{\{I_t=i\}}$ and summing over rounds, we have

$$
\begin{aligned}
&\sum_{k=1}^T \mathbb{1}_{\{I_k=i\}}\alpha x_k^\top \theta_K \\
&\leq \sum_{k=1}^T \mathbb{1}_{\{I_k=i\}} \left( 2 + \frac{4d}{\alpha}\sqrt{K\log\left(1 + \frac{kL^2}{d\lambda}\right)} \right) \left( \omega(t) + LS\sqrt{\frac{1}{2}\log\left(\frac{2KT}{\delta}\right)} \right) \|x_k\|_{V_{k,i}^{-1}}.
\end{aligned}
\tag{76}
$$

Here, we use Lemma 11 from (Abbasi-Yadkori et al., 2011) and get

$$\sum_{k=1}^{T} \mathbb{1}_{\{I_k=i\}} \|x_k\|_{V_{k,i}^{-1}}^2 \leq 2d \log\left(1 + \frac{N_i(t)L^2}{d\lambda}\right) \leq 2d \log\left(1 + \frac{tL^2}{d\lambda}\right). \tag{77}$$

According to $\sum_{k=1}^{T} \mathbb{1}_{\{I_k=i\}} \|x_k\|_{V_{k,i}^{-1}} \leq \sqrt{N_i(t) \sum_{k=1}^{T} \mathbb{1}_{\{I_k=i\}} \|x_k\|_{V_{k,i}^{-1}}^2}$, we have

$$\sum_{k=1}^{T} \mathbb{1}_{\{I_k=i\}} \|x_k\|_{V_{k,i}^{-1}}^2 \leq \sqrt{N_i(t) 2d \log\left(1 + \frac{tL^2}{d\lambda}\right)}. \tag{78}$$

Thus, we have

$$\sum_{k=1}^{T} \mathbb{1}_{\{I_k=i\}} \alpha x_k^\top \theta_K$$
$$\leq \sqrt{N_i(t) 2d \log\left(1 + \frac{tL^2}{d\lambda}\right)} \left(2 + \frac{4d}{\alpha}\sqrt{K \log\left(1 + \frac{tL^2}{d\lambda}\right)}\right) \left(\omega(t) + LS\sqrt{\frac{1}{2}\log\left(\frac{2KT}{\delta}\right)}\right), \tag{79}$$

and

$$N_i(t) = \sum_{k=1}^{T} \mathbb{1}_{\{I_k=i\}}$$
$$\leq \frac{2d}{(\alpha\gamma)^2} \log\left(1 + \frac{tL^2}{d\lambda}\right) \left(2 + \frac{4d}{\alpha}\sqrt{K \log\left(1 + \frac{tL^2}{d\lambda}\right)}\right)^2 \left(\omega(t) + LS\sqrt{\frac{1}{2}\log\left(\frac{2KT}{\delta}\right)}\right)^2, \tag{80}$$

where $\gamma = \min_{x\in\mathcal{D}}\langle x, \theta_K\rangle$.

## D   Attacks on Generalized Linear Contextual Bandits

In the generalized linear model (GLM), there is a fixed, strictly increasing link function $\mu : \mathbb{R} \to \mathbb{R}$ such that the reward satisfies

$$r_{t,I_t} = \mu(\langle x_t, \theta_{I_t}\rangle) + \eta_t,$$

where $\eta_t$ is a conditionally independent zero-mean $R$-subgaussian noise and $\langle\cdot,\cdot\rangle$ denotes the inner product. If we consider the $\sigma$-algebra $F_t = \sigma(\eta_1, \ldots, \eta_t)$, $\eta_t$ becomes $F_t$ measurable.

Hence, the expected reward of arm $i$ under context $x_t$ follows the GLM setting: $\mathbb{E}[r_{t,i}] = \mu(\langle x_t, \theta_i\rangle)$ for all $t$ and all arm $i$. One can verify that $\mu(x) = x$ leads to the linear model and $\mu(x) = \exp(x)/(1 + \exp(x))$ leads to the logistic model.

We assume that the link function $\mu$ is continuously twice differentiable, Lipschitz with constant $k_\mu$ and such that $c_\mu = \inf_{\theta\in\Theta, x\in\mathcal{D}} \dot{\mu}(x^\top\theta) > 0$, where $\dot{\mu}$ denote the first derivatives of $\mu$. It can be verified that the link function of the linear model is Lipschitz with constant $k_\mu = 1$ and which of the logistic model is Lipschitz with constant $k_\mu = 1/4$.

The agent is interested in minimizing the cumulative pseudo-regret, and the cumulative pseudo-regret for the GLM can be formally written as

$$R_T = \sum_{t=1}^{T} \left(\mu(\langle x_t, \theta_{I_t^*}\rangle) - \mu(\langle x_t, \theta_{I_t}\rangle)\right), \tag{81}$$

where $I_t^* = \arg\max_i \mu(\langle x_t, \theta_i\rangle)$.

For the GLM consideblack here, $\mu$ is a strictly increasing function and $I_t^* = \arg\max_i \mu(\langle x_t, \theta_i\rangle) = \arg\max_i\langle x_t, \theta_i\rangle$. As the link function $\mu$ is strictly increasing, the target arm is not the worst arm under Assumption 1.

### D.1 Overview of UCB-GLM

For reader's convenience, we first provide a brief overview of the UCB-GLM algorithm (Li et al., 2017). The UCB-GLM algorithm is summarized in Algorithm 3.

The algorithm is simply initialized by play every arm $j$ times to ensure a unique solution of $\hat{\theta}_i$ for each arm $i$. We assume that after playing arm $i$ $J$ times, $V_i$ is invertible and the minimal eigenvalue of $V_i$ is greater or equal to $\lambda_0$ for all arm $i$. We assume that $x_t$ is drawn iid from some distribution $v$ with support in the unit ball and set $\Sigma := \mathbb{E}[x_t x_t^\top]$. Proposition 1 in (Li et al., 2017) shows that there exist positive, universal constants $D_1$ and $D_2$ such that $\lambda_{\min}(V_i) \geq \lambda_0$ with probability at least $1 - \delta$, as long as

$$J \geq \left( \frac{D_1 \sqrt{d} + D_2 \sqrt{\log(1/\delta)}}{\lambda_{\min}(\Sigma)} \right)^2 + \frac{2\lambda_0}{\lambda_{\min}(\Sigma)}. \tag{82}$$

---

**Algorithm 3** UCB-GLM  (Li et al., 2017)

---

**Require:**

    number of arms $K$, number of rounds $T$, number of initial rounds $j$.

1: Initialize for every arm $i$.
2: Play every arm $J$ times. At each time, update $\bar{V}_{I_t} \leftarrow V_{I_t} + x_t x_t^\top$.
3: **for** $t = KJ + 1, KJ + 2, \ldots, T$ **do**
4:     observe the context $x_t$.
5:     **for** $i = 1, 2, \ldots, K$ **do**
6:         Calculate the maximum-likelihood estimator $\hat{\theta}_i$ by solving the equation

$$\sum_{n \in \tau_i(t-1)} (r_n - \mu(x_n^\top \hat{\theta}_i)) x_n = 0.$$

7:         Compute the upper confidence bound: $p_{t,i} \leftarrow x_t^\top \hat{\theta}_i + \beta_{t,i} \sqrt{x_t^\top \bar{V}_i^{-1} x_t}$.
8:     **end for**
9:     Pull arm $I_t = \arg\max_i p_{t,i}$.
10:    The environment generates reward $r_t$ according to arm $I_t$.
11:    The agent receive $r_t$.
12:    $\bar{V}_{I_t} \leftarrow \bar{V}_{I_t} + x_t x_t^\top$.
13: **end for**

---

By following the setup in (Li et al., 2017), we set $\beta_{t,i} = \frac{2R}{c_\mu} \sqrt{\log \frac{K}{\delta} + d \log \left( 1 + \frac{L^2 N_i(t)}{\lambda_0 d} \right)}$ for GLM.

### D.2 White-box Attack Strategy

Similar to the white-box attack strategy for linear contextual bandits, the white-box attack strategy for GLM works as follows. When the agent chooses arm $K$, the attacker does not attack. When the agent chooses arm $I_t \neq K$, the attacker changes it to arm

$$I_t^0 = \begin{cases} K & \text{with probability} \quad \epsilon_t \\ I_t^\dagger & \text{with probability} \quad 1 - \epsilon_t \end{cases} \tag{83}$$

where $I_t^\dagger = \arg\min_i \langle x_t, \theta_i \rangle$ and

$$\epsilon_t = \frac{\mu((1-\alpha)\langle x_t, \theta_K \rangle) - \mu(\min_i \langle x_t, \theta_i \rangle)}{\mu(\langle x_t, \theta_K \rangle) - \mu(\min_i \langle x_t, \theta_i \rangle)}. \tag{84}$$

The attacker can manipulate the agent into learning some particular coefficient vectors. In particular, for arm $K$ (the target arm), the agent obtains the true coefficient vector $\theta_K$. For any arm $i \neq K$, because of

the attacks, the agent will obtain an estimate of arm $i$'s coefficient vector to be close to $(1-\alpha)\theta_K$. Under the white-box attack, when the agent pulls a non-target arm $I_t \neq K$, the mean reward received by the agent should satisfy

$$\mathbb{E}[r_{t,I_t^0}|F_{t-1}, I_t] = \mu((1-\alpha)\langle x_t, \theta_K\rangle).$$

**Proposition 2.** *For $\delta > 0$. Suppose the agent employs a generalized linear contextual bandit algorithm whose cumulative pseudo-regret is upper bounded by $o(T)$ over $T$ with probability at least $1 - \delta$. When using the proposed white-box attack scheme, with probability at least $1 - \delta$, the attacker can force the agent to pull the target arm for $T - o(T)$ times, while the total attack cost $|\mathcal{C}|$ is upper bounded by $o(T)$.*

The proof of Proposition 2 is provided in Appendix E.1. To further illustrate the proposed attack scheme, we now provide a finer analysis the impact of this attack on UCB-GLM described in Algorithm 3.

**Lemma 5.** *Under the proposed white-box attack, the estimate of $\theta_i$ for each arm $i \neq K$ obtained by UCB-GLM agent as described in Algorithm 3 satisfies*

$$|x_t^\top \hat{\theta}_{t,i} - x_t^\top (1-\alpha)\theta_K| \leq \frac{2k_\mu LS + 2R}{c_\mu}\sqrt{\log\frac{K}{\delta} + d\log\left(1 + \frac{L^2 N_i(t)}{\lambda_0 d}\right)}\|x_t\|_{\bar{V}_{t,i}^{-1}}. \tag{85}$$

The proof of Lemma 5 is provided in Appendix E.2.

**Theorem 3.** *Define $\gamma = \min_{x \in \mathcal{D}}\langle x, \theta_K\rangle$. Under the same assumptions as in Lemma 5, for any $\delta > 0$ with probability at least $1 - 2\delta$, for all $T \geq 0$, the attacker can manipulate the UCB-GLM agent into pulling the target arm in at least $T - |\mathcal{C}|$ rounds, using an attack cost*

$$|\mathcal{C}| \leq \frac{4d(K-1)}{(\alpha\gamma)^2}\log\left(1 + \frac{tL^2}{d\lambda_0}\right)\left(\frac{2k_\mu LS + 4R}{c_\mu}\right)^2\left(\log\frac{K}{\delta} + d\log\left(1 + \frac{L^2 T}{\lambda_0 d}\right)\right). \tag{86}$$

The proof of Theorem 3 is provided in Appendix E.3.

### D.3 Black-box Attack Strategy

The modified black-box attack strategy for GLM works as follows. When the agent chooses arm $K$, the attacker does not attack. When the agent chooses arm $I_t \neq K$, the attacker changes it to arm

$$I_t^0 = \begin{cases} K & \text{with probability} \quad \epsilon_t \\ I_t^\dagger & \text{with probability} \quad 1 - \epsilon_t \end{cases} \tag{87}$$

where

$$I_t^\dagger = \arg\min_{i \neq K}\left(\langle x_t, \hat{\theta}_{t,i}^0\rangle - \beta_{t,i}^0\|x_t\|_{(\bar{V}_{t,i}^0)^{-1}}\right), \tag{88}$$

and

$$\beta_{t,i}^0 = 2\phi_i\frac{k_\mu LS + R}{c_\mu}\sqrt{\log\frac{K}{\delta} + d\log\left(1 + \frac{L^2 N_i(t)}{\lambda_0 d}\right)}, \tag{89}$$

$\phi_i = \frac{k_\mu}{c_\mu \alpha}$ when $i \neq K$ and $\phi_K = 1 + \frac{k_\mu}{c_\mu}$, and

$$\epsilon_t = \text{clip}\left(\frac{c_\mu}{c_\mu + k_\mu}, \frac{\mu((1-\alpha)x_t^\top\hat{\theta}_{t,K}^0) - \mu(x_t^\top\hat{\theta}_{t,I_t^\dagger}^0)}{\mu(x_t^\top\hat{\theta}_{t,K}^0) - \mu(x_t^\top\hat{\theta}_{t,I_t^\dagger}^0)}, 1 - \alpha\frac{c_\mu}{k_\mu}\right), \tag{90}$$

with $\text{clip}(a, x, b) = \min(b, \max(x, a))$ where $a \leq b$.

For notational convenience, we set $I_t^\dagger = K$ and $\epsilon_t = 1$ when $I_t = K$. We define that, if $i \neq K$, $\tau_i^\dagger(t) := \{s : s \leq t, I_s^\dagger = i\}$ and $N_i^\dagger(t) = |\tau_i^\dagger(t)|$ ; $\tau_K^\dagger(t) := \{s : s \leq t\}$ and $N_K^\dagger(t) = |\tau_K^\dagger(t)|$.

Calculate the maximum-likelihood estimator $\hat{\theta}_{t,i}^0$ by solving the equation

$$\sum_{n \in \tau_i(t-1)^\dagger} (w_{t,i} r_n - \mu(x_n^\top \hat{\theta}_{t,i})) x_n = 0.$$

where $\bar{V}_{t,i}^0 = \sum_{k \in \tau_i^\dagger(t-1)} x_k x_k^\top$

$$w_{t,i} = \begin{cases} 1/\epsilon_t & \text{if } i = I_t^0 = K \\ 1/(1 - \epsilon_t) & \text{if } i = I_t^0 = I_t^\dagger \\ 0 & \text{if } i \neq I_t^0 \end{cases}. \tag{91}$$

First, we analyze the estimation $\hat{\theta}_{t,i}^0$ at the attacker side. We establish a confidence ellipsoid of $\langle x_t, \hat{\theta}_{t,i}^0 \rangle$ at the attacker.

**Lemma 6.** *Assume the attacker performs the proposed black-box action poisoning attack. With probability $1 - 2\delta$, we have*

$$|x_t^\top \hat{\theta}_{t,i}^0 - x_t^\top \theta_i| \leq \beta_{t,i}^0 \|x_t\|_{(\bar{V}_{t,i}^0)^{-1}}. \tag{92}$$

*holds for all arm $i$ and all $t \geq 0$ simultaneously.*

The proof of Lemma 6 is provided in Appendix E.4.

**Lemma 7.** *Under the black-box attack, with probability $1 - 2\delta$, the estimate obtained by an UCB-GLM agent satisfies*

$$\left| \mathbb{E}[r_{t,I_t^0} | F_{t-1}, I_t] - \mu((1 - \alpha) x_t^\top \theta_K) \right| \leq 2k_\mu \beta_{t,K}^0 \|x_t\|_{(\bar{V}_{t,K}^0)^{-1}} + 2k_\mu \beta_{t,I_t^\dagger}^0 \|x_t\|_{\left(\bar{V}_{t,I_t^\dagger}^0\right)^{-1}}$$

*simultaneously for all $t \geq 0$ when $I_t \neq K$.*

**Lemma 8.** *Assume the attacker performs the proposed black-box action poisoning attack. With a probability at least $1 - \frac{3K\delta}{K}$, the estimate $\hat{\theta}_{t,i}$ obtained by the UCB-GLM agent will satisfy*

$$|x_t^\top \hat{\theta}_{t,i} - x_t^\top (1 - \alpha) \theta_K|$$
$$\leq \frac{2k_\mu LS + 2R}{c_\mu} \left( 1 + \frac{16k_\mu^2 d}{c_\mu \alpha} \sqrt{K \log\left(1 + \frac{tL^2}{d\lambda_0}\right)} \right) \sqrt{\log \frac{K}{\delta} + d \log\left(1 + \frac{L^2 t}{\lambda_0 d}\right)} \|x\|_{\bar{V}_{t,i}^{-1}} \tag{93}$$

*simultaneously for all arm $i \neq K$ and all $t \geq 0$.*

**Theorem 4.** *Define $\gamma = \min_{x \in \mathcal{D}} \langle x, \theta_K \rangle$. Assume the attacker performs the proposed black-box action poisoning attack. For any $\delta > 0$ with probability at least $1 - 2\delta$, for all $T \geq 0$, the attacker can manipulate the UCB-GLM agent into pulling the target arm in at least $T - |\mathcal{C}|$ rounds, using an attack cost*

$$|\mathcal{C}| \leq \frac{4d(K-1)}{(\alpha\gamma)^2} \left( \frac{2k_\mu LS + 2R}{c_\mu} \right)^2 \log\left(1 + \frac{TL^2}{d\lambda_0}\right)$$
$$\times \left( \log \frac{K}{\delta} + d \log\left(1 + \frac{L^2 T}{\lambda_0 d}\right) \right) \left( 1 + \frac{16k_\mu^2 d}{c_\mu \alpha} \sqrt{K \log\left(1 + \frac{TL^2}{d\lambda_0}\right)} \right)^2. \tag{94}$$

The proof of Theorem 4 is provided in Appendix E.7. Theorem 4 shows that our black-box attack strategy can manipulate a UCB-GLM agent into pulling a target arm $T - O(\log^3 T)$ times with attack cost scaling as $O(\log^3 T)$. Compablack with the result for the white-box attack, the black-box attack only brings an additional $\log T$ factor.

# E    Attack Cost Analysis of GLM

## E.1    Proof of Proposition 2

Let us consider a contextual bandit problem $P_1$, in which the arm $K$ (the target arm) is associated with a fixed coefficient vector $\theta_K$ and all other arms are associated with the coefficient vector $(1-\alpha)\theta_K$. For a generalized linear contextual bandit algorithm $\mathscr{A}$, we suppose that the cumulative pseudo-regret regret of algorithm $\mathscr{A}$ for the problem $P_1$ is upper bounded with probability at least $1-\delta$ by a function $f_{\mathscr{A}}(T)$ such that $f_{\mathscr{A}}(T) = o(T)$.

Under the proposed white-box attack, when the agent pulls a non-target arm $I_t \neq K$, the mean reward received by the agent should satisfy $\mathbb{E}[r_{t,I_t^0}|F_{t-1}, I_t] = \mu((1-\alpha)\langle x_t, \theta_K \rangle)$. In the observation of the agent, the target arm becomes optimal and the non-target arms are associated with the coefficient vector $(1-\alpha)\theta_K$. For the agent, the combination of the attacker and the environment form problem $P_1$. The cumulative pseudo-regret should satisfy

$$
\begin{aligned}
R_T &= \sum_{t=1}^{T} \mathbb{1}_{\{I_t \neq K\}} \left( \mu\left(\langle x_t, \theta_K \rangle\right) - \mu\left(\langle x_t, (1-\alpha)\theta_K \rangle\right) \right) \\
&\geq \sum_{t=1}^{T} \mathbb{1}_{\{I_t \neq K\}} c_\mu \langle x_t, \alpha\theta_K \rangle \\
&\geq \sum_{t=1}^{T} \mathbb{1}_{\{I_t \neq K\}} c_\mu \gamma,
\end{aligned}
$$

which is equivalent to $\sum_{t=1}^{T} \mathbb{1}_{\{I_t \neq K\}} \leq R_T/(c_\mu \gamma)$. Since $R_T$ is upper bounded by $f_{\mathscr{A}}(T) = o(T)$, $|\mathcal{C}| = \sum_{t=1}^{T} \mathbb{1}_{\{I_t \neq K\}}$ is also upper bounded by $o(T)$.

## E.2    Proof of Lemma 5

The maximum-likelihood estimation can be written as the solution to the following equation

$$
\sum_{n \in \tau_i(t-1)} (r_n - \mu(x_n^\top \hat{\theta}_{t,i})) x_n = 0. \tag{95}
$$

Define $g_{t,i}(\theta) = \sum_{n \in \tau_i(t-1)} \mu(x_n^\top \theta)) x_n$. $g_{t,i}(\hat{\theta}_{t,i}) = \sum_{n \in \tau_i(t-1)} r_n x_n$. Since $\mu$ is continuously twice differentiable, $\nabla g_{t,i}$ is continuous, and for any $\theta \in \Theta$, $\nabla g_{t,i}(\theta) = \sum_{n \in \tau_i(t-1)} x_n x_n^\top \dot{\mu}(x_n^\top \theta))$. $\nabla g_{t,i}(\theta)$ denotes the Jacobian matrix of $g_{t,i}$ at $\theta$. By the Fundamental Theorem of Calculus,

$$
g_{t,i}(\hat{\theta}_{t,i}) - g_{t,i}((1-\alpha)\theta_K) = G_{t,i}(\hat{\theta}_{t,i} - (1-\alpha)\theta_K), \tag{96}
$$

where

$$
G_{t,i} = \int_0^1 \nabla g_{t,i}\left(s\hat{\theta}_{t,i} + (1-s)(1-\alpha)\theta_K\right) \, \mathrm{d}s. \tag{97}
$$

Note that $\nabla g_{t,i}(\theta) = \sum_{n \in \tau_i(t-1)} x_n x_n^\top \dot{\mu}(x_n^\top \theta))$. According to the assumption that $c_\mu = \inf_{\theta \in \Theta, x \in \mathcal{D}} \dot{\mu}(x^\top \theta) > 0$, we have $G_{t,i} \succeq c_\mu V_{t,i} \succeq c_\mu V_{KJ,i} \succeq \lambda_0 I \succ 0$, where in the last two step we used the assumption that the minimal eigenvalue of $V_i$ is greater or equal to $\lambda_0$ after playing arm $i$ $J$ times. Thus, $G_{t,i}$ is positive definite and non-singular. Therefore,

$$
\hat{\theta}_i - (1-\alpha)\theta_K = G_{t,i}^{-1}\left(g_{t,i}(\hat{\theta}_i) - g_{t,i}((1-\alpha)\theta_K)\right). \tag{98}
$$

For arm $K$, $g_{t,i}(\hat{\theta}_i) - g_{t,i}(\theta_K) = \sum_{n \in \tau_i(t-1)} \eta_n x_n$.

For all arm $i \neq K$, the right hand side of Equation 98 is equivalent to

$$
\begin{aligned}
& g_{t,i}(\hat{\theta}_i) - g_{t,i}((1-\alpha)\theta_K) \\
=& \sum_{n \in \tau_i(t-1)} (r_n - \mu((1-\alpha)x_n^\top \theta_K))x_n \\
=& \sum_{n \in \tau_i(t-1)} (\mu(x_n^\top \theta_{I_n^0}) - \mu((1-\alpha)x_n^\top \theta_K))x_n + \sum_{n \in \tau_i(t-1)} \eta_n x_n.
\end{aligned}
\tag{99}
$$

We set $Z_1 = \sum_{n \in \tau_i(t-1)} (\mu(x_n^\top \theta_{I_n^0}) - \mu((1-\alpha)x_n^\top \theta_K))x_n$ and $Z_2 = \sum_{n \in \tau_i(t-1)} \eta_n x_n$.

We have $g_{t,i}(\hat{\theta}_i) - g_{t,i}((1-\alpha)\theta_K) = Z_1 + Z_2$ and

$$
x_t^\top (\hat{\theta}_i - (1-\alpha)\theta_K) = x_t^\top G_{t,i}^{-1}(Z_1 + Z_2).
\tag{100}
$$

For any context $x \in \mathcal{D}$ and arm $i \neq K$, we have

$$
\begin{aligned}
& |x^\top (\hat{\theta}_{t,i} - (1-\alpha)\theta_K)| \\
=& |x^\top G_{t,i}^{-1}(Z_1 + Z_2)| \\
\leq & |x^\top G_{t,i}^{-1} Z_1| + |x^\top G_{t,i}^{-1} Z_2|.
\end{aligned}
\tag{101}
$$

We first bound $|x^\top G_{t,i}^{-1} Z_2|$. Since $G_{t,i}$ is positive definite and $G_{t,i}^{-1}$ is also positive definite, $|x^\top G_{t,i}^{-1} Z_2| \leq \|x\|_{G_{t,i}^{-1}} \|Z_2\|_{G_{t,i}^{-1}}$.

Since $G_{t,i} \succeq c_\mu V_{t,i}$ implies that $G_{t,i}^{-1} \preceq c_\mu^{-1} \bar{V}_{t,i}^{-1}$, we have $\|x\|_{G_{t,i}^{-1}} \leq \frac{1}{\sqrt{c_\mu}} \|x\|_{\bar{V}_{t,i}^{-1}}$ holds for any $x \in \mathbb{R}^d$. Thus,

$$
|x^\top G_{t,i}^{-1} Z_2| \leq \frac{1}{c_\mu} \|x\|_{\bar{V}_{t,i}^{-1}} \|Z_2\|_{\bar{V}_{t,i}^{-1}}.
\tag{102}
$$

Note that $V_{t,i} = \bar{V}_{t,i} + \lambda \mathbf{I}$. Hence, for all vector $x \in \mathbb{R}^d$

$$
\|x\|_{\bar{V}_{t,i}^{-1}}^2 = \|x\|_{V_{t,i}^{-1}}^2 + x^\top (\bar{V}_{t,i}^{-1} - V_{t,i}^{-1})x.
\tag{103}
$$

Since $(A+B)^{-1} = A^{-1} - A^{-1}B(A+B)^{-1}$,

$$
V_{t,i}^{-1} = \bar{V}_{t,i}^{-1} - \lambda \bar{V}_{t,i}^{-1} V_{t,i}^{-1}.
\tag{104}
$$

The above implies that

$$
\begin{aligned}
0 \leq & x^\top (\bar{V}_{t,i}^{-1} - V_{t,i}^{-1})x \\
=& x^\top \left( \lambda \bar{V}_{t,i}^{-1} V_{t,i}^{-1} \right) x \\
\leq & \frac{\lambda}{\lambda_0} \|x\|_{V_{t,i}^{-1}}^2.
\end{aligned}
\tag{105}
$$

and $\|x\|_{\bar{V}_{t,i}^{-1}}^2 \leq (1 + \frac{\lambda}{\lambda_0}) \|x\|_{V_{t,i}^{-1}}^2$.

From Theorem 1 and Lemma 11 in (Abbasi-Yadkori et al., 2011), we know that for any $\delta > 0$, with probability at least $1 - \delta$

$$
\begin{aligned}
\left\| \sum_{k \in \tau_i(t-1)} x_k \eta_k \right\|_{V_{t,i}^{-1}}^2 &\leq 2R^2 \log \left( \frac{K \det(V_{t,i})^{1/2} \det(\lambda \mathbf{I})^{-1/2}}{\delta} \right) \\
&\leq 2R^2 \left( \log \frac{K}{\delta} + d \log \left( 1 + \frac{L^2 N_i(t)}{\lambda d} \right) \right),
\end{aligned}
\tag{106}
$$

for all arms and all $t > 0$.

Set $\lambda = \lambda_0$, we have

$$\|Z_2\|_{\bar{V}_{t,i}^{-1}} \le 2R\sqrt{\log \frac{K}{\delta} + d\log\left(1 + \frac{L^2 N_i(t)}{\lambda_0 d}\right)}. \tag{107}$$

Now we bound $|x^\top G_{t,i}^{-1} Z_1|$. Similarly,

$$|x^\top G_{t,i}^{-1} Z_1| \le \frac{1}{c_\mu}\|x\|_{\bar{V}_{t,i}^{-1}}\|Z_1\|_{\bar{V}_{t,i}^{-1}}. \tag{108}$$

In our model, we have $0 < \langle x_t, \theta_i \rangle \le \|x_t\|_2^2 \|\theta_i\|_2^2 = LS$. Further,

$$\begin{aligned}
0 &\le \left|\mu(x_k^\top \theta_{I_k^0}) - \mu((1-\alpha)x_k^\top \theta_K)\right| \\
&\le k_\mu \left|x_k^\top \theta_{I_k^0} - (1-\alpha)x_k^\top \theta_K\right| \\
&\le k_\mu LS.
\end{aligned} \tag{109}$$

Since we have $\mathbb{E}[\mu(x_k^\top \theta_{I_k^0})|F_{k-1}] = \mu((1-\alpha)x_k^\top \theta_K)$, $\left\{\mu(x_k^\top \theta_{I_k^0}) - \mu\left((1-\alpha)x_k^\top \theta_K\right)\right\}_{k\in\tau_i(t-1)}$ is a bounded martingale difference sequence w.r.t the filtration $\{F_k\}_{k\in\tau_i(t-1)}$ and is also $k_\mu LS$-sub-Gaussian-sub-Gaussian.

From Theorem 1 and Lemma 11 in (Abbasi-Yadkori et al., 2011), we know that for any $\delta > 0$, with probability at least $1 - \frac{K-1}{K}\delta$

$$\|Z_1\|_{\bar{V}_{t,i}^{-1}} \le 2k_\mu LS\sqrt{\log \frac{K}{\delta} + d\log\left(1 + \frac{L^2 N_i(t)}{\lambda_0 d}\right)}, \tag{110}$$

for any arm $i \ne K$ and all $t > 0$.

In summary, for all arm $i \ne K$,

$$|x^\top(\hat{\theta}_i - (1-\alpha)\theta_K)| \le \frac{2k_\mu LS + 2R}{c_\mu}\sqrt{\log \frac{K}{\delta} + d\log\left(1 + \frac{L^2 N_i(t)}{\lambda_0 d}\right)}\|x\|_{\bar{V}_{t,i}^{-1}}. \tag{111}$$

### E.3 Proof of Theorem 3

For round $t$ and context $x_t$, if UCB-GLM pulls arm $i \ne K$, we have

$$x_t^\top \hat{\theta}_{t,K} + \beta_{t,K}\sqrt{x_t^\top \bar{V}_{t,K}^{-1} x_t} \le x_t^\top \hat{\theta}_{t,i} + \beta_{t,i}\sqrt{x_t^\top \bar{V}_{t,i}^{-1} x_t},$$

Recall $\beta_{t,i} = \frac{2R}{c_\mu}\sqrt{\log \frac{K}{\delta} + d\log\left(1 + \frac{L^2 N_i(t)}{\lambda_0 d}\right)}$ in GLM.

Since the attacker does not attack the target arm, the confidence bound of arm $K$ does not change and $x_t^\top \theta_K \le x_t^\top \hat{\theta}_{t,K} + \beta_{t,K}\sqrt{x_t^\top V_{t,K}^{-1} x_t}$ holds with probability $1 - \frac{\delta}{K}$.

Thus, by Lemma 5,

$$\begin{aligned}
x_t^\top \theta_K &\le x_t^\top \hat{\theta}_{t,K} + \beta_{t,K}\sqrt{x_t^\top V_{t,K}^{-1} x_t} \\
&\le x_t^\top \hat{\theta}_{t,i} + \beta_{t,i}\sqrt{x_t^\top V_{t,i}^{-1} x_t} \\
&\le x_t^\top \hat{\theta}_{t,i} + \frac{2k_\mu LS + 2R}{c_\mu}\frac{c_u}{2R}\beta_{t,i}\sqrt{x_t^\top V_{t,i}^{-1} x_t} + \beta_{t,i}\sqrt{x_t^\top V_{t,i}^{-1} x_t} \\
&\le x_t^\top(1-\alpha)\theta_K + \frac{k_\mu LS + 2R}{R}\beta_{t,i}\|x_t\|_{\bar{V}_{t,i}^{-1}}.
\end{aligned} \tag{112}$$

By multiplying both sides $\mathbb{1}_{\{I_t=i\}}$ and summing over rounds, we have

$$\sum_{k=1}^{T} \mathbb{1}_{\{I_k=i\}} \alpha x_k^\top \theta_K \le \sum_{k=1}^{T} \mathbb{1}_{\{I_k=i\}} \frac{k_\mu LS + 2R}{R} \beta_{t,i} \|x_t\|_{\bar{V}_{t,i}^{-1}}. \tag{113}$$

Here, we use Lemma 11 from (Abbasi-Yadkori et al., 2011) and obtain

$$\sum_{k=1}^{T} \mathbb{1}_{\{I_k=i\}} \|x_k\|_{V_{k,i}^{-1}}^2 \le 2d \log(1 + \frac{N_i(t)L^2}{d\lambda}) \le 2d \log\left(1 + \frac{tL^2}{d\lambda}\right). \tag{114}$$

According to $\sum_{k=1}^{T} \mathbb{1}_{\{I_k=i\}} \|x_k\|_{V_{k,i}^{-1}} \le \sqrt{N_i(t) \sum_{k=1}^{T} \mathbb{1}_{\{I_k=i\}} \|x_k\|_{V_{k,i}^{-1}}^2}$, we have

$$\sum_{k=1}^{T} \mathbb{1}_{\{I_k=i\}} \|x_k\|_{V_{k,i}^{-1}} \le \sqrt{N_i(t) 2d \log\left(1 + \frac{tL^2}{d\lambda}\right)}. \tag{115}$$

Set $\lambda = \lambda_0$, we have $\|x\|_{\bar{V}_{t,i}^{-1}}^2 \le (1 + \frac{\lambda}{\lambda_0})\|x\|_{V_{t,i}^{-1}}^2 = 2\|x\|_{V_{t,i}^{-1}}^2$. Thus, we have

$$\sum_{k=1}^{T} \mathbb{1}_{\{I_k=i\}} \alpha x_k^\top \theta_K \le \frac{k_\mu LS + 2R}{R} \beta_{t,i} \sqrt{4N_i(t)d \log\left(1 + \frac{tL^2}{d\lambda_0}\right)}, \tag{116}$$

and

$$N_i(t) = \sum_{k=1}^{T} \mathbb{1}_{\{I_k=i\}} \le \frac{4d}{(\alpha\gamma)^2} \log\left(1 + \frac{tL^2}{d\lambda_0}\right) \left(\frac{k_\mu LS + 2R}{R} \beta_{t,i}\right)^2, \tag{117}$$

where $\gamma = \min_{x\in\mathcal{D}}\langle x, \theta_K\rangle$.

### E.4   Proof of Lemma 6

The attacker calculate the maximum-likelihood estimator $\hat{\theta}_{t,i}^0$ by solving the equation

$$\sum_{n\in\tau_i(t-1)^\dagger} (w_{n,i}r_n - \mu(x_n^\top\hat{\theta}_i))x_n = 0. \tag{118}$$

Note that $g_{t,i}^0(\theta) = \sum_{n\in\tau_i^\dagger(t-1)} \mu(x_n^\top\theta))x_n$. $g_{t,i}^0(\hat{\theta}_{t,i}^0) = \sum_{n\in\tau_i^\dagger(t-1)} w_{n,i}r_n x_n$.

For all arm $i$,

$$\begin{aligned}
&g_{t,i}^0(\hat{\theta}_{t,i}^0) - g_{t,i}^0(\theta_i) \\
&= \sum_{n\in\tau_i^\dagger(t-1)} (w_{n,i}r_n - \mu(x_n^\top\theta_i))x_n \\
&= \sum_{n\in\tau_i^\dagger(t-1)} (w_{n,i}\mu(x_n^\top\theta_{I_n^0}) - \mu(x_n^\top\theta_i))x_n + \sum_{n\in\tau_i^\dagger(t-1)} w_{n,i}\eta_n x_n.
\end{aligned} \tag{119}$$

Similarly, we set $Z_3 = \sum_{n\in\tau_i^\dagger(t-1)} w_{n,i}\eta_n x_n$ and $Z_4 = \sum_{n\in\tau_i^\dagger(t-1)} (w_{n,i}\mu(x_n^\top\theta_{I_n^0}) - \mu((1-\alpha)x_n^\top\theta_K))x_n$.

We have $g_{t,i}^0(\hat{\theta}_{t,i}^0) - g_{t,i}^0(\theta_i) = Z_3 + Z_4$ and

$$x_t^\top(\hat{\theta}_{t,i}^0 - \theta_i) = x_t^\top(G_{t,i}^0)^{-1}(Z_3 + Z_4), \tag{120}$$

where

$$G_{t,i}^0 = \int_0^1 \nabla g_{t,i}^0 \left( s\hat{\theta}_{t,i}^0 + (1-s)\theta_i \right) \, ds. \tag{121}$$

For any context $x \in \mathcal{D}$, we have

$$
\begin{aligned}
&|x^\top(\hat{\theta}_{t,i}^0 - \theta_i)| \\
=&|x^\top(G_{t,i}^0)^{-1}(Z_3 + Z_4)| \\
\leq&|x^\top(G_{t,i}^0)^{-1}Z_3| + |x^\top(G_{t,i}^0)^{-1}Z_4|.
\end{aligned}
\tag{122}
$$

We first bound $|x^\top(G_{t,i}^0)^{-1}Z_3|$. We have

$$|x^\top(G_{t,i}^0)^{-1}Z_3| \leq \frac{1}{c_\mu}\|x\|_{(\bar{V}_{t,i}^0)^{-1}}\|Z_3\|_{(\bar{V}_{t,i}^0)^{-1}}, \tag{123}$$

where $\bar{V}_{t,i}^0 = \sum_{n \in \tau_i(t-1)^\dagger} x_n x_n^\top$.

Note that $V_{t,i} = \bar{V}_{t,i}^0 + \lambda\mathbf{I}$. Hence,

$$
\begin{aligned}
\|Z_3\|_{(\bar{V}_{t,i}^0)^{-1}}^2 =&\|Z_3\|_{(V_{t,i}^0)^{-1}}^2 + Z_3^\top((\bar{V}_{t,i}^0)^{-1} - (V_{t,i}^0)^{-1})Z_3 \\
\leq&(1 + \frac{\lambda}{\lambda_0})\|Z_3\|_{(V_{t,i}^0)^{-1}}^2.
\end{aligned}
\tag{124}
$$

From Theorem 1 and Lemma 11 in (Abbasi-Yadkori et al., 2011), we know that for any $\delta > 0$, with probability at least $1 - \delta$

$$
\left\| \sum_{k \in \tau_i(t-1)} x_k \eta_k \right\|_{V_{t,i}^{-1}}^2 \leq 2R^2 \log\left( \frac{K \det(V_{t,i})^{1/2} \det(\lambda\mathbf{I})^{-1/2}}{\delta} \right) \tag{125}
$$
$$
\leq 2R^2 \left( \log\frac{K}{\delta} + d\log\left(1 + \frac{L^2 N_i(t)}{\lambda d}\right) \right),
$$

for all arms and all $t > 0$.

Set $\lambda = \lambda_0$, we have

$$\|Z_3\|_{(\bar{V}_{t,i}^0)^{-1}}^2 \leq 2\phi_i R \sqrt{\log\frac{K}{\delta} + d\log\left(1 + \frac{L^2 N_i^0(t)}{\lambda_0 d}\right)}. \tag{126}$$

Now we bound $|x^\top(G_{t,i}^0)^{-1}Z_4|$. Similarly,

$$|x^\top(G_{t,i}^0)^{-1}Z_4| \leq \frac{1}{c_\mu}\|x\|_{(\bar{V}_{t,i}^0)^{-1}}\|Z_4\|_{(\bar{V}_{t,i}^0)^{-1}}. \tag{127}$$

In our model, we have $0 < \langle x_t, \theta_i \rangle \leq \|x_t\|_2^2\|\theta_i\|_2^2 = LS$. Further,

$$0 \leq \left| w_{k,i}\mu(x_k^\top \theta_{I_k^0}) - \mu((1-\alpha)x_k^\top \theta_K) \right| \leq \phi_i k_\mu LS. \tag{128}$$

Since we have $\mathbb{E}[w_{k,i}\mu(x_k^\top \theta_{I_k^0})|F_{k-1}] = \mu(x_k^\top \theta_i)$, $\left\{ w_{k,i}\mu(x_k^\top \theta_{I_k^0}) - \mu\left(x_k^\top \theta_i\right) \right\}_{k \in \tau_i(t-1)}$ is a bounded martingale difference sequence w.r.t the filtration $\{F_k\}_{k \in \tau_i(t-1)}$ and is also $\phi_i k_\mu LS$-sub-Gaussian-sub-Gaussian.

From Theorem 1 and Lemma 11 in (Abbasi-Yadkori et al., 2011), we know that for any $\delta > 0$, with probability at least $1 - \delta$

$$\|Z_4\|_{(\bar{V}_{t,i}^0)^{-1}}^2 \leq 2\phi_i k_\mu LS \sqrt{\log\frac{K}{\delta} + d\log\left(1 + \frac{L^2 N_i^0(t)}{\lambda_0 d}\right)}. \tag{129}$$

for any arm $i \neq K$ and all $t > 0$.

In summary, for all arm $i \neq K$,

$$|x^\top(\hat{\theta}_i - (1-\alpha)\theta_K)| \leq 2\phi_i \frac{k_\mu LS + R}{c_\mu} \sqrt{\log \frac{K}{\delta} + d\log\left(1 + \frac{L^2 N_i(t)}{\lambda_0 d}\right)} \|x\|_{(\bar{V}_{t,i}^0)^{-1}}. \tag{130}$$

### E.5 Proof of Lemma 7

Recall the definition of $\epsilon_t$:

$$\epsilon_t = \text{clip}\left(\frac{c_\mu}{c_\mu + k_\mu}, \frac{\mu((1-\alpha)x_t^\top \hat{\theta}_{t,K}^0) - \mu(x_t^\top \hat{\theta}_{t,I_t^\dagger}^0)}{\mu(x_t^\top \hat{\theta}_{t,K}^0) - \mu(x_t^\top \hat{\theta}_{t,I_t^\dagger}^0)}, 1 - \alpha \frac{c_\mu}{k_\mu}\right), \tag{131}$$

and the definition of $I_t^\dagger$:

$$I_t^\dagger = \arg\min_{i \neq K}\left(\langle x_t, \hat{\theta}_{t,i}^0 \rangle - \beta_{t,i}^0 \|x_t\|_{(\bar{V}_{t,i}^0)^{-1}}\right). \tag{132}$$

By Lemma 6, $\langle x_t, \hat{\theta}_{t,I_t^\dagger}^0 \rangle - \beta_{t,I_t^\dagger}^0 \|x_t\|_{(\bar{V}_{t,I_t^\dagger}^0)^{-1}} \leq \min_i \langle x_t, \theta_i \rangle$ with probability $1 - 2\delta$. Thus, with probability $1 - 2\delta$, $\mu(x_t^\top \hat{\theta}_{t,I_t^\dagger}^0) - \min_i \mu(x_t^\top \theta_i) \leq k_\mu \beta_{t,I_t^\dagger}^0 \|x_t\|_{(\bar{V}_{t,I_t^\dagger}^0)^{-1}}$.

Because $\epsilon_t$ is bounded by $[\frac{c_\mu}{c_\mu + k_\mu}, 1 - \alpha\frac{c_\mu}{k_\mu}]$, we can analyze $\mathbb{E}[r_{t,I_t^0}|F_{t-1}, I_t]$ in four cases.

**Case 1**: when $\langle x_t, \hat{\theta}_{t,K}^0 \rangle < \langle x_t, \hat{\theta}_{t,I_t^\dagger}^0 \rangle$, we have $\epsilon_t = 1 - \alpha\frac{c_\mu}{k_\mu}$ and $\mu(\langle x_t, \hat{\theta}_{t,K}^0 \rangle) < \mu(\langle x_t, \hat{\theta}_{t,I_t^\dagger}^0 \rangle)$. Thus,

$$\mathbb{E}[r_{t,I_t^0}|F_{t-1}, I_t] = (1 - \alpha\frac{c_\mu}{k_\mu})\mu(x_t^\top \theta_K) + \alpha\frac{c_\mu}{k_\mu}\mu(x_t^\top \theta_{I_t^\dagger}). \tag{133}$$

Then, by Lemma 6,

$$(1 - \alpha\frac{c_\mu}{k_\mu})\mu(x_t^\top \theta_K) + \alpha\frac{c_\mu}{k_\mu}\mu(x_t^\top \theta_{I_t^\dagger}) - \mu((1-\alpha)x_t^\top \theta_K)$$

$$\leq(1 - \alpha\frac{c_\mu}{k_\mu})\left(\mu(x_t^\top \hat{\theta}_{t,K}^0) + k_\mu \beta_{t,K}^0 \|x_t\|_{(\bar{V}_{t,K}^0)^{-1}}\right) + \alpha\frac{c_\mu}{k_\mu}\left(\mu(x_t^\top \hat{\theta}_{t,I_t^\dagger}^0) + k_\mu \beta_{t,I_t^\dagger}^0 \|x_t\|_{\left(\bar{V}_{t,I_t^\dagger}^0\right)^{-1}}\right)$$

$$- \mu((1-\alpha)x_t^\top \theta_K) \tag{134}$$

$$\leq\mu(x_t^\top \hat{\theta}_{t,I_t^\dagger}^0) + \alpha c_\mu \beta_{t,I_t^\dagger}^0 \|x_t\|_{\left(\bar{V}_{t,I_t^\dagger}^0\right)^{-1}} + (1 - \alpha\frac{c_\mu}{k_\mu})k_\mu \beta_{t,K}^0 \|x_t\|_{(\bar{V}_{t,K}^0)^{-1}} - \mu((1-\alpha)x_t^\top \theta_K)$$

$$\leq(k_\mu - \alpha c_\mu)\beta_{t,K}^0 \|x_t\|_{(\bar{V}_{t,K}^0)^{-1}} + (k_\mu + \alpha c_\mu)\beta_{t,I_t^\dagger}^0 \|x_t\|_{\left(\bar{V}_{t,I_t^\dagger}^0\right)^{-1}},$$

where the second inequality is obtains by $\mu(\langle x_t, \hat{\theta}_{t,K}^0 \rangle) < \mu(\langle x_t, \hat{\theta}_{t,I_t^\dagger}^0 \rangle)$ and the last inequality is obtained by $\mu(x_t^\top \hat{\theta}_{t,I_t^\dagger}^0) - \min_i \mu(x_t^\top \theta_i) \leq k_\mu \beta_{t,I_t^\dagger}^0 \|x_t\|_{(\bar{V}_{t,I_t^\dagger}^0)^{-1}}$ and Assumption 1.

On the other side, we have

$$(1 - \alpha \frac{c_\mu}{k_\mu})\mu(x_t^\top \theta_K) + \alpha \frac{c_\mu}{k_\mu}\mu(x_t^\top \theta_{I_t^\dagger}) - \mu((1-\alpha)x_t^\top \theta_K)$$

$$\geq (1 - \alpha \frac{c_\mu}{k_\mu})\left(\mu(x_t^\top \hat{\theta}_{t,K}^0) - k_\mu \beta_{t,K}^0 \|x_t\|_{(\bar{V}_{t,K}^0)^{-1}}\right) + \alpha \frac{c_\mu}{k_\mu}\left(\mu(x_t^\top \hat{\theta}_{t,I_t^\dagger}^0) - k_\mu \beta_{t,I_t^\dagger}^0 \|x_t\|_{\left(\bar{V}_{t,I_t^\dagger}^0\right)^{-1}}\right)$$

$$- \mu((1-\alpha)x_t^\top \theta_K)$$

$$\geq \mu(x_t^\top \hat{\theta}_{t,K}^0) - \mu((1-\alpha)x_t^\top \theta_K) - \alpha c_\mu \beta_{t,I_t^\dagger}^0 \|x_t\|_{\left(\bar{V}_{t,I_t^\dagger}^0\right)^{-1}} - (1 - \alpha \frac{c_\mu}{k_\mu})k_\mu \beta_{t,K}^0 \|x_t\|_{(\bar{V}_{t,K}^0)^{-1}} \quad (135)$$

$$\geq \mu(x_t^\top \theta_K) - \mu((1-\alpha)x_t^\top \theta_K) - \alpha c_\mu \beta_{t,I_t^\dagger}^0 \|x_t\|_{\left(\bar{V}_{t,I_t^\dagger}^0\right)^{-1}} - (2 - \alpha \frac{c_\mu}{k_\mu})k_\mu \beta_{t,K}^0 \|x_t\|_{(\bar{V}_{t,K}^0)^{-1}}$$

$$\geq - (2k_\mu - \alpha c_\mu)\beta_{t,K}^0 \|x_t\|_{(\bar{V}_{t,K}^0)^{-1}} - \alpha c_\mu \beta_{t,I_t^\dagger}^0 \|x_t\|_{\left(\bar{V}_{t,I_t^\dagger}^0\right)^{-1}}.$$

**Case 2**: when $\mu(x_t^\top \hat{\theta}_{t,K}^0) \geq \mu(x_t^\top \hat{\theta}_{t,I_t^\dagger}^0) > (1 + \frac{c_\mu}{k_\mu})\mu((1-\alpha)x_t^\top \hat{\theta}_{t,K}^0) - \frac{c_\mu}{k_\mu}\mu(x_t^\top \hat{\theta}_{t,K}^0)$ and $\epsilon_t = \frac{c_\mu}{c_\mu + k_\mu}$, we have

$$\mathbb{E}[r_{t,I_t^0}|F_{t-1}, I_t] = \frac{c_\mu}{c_\mu + k_\mu}\mu(x_t^\top \theta_K) + \frac{k_\mu}{c_\mu + k_\mu}\mu(x_t^\top \theta_{I_t^\dagger}). \quad (136)$$

By Lemma 6, with probability $1 - 2\delta$,

$$\mu(x_t^\top \hat{\theta}_{t,I_t^\dagger}^0) - \min_i \mu(x_t^\top \theta_i) \leq k_\mu \beta_{t,I_t^\dagger}^0 \|x_t\|_{(\bar{V}_{t,I_t^\dagger}^0)^{-1}}.$$

Since $\min_i x_t^\top \theta_i \leq (1 - 2\alpha)x_t^\top \theta_K$, we have

$$\mu(x_t^\top \hat{\theta}_{t,I_t^\dagger}^0) \leq \mu((1-2\alpha)x_t^\top \theta_K) + k_\mu \beta_{t,I_t^\dagger}^0 \|x_t\|_{(\bar{V}_{t,I_t^\dagger}^0)^{-1}}$$

and then

$$\mu(x_t^\top \theta_{I_t^\dagger}) \leq \mu((1-2\alpha)x_t^\top \theta_K) + 2k_\mu \beta_{t,I_t^\dagger}^0 \|x_t\|_{(\bar{V}_{t,I_t^\dagger}^0)^{-1}}.$$

Thus,

$$\frac{c_\mu \mu(x_t^\top \theta_K)}{c_\mu + k_\mu} + \frac{k_\mu \mu(x_t^\top \theta_{I_t^\dagger})}{c_\mu + k_\mu} - \mu((1-\alpha)x_t^\top \theta_K)$$

$$= \frac{c_\mu}{c_\mu + k_\mu}\mu(x_t^\top \theta_K) - \frac{c_\mu}{c_\mu + k_\mu}\mu((1-\alpha)x_t^\top \theta_K)$$

$$+ \frac{k_\mu}{c_\mu + k_\mu}\mu(x_t^\top \theta_{I_t^\dagger}) - \frac{k_\mu}{c_\mu + k_\mu}\mu((1-\alpha)x_t^\top \theta_K) \quad (137)$$

$$\leq \frac{c_\mu}{c_\mu + k_\mu}\left(\mu(x_t^\top \theta_K) - \mu((1-\alpha)x_t^\top \theta_K)\right)$$

$$+ \frac{k_\mu}{c_\mu + k_\mu}\left(\mu((1-2\alpha)x_t^\top \theta_K) - \mu((1-\alpha)x_t^\top \theta_K)\right) + \frac{2k_\mu^2}{c_\mu + k_\mu}\beta_{t,I_t^\dagger}^0 \|x_t\|_{(\bar{V}_{t,I_t^\dagger}^0)^{-1}}.$$

According to the definition of $k_\mu$ and $c_\mu$ and Lemma 6,

$$
\begin{aligned}
&\frac{c_\mu \mu(x_t^\top \theta_K)}{c_\mu + k_\mu} + \frac{k_\mu \mu(x_t^\top \theta_{I_t^\dagger})}{c_\mu + k_\mu} - \mu((1-\alpha)x_t^\top \theta_K) \\
\leq & \frac{c_\mu}{c_\mu + k_\mu} k_\mu \left( x_t^\top \theta_K - (1-\alpha)x_t^\top \theta_K \right) + \frac{k_\mu}{c_\mu + k_\mu} c_\mu \left( (1-2\alpha)x_t^\top \theta_K - (1-\alpha)x_t^\top \theta_K \right) \\
& + \frac{2k_\mu^2}{c_\mu + k_\mu} \beta_{t,I_t^\dagger}^0 \|x_t\|_{(\bar{V}_{t,I_t^\dagger}^0)^{-1}} \\
= & \frac{2k_\mu^2}{c_\mu + k_\mu} \beta_{t,I_t^\dagger}^0 \|x_t\|_{(\bar{V}_{t,I_t^\dagger}^0)^{-1}}.
\end{aligned}
\tag{138}
$$

In addition, by Lemma 6,

$$
\begin{aligned}
&\frac{c_\mu \mu(x_t^\top \theta_K)}{c_\mu + k_\mu} + \frac{k_\mu \mu(x_t^\top \theta_{I_t^\dagger})}{c_\mu + k_\mu} - \mu((1-\alpha)x_t^\top \theta_K) \\
= & \frac{c_\mu \mu(x_t^\top \theta_K)}{c_\mu + k_\mu} + \frac{k_\mu \mu(x_t^\top \theta_{I_t^\dagger})}{c_\mu + k_\mu} - \mu((1-\alpha)x_t^\top \hat{\theta}_{t,K}^0) \\
& + \mu((1-\alpha)x_t^\top \hat{\theta}_{t,K}^0) - \mu((1-\alpha)x_t^\top \theta_K) \\
\geq & \frac{c_\mu}{c_\mu + k_\mu} \mu(x_t^\top \theta_K) + \frac{k_\mu}{c_\mu + k_\mu} \mu(x_t^\top \theta_{I_t^\dagger}) - \frac{c_\mu}{c_\mu + k_\mu} \mu(x_t^\top \hat{\theta}_{t,K}^0) - \frac{k_\mu}{c_\mu + k_\mu} \mu(x_t^\top \hat{\theta}_{t,I_t^\dagger}^0) \\
& + \mu((1-\alpha)x_t^\top \hat{\theta}_{t,K}^0) - \mu((1-\alpha)x_t^\top \theta_K) \\
\geq & - (1 - \alpha + \frac{c_\mu}{c_\mu + k_\mu}) k_\mu \beta_{t,K}^0 \|x_t\|_{(\bar{V}_{t,K}^0)^{-1}} - \frac{k_\mu}{c_\mu + k_\mu} k_\mu \beta_{t,I_t^\dagger}^0 \|x_t\|_{\left(\bar{V}_{t,I_t^\dagger}^0\right)^{-1}},
\end{aligned}
\tag{139}
$$

where the first inequality is obtained by the condition of case 2:

$$
\mu(x_t^\top \hat{\theta}_{t,I_t^\dagger}^0) > (1 + \frac{c_\mu}{k_\mu}) \mu((1-\alpha)x_t^\top \hat{\theta}_{t,K}^0) - \frac{c_\mu}{k_\mu} \mu(x_t^\top \hat{\theta}_{t,K}^0)
$$

which is equivalent to

$$
\frac{c_\mu \mu(x_t^\top \hat{\theta}_{t,K}^0)}{c_\mu + k_\mu} + \frac{k_\mu \mu(x_t^\top \hat{\theta}_{t,I_t^\dagger}^0)}{c_\mu + k_\mu} > \mu((1-\alpha)x_t^\top \hat{\theta}_{t,K}^0).
$$

**Case 3**: when the attacker's estimates satisfy

$$
\begin{aligned}
& \frac{k_\mu}{\alpha c_\mu} \mu((1-\alpha)x_t^\top \hat{\theta}_{t,K}^0) - (\frac{k_\mu}{\alpha c_\mu} - 1) \mu(x_t^\top \hat{\theta}_{t,K}^0) \\
\leq & \mu(x_t^\top \hat{\theta}_{t,I_t^\dagger}^0) \\
\leq & (1 + \frac{c_\mu}{k_\mu}) \mu((1-\alpha)x_t^\top \hat{\theta}_{t,K}^0) - \frac{c_\mu}{k_\mu} \mu(x_t^\top \hat{\theta}_{t,K}^0)
\end{aligned}
\tag{140}
$$

and hence $\frac{c_\mu}{c_\mu + k_\mu} \leq \epsilon_t \leq 1 - \alpha \frac{c_\mu}{k_\mu}$, we have

$$
\mathbb{E}[r_{t,I_t^0} | F_{t-1}, I_t] = \epsilon_t \mu(x_t^\top \theta_K) + (1 - \epsilon_t) \mu(x_t^\top \theta_{I_t^\dagger}).
\tag{141}
$$

We can find that

$$
\begin{aligned}
&\epsilon_t \mu(x_t^\top \theta_K) + (1 - \epsilon_t)\mu(x_t^\top \theta_{I_t^\dagger}) - \mu((1-\alpha)x_t^\top \theta_K) \\
=&\epsilon_t(\mu(x_t^\top \theta_K) - \mu(x_t^\top \theta_{I_t^\dagger})) + \mu(x_t^\top \theta_{I_t^\dagger}) - \mu((1-\alpha)x_t^\top \theta_K) \\
=&\epsilon_t(\mu(x_t^\top \hat\theta_{t,K}^0) - \mu(x_t^\top \hat\theta_{t,I_t^\dagger}^0)) + \epsilon_t(\mu(x_t^\top \hat\theta_{t,I_t^\dagger}^0) - \mu(x_t^\top \theta_{I_t^\dagger})) + \epsilon_t(\mu(x_t^\top \theta_K) - \mu(x_t^\top \hat\theta_{t,K}^0)) \\
&+ \mu(x_t^\top \theta_{I_t^\dagger}) - \mu((1-\alpha)x_t^\top \theta_K) \\
=&\mu((1-\alpha)x_t^\top \hat\theta_{t,K}^0) - \mu(x_t^\top \hat\theta_{t,I_t^\dagger}^0) + \epsilon_t(\mu(x_t^\top \hat\theta_{t,I_t^\dagger}^0) - \mu(x_t^\top \theta_{I_t^\dagger})) + \epsilon_t(\mu(x_t^\top \theta_K) - \mu(x_t^\top \hat\theta_{t,K}^0)) \\
&+ \mu(x_t^\top \theta_{I_t^\dagger}) - \mu((1-\alpha)x_t^\top \theta_K) \\
=&\mu((1-\alpha)x_t^\top \hat\theta_{t,K}^0) - \mu((1-\alpha)x_t^\top \theta_K) + \epsilon_t(\mu(x_t^\top \theta_K) - \mu(x_t^\top \hat\theta_{t,K}^0)) + (1-\epsilon_t)\left(\mu(x_t^\top \theta_{I_t^\dagger}) - \mu(x_t^\top \hat\theta_{t,I_t^\dagger}^0)\right).
\end{aligned}
\tag{142}
$$

From Lemma 6,

$$
\begin{aligned}
&\left| \mathbb{E}[r_{t,I_t^0}|F_{t-1}, I_t] - \mu((1-\alpha)x_t^\top \theta_K) \right| \\
\leq& (1 - \alpha + \epsilon_t)k_\mu \beta_{t,K}^0 \|x_t\|_{(\bar{V}_{t,K}^0)^{-1}} \\
&+ (1-\epsilon_t)k_\mu \beta_{t,I_t^\dagger}^0 \|x_t\|_{\left(\bar{V}_{t,I_t^\dagger}^0\right)^{-1}}.
\end{aligned}
\tag{143}
$$

**Case 4**: when $\mu(x_t^\top \hat\theta_{t,I_t^\dagger}^0) < \frac{k_\mu}{\alpha c_\mu}\mu((1-\alpha)x_t^\top \hat\theta_{t,K}^0) - (\frac{k_\mu}{\alpha c_\mu} - 1)\mu(x_t^\top \hat\theta_{t,K}^0)$ and $\epsilon_t = 1 - \alpha\frac{c_\mu}{k_\mu}$, we have

$$
\mathbb{E}[r_{t,I_t^0}|F_{t-1}, I_t] = (1 - \alpha\frac{c_\mu}{k_\mu})\mu(x_t^\top \theta_K) + \alpha\frac{c_\mu}{k_\mu}\mu(x_t^\top \theta_{I_t^\dagger}).
\tag{144}
$$

Then, by Lemma 6,

$$
\begin{aligned}
&(1 - \alpha\frac{c_\mu}{k_\mu})\mu(x_t^\top \theta_K) + \alpha\frac{c_\mu}{k_\mu}\mu(x_t^\top \theta_{I_t^\dagger}) - \mu((1-\alpha)x_t^\top \theta_K) \\
\leq& (1 - \alpha\frac{c_\mu}{k_\mu})\mu(x_t^\top \theta_K) - \mu((1-\alpha)x_t^\top \theta_K) + \alpha\frac{c_\mu}{k_\mu}\mu(x_t^\top \hat\theta_{t,I_t^\dagger}^0) + \alpha c_\mu \beta_{t,I_t^\dagger}^0 \|x_t\|_{(\bar{V}_{t,I_t^\dagger}^0)^{-1}} \\
<& (1 - \alpha\frac{c_\mu}{k_\mu})\mu(x_t^\top \theta_K) - \mu((1-\alpha)x_t^\top \theta_K) + \mu((1-\alpha)x_t^\top \hat\theta_{t,K}^0) - (1 - \alpha\frac{c_\mu}{k_\mu})\mu(x_t^\top \hat\theta_{t,K}^0) + \alpha c_\mu \beta_{t,I_t^\dagger}^0 \|x_t\|_{(\bar{V}_{t,I_t^\dagger}^0)^{-1}} \\
\leq& (k_\mu - c_\mu)\beta_{t,K}^0 \|x_t\|_{(\bar{V}_{t,K}^0)^{-1}} + \alpha c_\mu \beta_{t,I_t^\dagger}^0 \|x_t\|_{(\bar{V}_{t,I_t^\dagger}^0)^{-1}}.
\end{aligned}
\tag{145}
$$

Since $x_t^\top \theta_{I_t^\dagger} > 0$, $\mu(x_t^\top \theta_K) - \mu(x_t^\top \theta_{I_t^\dagger}) \leq k_\mu x_t^\top \theta_K$. Hence, we also have

$$
\begin{aligned}
&(1 - \alpha\frac{c_\mu}{k_\mu})\mu(x_t^\top \theta_K) + \alpha\frac{c_\mu}{k_\mu}\mu(x_t^\top \theta_{I_t^\dagger}) - \mu((1-\alpha)x_t^\top \theta_K) \\
=&\mu(x_t^\top \theta_K) - \mu((1-\alpha)x_t^\top \theta_K) - \alpha\frac{c_\mu}{k_\mu}\left(\mu(x_t^\top \theta_K) - \mu(x_t^\top \theta_{I_t^\dagger})\right) \\
\geq& c_\mu \alpha x_t^\top \theta_K - \alpha\frac{c_\mu}{k_\mu}k_\mu x_t^\top \theta_K = 0.
\end{aligned}
\tag{146}
$$

Combining these four cases, we have

$$
\begin{aligned}
&\left| \mathbb{E}[r_{t,I_t^0}|F_{t-1}, I_t] - \mu((1-\alpha)x_t^\top \theta_K) \right| \\
\leq& 2k_\mu \beta_{t,K}^0 \|x_t\|_{(\bar{V}_{t,K}^0)^{-1}} + 2k_\mu \beta_{t,I_t^\dagger}^0 \|x_t\|_{\left(\bar{V}_{t,I_t^\dagger}^0\right)^{-1}}.
\end{aligned}
\tag{147}
$$

### E.6 Proof of Lemma 8

The agent's maximum-likelihood estimation can be written as the solution to the following equation

$$\sum_{n \in \tau_i(t-1)} (r_n - \mu(x_n^\top \hat{\theta}_{t,i}))x_n = 0. \tag{148}$$

As described in the section E.2, we have $g_{t,i}(\hat{\theta}_i) - g_{t,i}((1-\alpha)\theta_K) = Z_1 + Z_2$ and

$$x_t^\top(\hat{\theta}_i - (1-\alpha)\theta_K) = x_t^\top G_{t,i}^{-1}(Z_1 + Z_2). \tag{149}$$

We set $Z_1 = \sum_{n \in \tau_i(t-1)}(\mu(x_n^\top \theta_{I_n^0}) - \mu((1-\alpha)x_n^\top \theta_K))x_n$ and $Z_2 = \sum_{n \in \tau_i(t-1)} \eta_n x_n$.

In the white-box attack case, we have $\mathbb{E}[\mu(x_k^\top \theta_{I_k^0})|F_{k-1}] = \mu((1-\alpha)x_k^\top \theta_K)$ and hence $\mathbb{E}[Z_1|F_{k-1}] = \mathbf{0}$. Under the proposed black-box attack, $\mathbb{E}[Z_1|F_{k-1}] \neq \mathbf{0}$ but

$$\left| \mathbb{E}[\mu(x_t^\top \theta_{I_t^0})|F_{t-1}, I_t] - (1-\alpha)\langle x_t, \theta_K \rangle \right|$$
$$\leq 2k_\mu \beta_{t,K}^0 \|x_t\|_{(\bar{V}_{t,K}^0)^{-1}} + 2k_\mu \beta_{t,I_t^\dagger}^0 \|x_t\|_{\left(\bar{V}_{t,I_t^\dagger}^0\right)^{-1}}. \tag{150}$$

We set $Z_1 = Z_5 + Z_6$, where

$$Z_5 = \sum_{n \in \tau_i(t-1)} (\mu(x_n^\top \theta_{I_n^0}) - \mathbb{E}[\mu(x_t^\top \theta_{I_t^0})|F_{t-1}, I_t])x_n$$

and

$$Z_6 = \sum_{n \in \tau_i(t-1)} (\mathbb{E}[\mu(x_t^\top \theta_{I_t^0})|F_{t-1}, I_t] - \mu((1-\alpha)x_n^\top \theta_K))x_n.$$

For any context $x \in \mathcal{D}$ and arm $i \neq K$, we have

$$|x^\top(\hat{\theta}_{t,i} - (1-\alpha)\theta_K)|$$
$$\leq |x^\top G_{t,i}^{-1} Z_2| + |x^\top G_{t,i}^{-1} Z_5| + |x^\top G_{t,i}^{-1} Z_6|. \tag{151}$$

Since we have $\left\{ \mu(x_k^\top \theta_{I_k^0}) - \mathbb{E}[\mu(x_k^\top \theta_{I_k^0})|F_{k-1}, I_k] \right\}_{k \in \tau_i(t-1)}$ is a bounded martingale difference sequence w.r.t the filtration $\{F_k, I_k\}_{k \in \tau_i(t-1)}$ and is also $k_\mu LS$-sub-Gaussian-sub-Gaussian.

From Theorem 1 and Lemma 11 in (Abbasi-Yadkori et al., 2011), we know that for any $\delta > 0$, with probability at least $1 - \frac{K-1}{K}\delta$

$$\|Z_5\|_{\bar{V}_{t,i}^{-1}} \leq 2k_\mu LS \sqrt{\log \frac{K}{\delta} + d \log \left(1 + \frac{L^2 N_i(t)}{\lambda_0 d}\right)}, \tag{152}$$

for any arm $i \neq K$ and all $t > 0$.

We have the fact that

$$\|x_t\|_{V_{t,i}^{-1}}^2 = x_t^\top V_{t,i}^{-1} \left( \sum_{k \in \tau_i(t-1)} x_k x_k^\top + \lambda \mathbf{I} \right) V_{t,i}^{-1} x_t$$

$$\geq x_t^\top V_{t,i}^{-1} \left( \sum_{k \in \tau_i(t-1)} x_k x_k^\top \right) V_{t,i}^{-1} x_t \tag{153}$$

$$= \sum_{k \in \tau_i(t-1)} (x_t^\top V_{t,i}^{-1} x_k)^2.$$

Hence,

$$\|x_t\|^2_{G^{-1}_{t,i}} = x_t^\top G^{-1}_{t,i} G_{t,i} G^{-1}_{t,i} x_t$$

$$\geq c_\mu x_t^\top V^{-1}_{t,i} \left( \sum_{k \in \tau_i(t-1)} x_k x_k^\top \right) V^{-1}_{t,i} x_t \tag{154}$$

$$= c_\mu \sum_{k \in \tau_i(t-1)} (x_t^\top G^{-1}_{t,i} x_k)^2,$$

and hence $\sum_{k \in \tau_i(t-1)} (x_t^\top G^{-1}_{t,i} x_k)^2 \leq \frac{1}{c_\mu^2} \|x_t\|^2_{V^{-1}_{t,i}}$.

Then, $|x^\top G^{-1}_{t,i} Z_6|$ can be upper bounded by

$$|x^\top G^{-1}_{t,i} Z_6|$$

$$\leq \sqrt{\sum_{k \in \tau_i(t-1)} \left( \mathbb{E}[r_{k,I^0_k} | F_{k-1}, I_k] - \mu((1-\alpha)x_k^\top \theta_K) \right)^2} \sqrt{\sum_{k \in \tau_i(t-1)} (x_t^\top G^{-1}_{t,i} x_k)^2} \tag{155}$$

$$\leq \left( \sum_{k \in \tau_i(t-1)} \left( 2k_\mu \beta^0_{k,K} \|x_k\|_{(\bar{V}^0_{k,K})^{-1}} + 2k_\mu \beta^0_{k,I^\dagger_k} \|x_k\|_{\left(\bar{V}^0_{k,I^\dagger_k}\right)^{-1}} \right)^2 \right)^{\frac{1}{2}} \frac{1}{c_\mu} \|x_t\|_{\bar{V}^{-1}_{t,i}},$$

where the first inequality is obtained from Cauchy-Schwarz inequality, the second inequality is obtained from Lemma 7 and Equation 153.

In addition, by the fact that $(a+b)^2 \leq 2a^2 + 2b^2$ for any real number, we have

$$\sum_{k \in \tau_i(t-1)} \left( 2k_\mu \beta^0_{k,K} \|x_k\|_{(\bar{V}^0_{k,K})^{-1}} + 2k_\mu \beta^0_{k,I^\dagger_k} \|x_k\|_{\left(\bar{V}^0_{k,I^\dagger_k}\right)^{-1}} \right)^2$$

$$\leq \sum_{k \in \tau_i(t-1)} 2 \left( 2k_\mu \beta^0_{k,K} \|x_k\|_{(\bar{V}^0_{k,K})^{-1}} \right)^2 + \sum_{k \in \tau_i(t-1)} 2 \left( 2k_\mu \beta^0_{k,I^\dagger_k} \|x_k\|_{\left(\bar{V}^0_{k,I^\dagger_k}\right)^{-1}} \right)^2. \tag{156}$$

Here, we use Lemma 11 from (Abbasi-Yadkori et al., 2011) and get, for any arm $i$,

$$\sum_{k \in \tau^\dagger_i(t-1)} \|x_k\|^2_{(V^0_{k,i})^{-1}} \leq 2d \log \left( 1 + \frac{N_i(t)L^2}{d\lambda} \right) \leq 2d \log \left( 1 + \frac{tL^2}{d\lambda} \right). \tag{157}$$

Set $\lambda = \lambda_0$, we have $\|x\|^2_{\bar{V}^{-1}_{t,i}} \leq (1 + \frac{\lambda}{\lambda_0}) \|x\|^2_{V^{-1}_{t,i}} \leq 2\|x\|^2_{V^{-1}_{t,i}}$.

By the fact that $\sum_i \tau_i(t-1) = \tau^\dagger_K(t-1)$, and $\sum_{i \neq K} \tau_i(t-1) = \sum_{i \neq K} \tau^\dagger_i(t-1)$, we have, for any arm $i$, $\tau_i(t-1) \subseteq \tau^\dagger_K(t-1)$, and $\tau_i(t-1) \subseteq \sum_{j \neq K} \tau^\dagger_j(t-1)$. Thus,

$$\sum_{k \in \tau_i(t-1)} \|x_k\|^2_{(\bar{V}^0_{k,K})^{-1}} \leq \sum_{k \in \tau^\dagger_K(t-1)} \|x_k\|^2_{(\bar{V}^0_{k,K})^{-1}} \leq 4d \log \left( 1 + \frac{tL^2}{d\lambda_0} \right), \tag{158}$$

and

$$\sum_{k \in \tau_i(t-1)} \|x_k\|^2_{\left(\bar{V}^0_{k,I^\dagger_k}\right)^{-1}} \leq \sum_{i \neq K} \sum_{k \in \tau^\dagger_i(t-1)} \|x_k\|^2_{(\bar{V}^0_{k,i})^{-1}} \leq 4(K-1)d \log \left( 1 + \frac{tL^2}{d\lambda_0} \right). \tag{159}$$

By combining Equation 156, Equation 158 and Equation 159 and when $K \geq 3$, we have

$$
\sum_{k \in \tau_i(t-1)} \left( 2k_\mu \beta_{k,K}^0 \|x_k\|_{(V_{k,K}^0)^{-1}} + 2k_\mu \beta_{k,I_k^\dagger}^0 \|x_k\|_{\left(V_{k,I_k^\dagger}^0\right)^{-1}} \right)^2
$$

$$
\leq \sum_{k \in \tau_i(t-1)} 2 \left( 2k_\mu \beta_{k,K}^0 \|x_k\|_{(V_{k,K}^0)^{-1}} \right)^2 + \sum_{k \in \tau_i(t-1)} 2 \left( 2k_\mu \beta_{k,I_k^\dagger}^0 \|x_k\|_{\left(V_{k,I_k^\dagger}^0\right)^{-1}} \right)^2
$$

$$
\leq \left( 2\phi_K \frac{k_\mu LS + R}{c_\mu} \right)^2 \left( \log \frac{K}{\delta} + d \log \left( 1 + \frac{L^2 t}{\lambda_0 d} \right) \right) \times 8k_\mu^2 \times 16d^2 \log \left( 1 + \frac{tL^2}{d\lambda_0} \right)
$$

$$
+ \left( 2\frac{K_\mu}{c_\mu \alpha} \frac{k_\mu LS + R}{c_\mu} \right)^2 \left( \log \frac{K}{\delta} + d \log \left( 1 + \frac{L^2 t}{\lambda_0 d} \right) \right) \times 8k_\mu^2 \times 16d^2 (K-1) \log \left( 1 + \frac{tL^2}{d\lambda_0} \right)
$$

$$
\leq 128 k_\mu^2 d^2 \left( \frac{2k_\mu LS + 2R}{c_\mu} \right)^2 \left( \log \frac{K}{\delta} + d \log \left( 1 + \frac{L^2 t}{\lambda_0 d} \right) \right) \log \left( 1 + \frac{tL^2}{d\lambda_0} \right) \left( (K-1) \frac{k_\mu^2}{c_\mu^2 \alpha^2} + (1 + \frac{k_\mu}{c_\mu})^2 \right)
$$

$$
\leq 128 k_\mu^2 d^2 \left( \frac{2k_\mu LS + 2R}{c_\mu} \right)^2 \left( \log \frac{K}{\delta} + d \log \left( 1 + \frac{L^2 t}{\lambda_0 d} \right) \right) \log \left( 1 + \frac{tL^2}{d\lambda_0} \right) \times 2K \frac{k_\mu^2}{c_\mu^2 \alpha^2}.
$$

$$(160)$$

In summary, we have

$$
|x_t^\top \hat{\theta}_{t,i} - x_t^\top (1-\alpha)\theta_K|
$$
$$
\leq \frac{2k_\mu LS + 2R}{c_\mu} \left( 1 + \frac{16 k_\mu^2 d}{c_\mu \alpha} \sqrt{K \log \left( 1 + \frac{tL^2}{d\lambda_0} \right)} \right) \sqrt{\log \frac{K}{\delta} + d \log \left( 1 + \frac{L^2 t}{\lambda_0 d} \right)} \|x\|_{\bar{V}_{t,i}^{-1}}.
$$

$$(161)$$

### E.7  Proof of Theorem 4

For round $t$ and context $x_t$, if UCB-GLM pulls arm $i \neq K$, we have

$$
x_t^\top \hat{\theta}_{t,K} + \beta_{t,K} \sqrt{x_t^\top \bar{V}_{t,K}^{-1} x_t} \leq x_t^\top \hat{\theta}_{t,i} + \beta_{t,i} \sqrt{x_t^\top \bar{V}_{t,i}^{-1} x_t}.
$$

Recall $\beta_{t,i} = \frac{4R}{c_\mu} \sqrt{\log \frac{K}{\delta} + d \log \left( 1 + \frac{L^2 N_i(t)}{\lambda_0 d} \right)}$.

Since the attacker does not attack the target arm, the confidence bound of arm $K$ does not change and $x_t^\top \theta_K \leq x_t^\top \hat{\theta}_{t,K} + \beta_{t,K} \sqrt{x_t^\top V_{t,K}^{-1} x_t}$ holds with probability $1 - \frac{\delta}{K}$.

Thus, by Lemma Equation 8,

$$
x_t^\top \theta_K \leq x_t^\top \hat{\theta}_{t,i} + \beta_{t,i} \sqrt{x_t^\top V_{t,i}^{-1} x_t}
$$
$$
\leq x_t^\top (1-\alpha)\theta_K + \frac{2k_\mu LS + 2R}{c_\mu} \left( 1 + \frac{16 k_\mu^2 d}{c_\mu \alpha} \sqrt{K \log \left( 1 + \frac{tL^2}{d\lambda_0} \right)} \right) \sqrt{\log \frac{K}{\delta} + d \log \left( 1 + \frac{L^2 t}{\lambda_0 d} \right)} \|x\|_{\bar{V}_{t,i}^{-1}}.
$$

$$(162)$$

By multiplying both sides $\mathbb{1}_{\{I_t=i\}}$ and summing over rounds, we have

$$
\sum_{k=1}^T \mathbb{1}_{\{I_k=i\}} \alpha x_k^\top \theta_K
$$
$$
\leq \sum_{k=1}^T \mathbb{1}_{\{I_k=i\}} \frac{2k_\mu LS + 2R}{c_\mu} \left( 1 + \frac{16 k_\mu^2 d}{c_\mu \alpha} \sqrt{K \log \left( 1 + \frac{tL^2}{d\lambda_0} \right)} \right) \sqrt{\log \frac{K}{\delta} + d \log \left( 1 + \frac{L^2 t}{\lambda_0 d} \right)} \|x\|_{\bar{V}_{t,i}^{-1}}.
$$

$$(163)$$

Here, we use Lemma 11 from (Abbasi-Yadkori et al., 2011) and obtain

$$\sum_{k=1}^{T} \mathbb{1}_{\{I_k=i\}} \|x_k\|_{V_{k,i}^{-1}}^2 \le 2d \log(1 + \frac{N_i(t)L^2}{d\lambda}) \le 2d \log\left(1 + \frac{tL^2}{d\lambda}\right). \tag{164}$$

According to $\sum_{k=1}^{T} \mathbb{1}_{\{I_k=i\}} \|x_k\|_{V_{k,i}^{-1}} \le \sqrt{N_i(t) \sum_{k=1}^{T} \mathbb{1}_{\{I_k=i\}} \|x_k\|_{V_{k,i}^{-1}}^2}$, we have

$$\sum_{k=1}^{T} \mathbb{1}_{\{I_k=i\}} \|x_k\|_{V_{k,i}^{-1}} \le \sqrt{N_i(t) 2d \log\left(1 + \frac{tL^2}{d\lambda}\right)}. \tag{165}$$

Set $\lambda = \lambda_0$, we have $\|x\|_{\bar{V}_{t,i}^{-1}}^2 \le (1 + \frac{\lambda}{\lambda_0})\|x\|_{V_{t,i}^{-1}}^2 \le 2\|x\|_{V_{t,i}^{-1}}^2$. Thus, we have

$$\sum_{k=1}^{T} \mathbb{1}_{\{I_k=i\}} \alpha x_k^\top \theta_K$$
$$\le \frac{2k_\mu LS + 2R}{c_\mu} \left(1 + \frac{16k_\mu^2 d}{c_\mu \alpha} \sqrt{K \log\left(1 + \frac{tL^2}{d\lambda_0}\right)}\right) \sqrt{\log \frac{K}{\delta} + d \log\left(1 + \frac{L^2 t}{\lambda_0 d}\right)} \sqrt{4N_i(t) d \log\left(1 + \frac{tL^2}{d\lambda_0}\right)}, \tag{166}$$

and

$$N_i(t) = \sum_{k=1}^{T} \mathbb{1}_{\{I_k=i\}}$$
$$\le \frac{4d}{(\alpha\gamma)^2} \left(\frac{2k_\mu LS + 2R}{c_\mu}\right)^2 \log\left(1 + \frac{tL^2}{d\lambda_0}\right)$$
$$\times \left(\log \frac{K}{\delta} + d \log\left(1 + \frac{L^2 t}{\lambda_0 d}\right)\right) \left(1 + \frac{16k_\mu^2 d}{c_\mu \alpha} \sqrt{K \log\left(1 + \frac{tL^2}{d\lambda_0}\right)}\right)^2, \tag{167}$$

where $\gamma = \min_{x \in \mathcal{D}} \langle x, \theta_K \rangle$.

