# OpenReview forum: "Action Poisoning Attacks on Linear Contextual Bandits"
_TMLR — Accepted by TMLR_

### Review · Reviewer_5QpP · 2022-12-29

**Summary Of Contributions:**

The paper proposes an attack on linear contextual bandit where the actions chosen by the defender ar modified but the defender is unaware of such change. The goal is to make the defender play an arm K as much as possible with least action modifications needed. The authors present a white box and a black box setting that are different based on whether the adversary knows the coefficients of the linear mean of each arm or not.

**Audience:**

Yes

**Broader Impact Concerns:**

None.

**Claims And Evidence:**

No

**Requested Changes:**

Please see the list of major and minor concerns in weakness and strength. I want to see those fixed.
My changes are more of what is typically a major revision in journals.

**Strengths And Weaknesses:**

This is a paper on a relevant topic and seems to have good results. However, I am concerned with the writing. I believe that the technical stuff is sound, but I have some parts that I do not understand because of the writing - I do not want to be second guessing as a reviewer, hence I believe that these things should be fixed. Below, I have the list of major and minor concerns:

Minor:
1) The sigma algebra is introduced on page 3, but is not used at all in the main paper (only in appendix).
2) last line of Sec 3.1: there is a forall quantification over t, x_t but these do not appear in the statement within the scope of the quantifier
3) Above algorithm 2, the step referred to in brakcets (5,7) do not match the description preceding the bracket.
4) omega(N) is introduced but no reason stated why which is surprising, it is not used immediately but only later on.
5) Referring to specific equation should start with capital E, e.g., Equation 5 and not equation 5 (many instances throughout the paper).
6) Sec 4.3 second line: equation written twice.

Major:
1) T is used for vector transpose as well at time horizon T - this is very confusing.
2) It is better to recognize random variables differently. Related to this, I am not sure whether x (context) is a random variable or sometimes a realized value of the random variable. The sigma algebra introduced on page 3 makes x's a random variable. (In some works, x can be fixed or adversarially chosen chosen http://proceedings.mlr.press/v15/chu11a/chu11a.pdf). Continuing along the same lines, in the proof of Lemma 1, I am not sure how the bounded martingale came about, especially if x's are random variables. This part needs more explanation and clarification. Clearly state what are random variables.
3) The paper is mainly a theory paper, thus it is surprising that all proofs are in appendix and there is not even a proof sketches in the main paper. Please add some proof sketch.

---

> ### Author Response · Authors · 2023-01-05
> **Response to Reviewer 5QpP**
>
> Thank you for your helpful comments and suggestions. We respond to your concerns below.
>
> >T is used for vector transpose as well at time horizon T - this is very confusing.
>
> We thank the reviewer for this suggestion. We have changed the symbol of the vector transpose in the revised paper.
>
> >It is better to recognize random variables differently. Related to this, I am not sure whether x (context) is a random variable or sometimes a realized value of the random variable. The sigma algebra introduced on page 3 makes x's a random variable.
>
> Thank you for your comments regarding the random variables. We agree with the reviewer that the context $x$ can be fixed or adversarially chosen. Our work can suit for the case that the adversarially chosen context. We rewrote the sigma algebra and make the random variables more clear. In Lemma 1, we also added the statement about the random variables.
>
> >The paper is mainly a theory paper, thus it is surprising that all proofs are in appendix and there is not even a proof sketches in the main paper. Please add some proof sketch.
>
> We thank the reviewer for the suggestion. We have added the proof sketch of the two main theorems. The changes are highlighted in the revised version.
>
> ----
> Response to the minor concerns:
>
> 1. The sigma algebra is also used in equation 7 at the page 7 of the main paper. We also added the statement about the random variables before equation 7.
>
> 2. Last line of Sec 3.1 is not clear and may make some misunderstanding. We modified it in the revised version.
>
> 3. We fixed the typos.
>
> 4. We change the order of $\omega(N)$ and $\beta_{t,i}$, where $\beta_{t,i}= \omega(N_i(t))$.
>
> 5. We fixed the typos.
>
> 6. We fixed the typos.

---

### Review · Reviewer_Dc1B · 2022-12-30

**Summary Of Contributions:**

This paper studied a new security threat to linear contextual bandits called action poisoning attacks. The attacker in this paper overwrites the arm selected by the bandit agent so that the generated reward alters. The bandit agent then uses the true arm and the reward of the altered arm to update its policy. Under this setup, the paper studied how to manipulate the selected action such that the bandit agent is misled to prefer some target action over the learning process. Two concrete scenarios are considered --- the white-box scenario where the attacker knows the environment, the true arm coefficients and the bandit algorithm, versus the black-box scenario in which the attacker does not know the arm coefficients. The authors provided attack algorithms in both scenarios that provably achieves T-o(T) target arm selections while inducing only o(T) cumulative attack cost. The authors complimented the theoretical results with solid empirical study to demonstrate that the attack algorithms are indeed effective and efficient on synthetic datasets. The paper exposed a severe security caveats in contextual bandits and helps guide the design of more robust algorithms in related applications.

**Audience:**

Yes

**Broader Impact Concerns:**

This paper has no negative ethical implications.

**Claims And Evidence:**

Yes

**Requested Changes:**

Based on the current draft, I suggest the authors to make the following changes, although I am OK if no changes are made in the final version because those changes can be hard to achieve.

(1). Think about the main assumption 1, and study theoretically (or provide some theoretical insights and arguments) what if the target arm is the worst arm on epsilon fraction of contexts, where epsilon fraction is in terms of the context distribution. How much degradation may happen in terms of attack efficiency?

(2). Think about how to relax the assumption that the expected reward must be positive on all contexts. This is really a strong assumption that barely holds in real-world applications of contextual bandits. I understand that this can be hard to achieve so I am OK if this assumption can not be relaxed, but maybe provide some thoughts on how to address that in follow-up works.

**Strengths And Weaknesses:**

Strengths:

(1). The paper considered a very novel problem in attacking contextual bandits. Prior works were focused mostly on reward poisoning attacks where the attacker perturbs the reward of the selected arm. However, less attention has been paid to the action poisoning scenario, which is equivalently important from a practical perspective. This paper is the first to design action poisoning attack algorithms on contextual bandits, and revealed the security vulnerability of typical contextual bandit algorithms such as LinUCB. This is very novel and important work along this topic.

(2). The paper is technically solid and strong. In both white-box and black-box attack scenarios, the authors designed attack algorithms that provably achieves T-o(T) target action selections while incurring only o(T) cumulative attack cost. A log of Lemmas are derived to achieve the main theoretical results in this paper. I can see that these Lemma are also technically interesting and hard to prove. Furthermore, the authors provided intuitive explanations and remarks after main theoretical results, which better facilitates readers to understand their results. I like these explanations as they make the paper easier to read.

Weaknesses:

(1). I am very concerned about the assumptions in this paper. There are two major assumptions made in this paper. First, the target action is not the worst action on all possible contexts. Although I feel this might be too restrictive at first glance, I was convinced that this assumption is necessary. However, what really makes me worried is the second major assumption that the expected reward of all arms must be strictly positive. This barely holds especially when the context set D is dense, e.g., in high dimensional space, it's very likely that some action coefficient is orthogonal to some context, and as a result, the expected reward is 0. This assumption is also used in proving the main theoretical results (e.g., Theorem 1) to lower bound the expected reward of the target arm. In practice, this may not hold.

(2). The proof of the necessity for Assumption 1 does not seem very rigorous to me. It argues around two situations that the attacker either attacks or does not attack the target arm. Also in the end of the proof it says that in order to boost the expected reward of the target arm by Delta, the cumulative attack cost is at least linear. However, why the attacker has to boost the expected reward of the target arm? Is it possible that the attacker intermittently increases the reward of the target arm (but does not boost the expected reward) and always reduce rewards of non-target arms so that the attack is still efficient? I just feel like the argument is not rigorous enough because it does not consider all possible attacks.

(3). The authors empirically showed that even when the target arm is the worst arm, the attack algorithm designed in this paper can still be efficient. This is great! However, that makes me wonder if the authors should further theoretically study this phenomenon. For example, if the target arm is the worst arm on epsilon fraction of contexts, how much degradation would happen in terms of the attack performance. This might be an interesting research question to study, although I understand that it can be hard to answer.

---

> ### Author Response · Authors · 2023-01-05
> **The concern about the proof of the necessity for Assumption 1**
>
> Thank you for your comments regarding the proof of the necessity for Assumption 1. To make the proof more clear, in the revised paper, we provide additional explanations of the proof of the necessity for Assumption 1. The changes are highlighted in the revised version.
>
> In the proof of the necessity for assumption 1,  we consider a sequence of contexts $\\{x_t\\}_{t \in [T]}$ that $x_t \in D_K$ for all $t < T_K$ and $\tau$ linearly depends on $T$.
>
> In the action poisoning attacks, if the attacker tries to reduce the rewards of non-target arms, he can change the non-target arm to a worse arm or the worst arm. However, the target arm $K$ is the worst arm for all $t < T_K$. The mean rewards of the non-target arm at the contexts $\\{x_t\\}_{t \in [T_K]}$ must be larger or equal to $x_t^\top \theta_K$. If the attacker tries to force the agent to pull the target arm in linear time at the first $T_K$ rounds, the attacker needs to let the post-attack arm $K$ be $\Delta$-optimal than the original arm $K$. This is the reason why the attacker has to boost the expected reward of the target arm.

---

> ### Author Response · Authors · 2023-01-05
> **Rethink about the Assumption 1**
>
> We thank the reviewer for the suggestion. However, using current analysis developed in the paper, it is challenging to formally analyze the relationship between the effectiveness of the action attack strategies and the portion of contexts when the target arm is the worst. On the other hand, we can try to find a new algorithm that can force the agent to choose the target arm when the target arm is not the worst. In the following, we provide our initial ideas for this direction.
>
> In the action attack strategies discussed in the paper, the attacker misleads the agent to obtain an estimate of the non-target arms' coefficient vectors that are close to $(1 - \alpha) \theta_K$. However, this can not be always achieved without assumption 1, since the target arm is the worst at some contexts but $(1 - \alpha) \theta_K$ is worse than the target arm. Action attacks can not force the agent to learn $(1 - \alpha) \theta_K$. We need some new attack strategies. Here, we provide our initial thoughts on theoretical insights to solve this challenge in the white-box attack.
>
> If Assumption 1 does not hold, we set $$\mathcal{D}_K := \{x \in \mathcal{D} : \langle x,\theta_{K} \rangle = \min_{i \in [K]} \langle x,\theta_{i} \rangle \}$$ as the set of contexts where the target arm is the worst. We consider such sequence of contexts $\{x_t\}_{t \in [T]}$ that $x_t \in \mathcal{D}_K$ for all $t < T_K$ and $T_K$ linearly depends on $T$.
>
> We consider a neighborhood of $\mathcal{D}_K$:
> $$\widetilde{\mathcal{D}}_K := \cup_{x \in \mathcal{D}_K} B_r(x),$$ where $B_r(x) = \{x' \in \mathcal{D} : \Vert x-x' \Vert_2 < r\}$. In the white-box attack case, we can find a coefficient vector $\theta^+$ such that $x^\top\theta_K > x^\top\theta^+$ for all context $x \in \mathcal{D}_K$ and $x^\top\theta_K > x^\top\theta^+$ for all context $x \in \mathcal{D}/\widetilde{\mathcal{D}}_K$. Then the attacker force the agent to obtain an estimate of the non-target arms' coefficient vectors that are close to $\theta^+$. Then the target arm is the optimal in $\mathcal{D}/\widetilde{\mathcal{D}}_K$, and the distribution of $\widetilde{\mathcal{D}}_K/\mathcal{D}_K $ is limited.
>
> The proposed attack strategies in this paper may make the target arm to be the optimal in $\mathcal{D}/\widetilde{\mathcal{D}}_K$ with some $r$. Thus, even when the target arm is the worst arm, the attack algorithm designed in this paper can still work.
>
> It is of interest to conduct rigorous analysis of this strategy in the future work. Furthermore, it is of interest to carry out the design and analysis for algorithms for the black box attack strategy under this scenario, which will be more challenging. We will leave it as future work. We have added these discussions in the future work section of the revised paper.

---

> ### Author Response · Authors · 2023-01-05
> **Positive rewards assumption**
>
> Thank you for your comments regarding the positive rewards. We agree with the reviewer that this may be a strong assumption in real-world applications of contextual bandits. We make the positive reward assumption for the formal analysis, our results actually can be generalized to the case with negative rewards as stated in Section 5.4.
>
> Here, we provide some ideas to formally analyze the action attacks without the positive reward assumption. In the proposed action attack strategies, the attacker let the agent obtain an estimate of the non-target arms' coefficient vectors that are close to $(1 - \alpha) \theta_K$. If the rewards are positive, the non-target arms are worse than the target arm. Without the positive reward assumption, we can consider the case that there exists a bias parameter in linear setting. The attacker can let the agent obtain an estimate of the non-target arms' coefficient vectors that are close to $\theta^+$, and we have that $x_t^\top\theta_K - x_t^\top\theta^+ = c$ for all context $x_t$ with some constant $c >0$. The new action attack strategies with the same structure but different choice of $\epsilon_t$ in this paper may still work. We will investigate this rigorously in our future work.

---

### Review · Reviewer_26LQ · 2022-12-30

**Summary Of Contributions:**

From the theoretical perspective, this paper proposed a provable action-based attacking algorithm on the linUCB, as well as a generalized linear model. From the empirical perspective, the author shows the effectiveness of their algorithm on multiple datasets. Their experiments not only include the LinUCB but are also compared with other usual algorithms including $\epsilon$-Greedy, LinTS and a robust linear bandits algorithm.

**Audience:**

Yes

**Claims And Evidence:**

No

**Requested Changes:**

Major concern. The linear contextual bandits settings the author states in their paper are not quite the same as those they compared with. So the author should make it more clear. (This will directly affect my decision. For the current version, I will say reject.)

Other concerns: (These will strengthen the work)

\--As I stated in the theoretical perspective of weakness, some statements and algorithm designs do not have a clear motivation. The author should make those more clear.

\-- Add more empirical results.

**Strengths And Weaknesses:**

Strength:

\- The authors propose an interesting problem -- backbox action-based attacking algorithm, which is more difficult compared to reward poisoning.

\- This paper is the first one discussing this problem in the contextual linear bandits. The theoretical proofs are correct and the empirical results seem convincing. One nontrivial proof is that, in the black-box attack, the attacker needs to simultaneously estimate the minimum arms in a pessimistic way and employ the attacking strategy based on this estimated arm.

Weakness:

\-- First of all, I think it is not quite right to state their model as “linear contextual bandits”, because they only tackle the disjoint linear models scenario in Section 3.1 in (Li et al., 2010) where each arm $i$ has a distinct $\theta_i$. But this setting is not much different from MAB, when people talked about “linear contextual bandits”, they usually referred to a shared $\theta^\*$, which is the simplified version of Section 3.2 (Li et al., 2010). So when goes to the latter scenario, I am not sure whether this algorithm still works well or not. My conjecture is that it might work but maybe not very optimal and requires some more complicated analysis.

Besides, from the theoretical perspective, I found that the motivations of some statements are not very clearly motivated (although correct) and results are relatively incremental.

\-- Eqn. (3) is basically a restatement of Assumption 1. To me, it is unclear what is the intuition behind this $\alpha$, and therefore the final $\alpha$-based result, although correct, is hard to interpret. For example, instead of assuming $ \langle x,\theta_K \rangle > \min_{i \in [K]} \langle x,\theta \rangle$, I wonder why not define  $\Delta = \langle x,\theta_K \rangle > \min_{i \in [K]} \langle x,\theta \rangle$ and state the result in terms of $\Delta$ ? The latter one seems more standard in the linUCB scenario.

\-- Only doing an attack on LinUCB is a little bit weak to me because LinUCB is similar to a hard elimination algorithm -- Once the learner decides some arm is inferior with high probability, it will stop pulling the arm forever.  So the main technique here is just cheating the learner to eliminate all the other arms, which is not very surprising to me.

\-- It is unclear to me why randomness in Eqn.(5) is necessary. Because the LinUCB is a deterministic algorithm, it seems to me totally possible to design a deterministic attacking algorithm to cheat the learner.

From the experiment perspective, first of all, as I stated before, they are not tackling the same setting so I am not sure how they did the experiments.

Also, I think maybe more content can be added to Section 5.2.

\-- The author does not choose Bogunovic et al., 2021) because it does not apply to linear contextual bandits setting. But to me, if the attacking algorithm works for general linear contextual bandits, then it must work for linear bandits with fixed candidate arms. Why not give a test?

\-- The (Ding et al., 2022), like the author stated, has bound $O(C\sqrt{T})$, which is not that good. Why not test on other robust algorithms even in MAB setting which has better $C$ dependent bounds?

---

> ### Author Response · Authors · 2023-01-05
> **About “linear contextual bandits” model statement**
>
> Thank you for pointing this out. We would like to note that in the existing literature, there are several classes of linear contextual bandit models. In the revised paper, we clarified that we study the action attacks on the disjoint linear contextual model, where each arm is associated with its own coefficient vector and at each time the agent observes a contextual vector $x_t$ that is shared with all arms. The disjoint linear contextual model is considered in bandit literature, for example in Li et al., 2010; Kong et al., 2020; Garcelon et al., 2020; Huang et al., 2021, etc. This was made clear in both the abstract and the main body of the paper.
>
> We agree with the reviewer that the disjoint linear model studied in this paper (and related references) is different from the linear contextual bandit model with a shared coefficient. In the linear contextual bandit model with a shared coefficient, the coefficient vector $\theta^*$ is shared with all arms. At each time, each arm $i$ is given a specific contextual vector $x_{i,t}$. These contextual vectors form the decision set. The decision set changes over time and can even be infinite. In this model, adversarial attacks may not always be successful. For example, (Wang et al., 2022) found that some attack goals can never be achieved in linear stochastic bandits due to the shared coefficient. This may also happen in the linear contextual bandit model. It is of interest to conduct rigorous analysis of action attacks in the shared coefficient model in future work. In the revised paper, we have added a discussion about this in the future work section.

---

> > ### Comment · Reviewer_26LQ · 2023-01-19
> > **Feedback**
> >
> > Thanks for clarifying.  I think with this new statement the major concern of the paper has been addressed.

---

> ### Author Response · Authors · 2023-01-05
> **The intuition behind this $\alpha$**
>
> Thank you for the suggestion. We added the discussion of the intuition behind this $\alpha$ after Equation 3. The changes are highlight in the revised version. Equation 3 describes the relationship between the target arm and the worst arm at context $x_t$. The action poisoning attack only indirectly impacts the agent’s rewards. The mean rewards at $x_t$ after the action attacks must be larger or equal to $\min_{i \in [K]}\langle x_t,\theta_{i} \rangle$. Under Equation 3, $(1 - 2\alpha) \langle x_t,\theta_{K} \rangle \ge \min_{i \in [K]}\langle x_t,\theta_{i} \rangle$ at any context $x_t$. Then, with $\langle x_t,\theta_{K} \rangle > 0$, $(1 - \alpha) \langle x_t,\theta_{K} \rangle > \min_{i \in [K]}\langle x_t,\theta_{i} \rangle$ at any context $x_t$. Thus, the attacker can indirectly change the agent's mean reward of the non-target arm to $(1 - \alpha) \langle x_t,\theta_{K}\rangle$.  Then, the agent's estimate of each non-target arm's coefficient vector is close to $(1 - \alpha)\theta_K$, which is worse than the target arm at any context. Equation 3 brings the possibility of successful action attacks.

---

> ### Author Response · Authors · 2023-01-05
> **Response to the concerns about the randomness in Eqn.(5) and the concerns about doing an attack on LinUCB**
>
> In this paper, we use LinUCB as an example to illustrate the effectiveness of our attack strategies. Our white-box attack strategy works for any no-regret disjoint linear contextual bandit algorithms as stated in Proposition 1.
>
> The randomness in Equation 5 is necessary for the results in Proposition 1, which holds for any no-regret disjoint linear contextual bandit algorithms (not limited to LinUCB). Recall that the goal of Equation 5 is to force the agent to obtain an estimate of a non-target arm $i$'s coefficient vector that is close to $(1 - \alpha) \theta_K$. There are two reasons why the randomness in Equation 5 is necessary to achieve the goal.
>
> (1) We would like to design a white-box attack strategy that works for any no-regret algorithm.  If the attacker knows the agent's algorithm, the algorithm's parameters and even the random seed of the agent's algorithm, the attacker can achieve the goal by some action attacks without randomness.
>
> (2) To force any agent to obtain an estimate of a non-target arm $i$'s coefficient vector that is close to $(1 - \alpha) \theta_K$, the attacker needs to control the mean reward of the non-target arm $i$ in $x_t^\top(1 - \alpha) \theta_K$ when at context $x_t$. There may not exist such an arm $j$ that $x_t^\top(1 - \alpha) \theta_K = x_t^\top \theta_j$. A combination of two arms in the form of Equation 5 is necessary in such a case, and hence randomness is required.
>
> In summary, we agree with the reviewer that, if we only focus on attacking LinUCB learner, it totally possible to design a deterministic attacking algorithm. However, in this paper, we assume that the attacker has no knowledge of the agent's algorithm. With the randomness, the proposed attack strategies are also effective for other learning algorithms such as epsilon-greedy and LinTS.

---

> ### Author Response · Authors · 2023-01-05
> **Response to the concerns about empirical results**
>
> We thank the reviewer for the suggestion. We added more simulation results in Section 5.2.
>
> In empirical results, we used the disjoint version of LinUCB, LinTS and epsilon-greedy. The algorithms separately estimate the $K$ coefficient vectors. The detailed algorithm of LinUCB is stated in Algorithm 2. We have added more details in the simulation section of the revised paper. The changes are highlighted in the revised paper.

---

### Decision · Action_Editors · 2023-02-19

**Recommendation:** Accept as is

**Comment:**

Based on discussions among the reviewers and the authors, the reviewers think that although the contribution and technical innovation might be moderate, it:
* studies a timely and interesting topic
* contains solid results (some assumptions may be unrealistic but necessary from a technical point of view)
* has novel and interesting methodology in theory and practice

The revised version of the paper addressed the reviewers' concerns about problem setup, necessity of some assumptions, details of simulations, and general presentation.

Overall it satisfies TMLR's acceptance criteria.

**Audience:**

Yes: the reviewers are in agreement that action poisoning in contextual bandits is a timely and practically relevant topic.

**Claims And Evidence:**

Yes: the paper provided solid theoretical analysis and empirical evidence to demonstrate the main claims.